# Convergence of Distributed Adaptive Optimization with Local Updates

**Ziheng Cheng**
University of California, Berkeley
ziheng_cheng@berkeley.edu

**Margalit Glasgow**
Massachusetts Institute of Technology
mglasgow@mit.edu

## Abstract

We study distributed adaptive algorithms with local updates (intermittent communication). Despite the great empirical success of adaptive methods in distributed training of modern machine learning models, the theoretical benefits of local updates within adaptive methods, particularly in terms of reducing communication complexity, have not been fully understood yet. In this paper, for the first time, we prove that *Local SGD* with momentum (*Local* SGDM) and *Local* Adam can outperform their minibatch counterparts in convex and weakly convex settings in certain regimes, respectively. Our analysis relies on a novel technique to prove contraction during local iterations, which is a crucial yet challenging step to show the advantages of local updates, under generalized smoothness assumption and gradient clipping strategy.

## 1 Introduction

Leveraging parallelism is crucial in accelerating the training of modern machine learning models for large scale optimization problems. In distributed environments such as large data-centers or in the federated learning setting, where the devices working together are spread apart, communication between the distributed workers is a key bottleneck. In this work, we consider the task of

$$\min_{x \in \mathbb{R}^d} f(x) := \mathbb{E}_{\xi \sim \mathcal{D}}[F(x; \xi)]. \tag{1.1}$$

in a distributed setting with $M$ workers. Each worker has access to $f$ via the stochastic gradient oracle $\nabla F(x; \xi)$, where $\xi$ is independently drawn from the distribution $\mathcal{D}$. In federated learning, this is known as the *homogeneous* setting, since all workers draw from the same data distribution.

Perhaps the simplest algorithm for distributed optimization is distributed *minibatch stochastic gradient descent (SGD)*, in which at each iteration, each worker computes a minibatch of gradients, and a gradient step is taken by averaging the gradient computed among the $M$ workers. However, such an algorithm requires communicating at each gradient step, which may be expensive. Thus numerous works have proposed distributed algorithms with less frequent communication. A popular and well-studied algorithm is *Local* SGD, also known as FedAvg (McMahan et al., 2017), where each worker runs SGD independently and periodically synchronizes with others by averaging the iterates.

Despite the success of *Local* SGD in federated learning (McMahan et al., 2017), it may not exhibit good performance when training Transformer-based large language models (LLMs). Many empirical studies suggest that adaptive methods (*e.g.*, Adam (Kingma & Ba, 2014)) are much better suited for natural language processing than vanilla SGD (Goodfellow et al., 2016; Zhang et al., 2020; Kunstner et al., 2023; Pan & Li, 2023). Furthermore, as shown in Zhang et al. (2019; 2020), language models tend to have unbounded global smoothness and heavy-tailed noise, which may also contribute to the worse performance of SGD. Parallelizing adaptive methods requires an even more expensive communication cost since additional terms, such as the momentum or the Adam denominator, need to be synchronized. Previous works on distributed adaptive optimization have utilized compression and quantization techniques to address this issue (Bernstein et al., 2018; Wangni et al., 2018; Wang et al., 2023). While Douillard et al. (2023) has shown the great empirical success of *Local* Adam, to the best of our knowledge, there are no theoretical results trying to improve training efficiency or adaptive methods from the perspective of intermittent communication.

In this paper, we investigate **distributed adaptive optimization algorithms in the homogeneous regime**, in order to establish theoretical guarantees for the benefits of local iterations in reducing communication complexity. We focus on the convex or weakly convex setting[1].

We propose a distributed version of Adam, namely, *Local* Adam, with gradient clipping. Our algorithm also reduces to *Local* SGD with momentum (*Local* SGDM), with some specific hyper-parameter choices.

- In Theorem 1,2, we establish the first convergence guarantee for *Local* SGDM in the convex setting, which outperforms the convergence rate of *Minibatch* SGDM. The rate we obtain is in line with the rate of *Local* SGD (Woodworth et al., 2020a) .
- In Theorem 3, we establish a convergence rate for *Local* Adam in the weakly convex setting. We show that *Local* Adam can provably improve communication efficiency compared to its minibatch baseline.

For the first time, we are able to show the benefits of local iterations for the two commonly used algorithms, SGDM and Adam. This suggests that one can improve the training efficiency of large models by using intermittent communication.

Additionally, our results hold under generalized smoothness and heavy-tailed noise. Our result is the first high probability bound for distributed optimization algorithms with local updates, to the best of our knowledge. The conventional in-expectation rate seems fail to capture some important properties like heavy/light tailed noise distribution. The high probability convergence guarantee can sometimes be more informative and useful in practice (Gorbunov et al., 2020).

As for technical contribution, we use a **novel technique to prove contraction for adaptive methods**, which bounds the consensus error between the iterates at different workers. This is a key step in proving benefits of local updates. Different from *Local* SGD, our update direction involves momentum or even distorted momentum due to the denominator in *Local* Adam, making it challenging to disentangle these accumulated stochastic gradients. To address this issue, we define and analyze an auxiliary sequence which is conditionally independent of the latest stochastic gradient and thus can construct a martingale. We will introduce the technique in more details in Section 5.

## 1.1 ORGANIZATION

Section 2 provides the most related work to ours. Section 3 provides the problem setup, assumptions and the *Local* Adam algorithm. We then show our main results for *Local* SGDM in Section 4.1 and *Local* Adam in Section 4.2. Finally, in Section 5, we present the proof sketch of *Local* Adam, highlighting the technical challenges and our solution.

## 1.2 NOTATION

Let $\| \cdot \|$ be the standard Euclidean norm of a vector or the spectral norm of a matrix. For any $x, y \in \mathbb{R}^d$, the expressions $x + y, x \odot y, \dfrac{x}{y}$ stand for coordinate-wise sum, product and division, respectively. And $x \preceq y$ means each coordinate of $x - y$ is no greater than $0$. Furthermore, we use $x^2, \sqrt{x}, |x|$ to denote the coordinate-wise square, square root and absolute value. We use $\mathbb{E}_m[X_m]$ to denote the average $\dfrac{1}{M} \sum_{m=1}^{M} X_m$. The coordinate-wise clipping operator $\mathbf{clip}(\cdot, \rho) : \mathbb{R}^d \to \mathbb{R}^d$ is defined as $[\mathbf{clip}(X, \rho)]_i = \mathrm{sgn}([X]_i) \cdot \min\{|X_i|, \rho\}$. We use $[N]$ to denote the set $\{1, 2, \ldots, N\}$. For a subset $\Omega_0 \subset \mathbb{R}^d$, let $\mathbf{conv}(\cdot)$ denote the convex hull of $\Omega_0$ and $\mathbf{B}_{R_0}(\Omega_0)$ denote the neighborhood of $\Omega_0$ with radius $R_0$. Finally, we use standard $\mathcal{O}(\cdot), \Omega(\cdot), \Theta(\cdot)$ to omit constant factors and $\tilde{\mathcal{O}}(\cdot)$ to omit logarithmic factors.

---

[1] Under the stronger assumptions of 3rd-order smoothness (Glasgow et al., 2022) and mean smoothness (Patel et al., 2022), there are demonstrated advantages of local iterations in the non-convex setting. While our theoretical results are for the convex or weakly convex setting, it is likely that local iterations are advantageous in practice for non-convex objectives, just in the same way *Local* SGD has been shown to be advantageous in practice for non-convex objectives (McMahan et al., 2017).

## 2 RELATED WORK

**Theoretical benefits of local updates in distributed optimization.** Algorithms with local updates have been used among practitioners for a long time to reduce communication complexity (McMahan et al., 2017). In the homogeneous and convex setting, *Local* SGD and its variants have been shown to outperform the minibatch baseline, for a fixed amount of gradient computations and communication rounds. Woodworth et al. (2020a) is the first to show that *Local* SGD can provably outperform *Minibatch* SGD. Yuan & Ma (2020) develops FedAC to further accelerate *Local* SGD. In the heterogeneous case, Woodworth et al. (2020b) demonstrates the advantages of *Local* SGD when heterogeneity is very low. Algorithms with local updates have also been studied in the non-convex setting (Karimireddy et al., 2020b; Yang et al., 2021; Glasgow et al., 2022), including momentum-based and adaptive methods (Reddi et al., 2020; Karimireddy et al., 2020a), though no advantage of local iterations over minibatch has been shown, without non-standard assumptions such as 3rd-order smoothness. Notably, Liu et al. (2022) is one closely related work to ours, which considers *Local* SGD with gradient clipping in homogeneous and non-convex setting and claims that the convergence guarantee is better than naive parallel of centralized clipped-SGD. However, it still cannot outperform minibatch baseline (with batch size $K$ for each worker in each round) and thus fails to demonstrate the benefits of local iterations.

**Convergence of centralized Adam.** Adam was first proposed by Kingma & Ba (2014) with convergence guarantee in online convex optimization. However, Reddi et al. (2019) found a gap in the original analysis of Adam and constructed a counter example to show its divergence. Since then, many works have developed convergence analyses of Adam with various assumptions and hyperparameter settings. Guo et al. (2021) assumed the denominator is bounded from below and above by two constants, which typically requires a bounded gradient assumption or the AdaBound variant (Luo et al., 2019). Défossez et al. (2020) assumed a bounded gradient and their convergence guarantee depends on $\textbf{poly}(d)$. Zhang et al. (2022b); Wang et al. (2022) considered a finite sum setting and showed that Adam converges to the neighborhood of stationary points. One closely related work to ours is Li et al. (2024c), which established a high probability bound without a bounded gradient assumption. However they assumed that noise is bounded almost surely. Another recent work (Wang et al., 2024) provided a guarantee of $\mathcal{O}\left(1/\varepsilon^4\right)$ with dependence on $\textbf{poly}(d)$. Beyond the guarantees on gradient norm given by non-convex analyses, no stronger bounds (*e.g.*, on function error) are known for Adam in the convex case.

**Convergence of distributed adaptive algorithms.** In the federated learning literature, Reddi et al. (2020) introduced a framework, FedOPT, to leverage both worker optimizer and server optimizer. Many works explored adaptive server optimizer while fixing worker side as vanilla SGD. The theoretical results of local adaptive algorithms are much fewer. Some works have studied *Local* Adam and *Local* AMSGrad with fixed momentum state during local iterations (Karimireddy et al., 2020a; Chen et al., 2020; Zhao et al., 2022). They also needed stringent assumptions such as a huge batch size depending on the inverse of target error, bounded stochastic gradients, vanishing difference between denominator, etc., which are not standard. Wang et al. (2021) explored adaptive worker optimizer based on centralized algorithm, where the state of worker optimizer changes in local updates. However, their analysis relied on an explicit assumptions (Wang et al., 2021, Assumption 1) on the contraction property of worker optimizer. Some recent works (Li et al., 2024a; Anyszka et al., 2024) discussed Polyak stepsizes with an exact local proximal operator, which is inaccessible in most cases by gradient-based optimizers. To the best of our knowledge, there is no end-to-end convergence guarantee for distributed adaptive algorithms with local iterations.

## 3 PROBLEM SETUP

Consider the distributed optimization problem

$$\min_{x \in \mathbb{R}^d} f(x) := \mathbb{E}_{\xi \sim \mathcal{D}}[F(x; \xi)]. \tag{3.1}$$

Here $\mathcal{D}$ is the data distribution and $f$ is the population loss function. We consider a setting with $M$ parallel workers, and a budget of $R$ total communication rounds, and $T$ total gradient computations at each worker. We will describe the implementation of the *local* and *minibatch* versions of

a centralized algorithm $\mathcal{A}$, which uses a single stochastic gradient in each iteration. And these are illustrated in Figure 1.

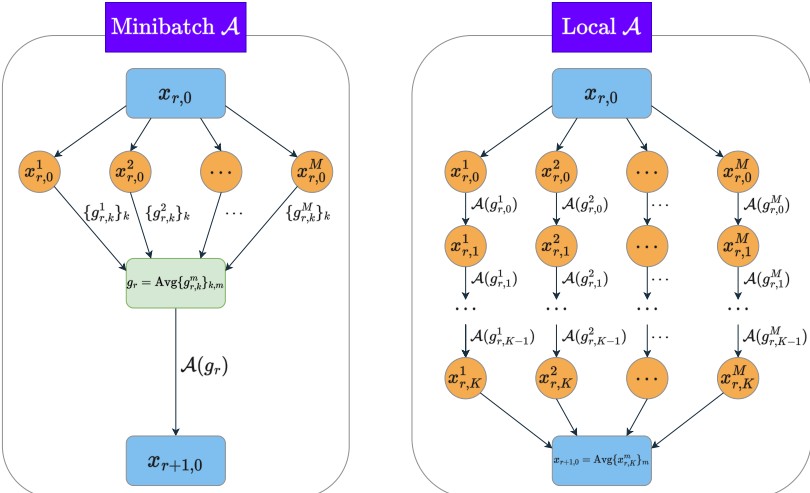

Figure 1: *Minibatch $\mathcal{A}$ v.s. Local $\mathcal{A}$* in one communication round. Minibatch version computes the average of all $KM$ gradients and then executes one step of $\mathcal{A}$, while local version runs $\mathcal{A}$ independently for $K$ steps at each worker.

In the *local* version of algorithm $\mathcal{A}$, in each round $r$ of the $R$ total communication rounds, each worker $m$ independently executes $K = T/R$ steps of local updates (according to the algorithm $\mathcal{A}$). For a worker $m$, we denote the $k$th gradient computed in round $r$ by $g_{r,k}^m$. Then the $M$ workers synchronize the iterates and related momentum state. We use *Minibatch $\mathcal{A}$* to denote a distributed implementation of $\mathcal{A}$ run for $R$ rounds, where $KM$ stochastic gradients are computed and averaged at each step. This is a fair baseline to compare the local update algorithms to, since the number of gradient calls and communication rounds are the same.

*Local* Adam is shown in Algorithm 1, which is a natural extension of centralized Adam (Kingma & Ba, 2014). The stochastic gradient is clipped by an coordinate-wise clipping operator with threshold $\rho$. After $K$ steps of local updates, all the workers average their current iterates $x_t^m$, their first order momentum $u_t^m$, and their second order momentum $v_t^m$. These averaged quantities become the values used at the beginning of the next local round. Note that there are two slight differences from original Adam. First, we do not involve bias correction here, *i.e.*, $u_t^m$ and $v_t^m$ are not divided by $1 - \beta_1^t$ or $1 - \beta_2^t$, respectively. Second, $\lambda$ in the denominator is in the square root, while it is outside of the denominator in original Adam. These modifications do not harm the spirit of Adam and are made for the convenience of analysis.

## 3.1 ASSUMPTIONS

Throughout this work, we will use the following assumptions.

**Assumption 1** (Lower-boundedness). *$f$ is closed, twice continuously differentiable and* $\inf_{x \in \mathbb{R}^d} f(x) =: f(x_*) =: f_* > -\infty$.

**Assumption 2** (Smoothness). *There exists some set $\Omega \subset \mathbb{R}^d$ and $L > 0$, such that for any $x, y \in \Omega$,*

$$\|\nabla f(x) - \nabla f(y)\| \le L\|x - y\|, \tag{3.2}$$

$$\|\nabla f(x)\|^2 \le 2L(f(x) - f_*). \tag{3.3}$$

Similar to Sadiev et al. (2023), we only requires some properties of $f$ on a subset $\Omega$ of $\mathbb{R}^d$, since we can prove that all the iterates will not leave this subset with high probability. In contrast, the typical smoothness assumption requires (3.2) on the entire domain.

---

**Algorithm 1** *Local* Adam

---

**Require:** initial model $x_0$, learning rate $\eta$, momentum $\beta_1, \beta_2 \in [0, 1)$

    Set $x_{0,0}^m = x_0$, $u_{0,-1}^m = 0$, $v_0 = 0$ for each worker $m \in [M]$

    **for** $r = 0, \cdots, R - 1$ **do**

        **for** each worker $m \in [M]$ in parallel **do**

            **for** $k = 0, \cdots, K - 1$ **do**

                $g_{r,k}^m = \nabla F(x_{r,k}^m; \xi_{r,k}^m)$, $\widehat{g_{r,k}^m} = \mathbf{clip}(g_{r,k}^m, \rho)$        ▷ Compute clipped stochastic gradient

                $u_{r,k}^m = \beta_1 u_{r,k-1}^m + (1 - \beta_1)\widehat{g_{r,k}^m}$               ▷ Update 1st-order momentum

                $v_{r,k}^m = \beta_2 v_{r,k-1}^m + (1 - \beta_2)\widehat{g_{r,k}^m} \odot \widehat{g_{r,k}^m}$        ▷ Update 2nd-order momentum

                $x_{r,k+1}^m = x_{r,k}^m - \dfrac{\eta}{\sqrt{v_{r,k}^m + \lambda^2}} \odot u_{r,k}^m$             ▷ Update model

            **end for**

        **end for**

        $x_{r+1,0}^m = \mathbb{E}_m[x_{r,K}^m]$, $u_{r+1,-1}^m = \mathbb{E}_m[u_{r,K-1}^m]$, $v_{r+1,-1}^m = v_{r+1} := \mathbb{E}_m[v_{r,K-1}^m]$

        ▷ Communicate and average

    **end for**

---

There are many works (Zhang et al., 2019; Crawshaw et al., 2022; Faw et al., 2023; Wang et al., 2022; Li et al., 2024c) that make weaker smoothness assumptions (typically called "generalized smoothness"), most of which are in the form of $(L_0, L_1)$-smoothness:

$$\|\nabla^2 f(x)\| \le L_0 + L_1\|\nabla f(x)\|, \ \forall x \in \mathbb{R}^d. \tag{3.4}$$

Li et al. (2024b) considers an extension called $\ell$-smoothness, which replaces the linear function of $\|\nabla f\|$ in the right hand side of (3.4) with a sub-quadratic function $\ell(\cdot)$. As pointed out in Li et al. (2024b, Corollary 3.6), all of these will induce Assumption 2 if $\Omega$ is some level-set of the objective function[2]. Therefore, we directly use this more general assumption to get cleaner results.

**Assumption 3** (Bounded $\alpha$-moment noise). *There exists some set $\Omega \subset \mathbb{R}^d$, $\alpha \ge 4$ and constant vector $\boldsymbol{\sigma} \succeq 0$ such that for any $x \in \Omega$,*

$$\mathbb{E}_{\xi \sim \mathcal{D}}|\nabla F(x; \xi) - \nabla f(x)|^\alpha \preceq \boldsymbol{\sigma}^\alpha. \tag{3.5}$$

*Let $\sigma_\infty := \|\boldsymbol{\sigma}\|_\infty = \max_i\{\sigma_i\}$, $\sigma := \|\boldsymbol{\sigma}\| = \left(\sigma_1^2 + \cdots + \sigma_d^2\right)^{1/2}$.*

**Remark 1.** *To get a high probability bound under generalized smoothness, the assumption on stochastic noise is crucial. Light-tailed noise with bounded exponential moment (e.g., bounded, sub-exponential, sub-gaussian) are considered in Harvey et al. (2019); Li & Orabona (2020); Li et al. (2024c). There are also attempts for heavy-tailed noise with finite $\alpha$-moment (Gorbunov et al., 2020; Cutkosky & Mehta, 2021; Faw et al., 2023). In the most literatures studying heavy-tailed noise, they restrict to the case where $1 < \alpha \le 2$. However, in the matter of getting a logarithmic dependence on $1/\delta$, where $\delta$ is the confidence level, the essence lies in whether we assume bounded exponential moment or just polynomial moment (see Appendix E for detailed discussions). For convenience, we only consider $\alpha \ge 4$ in this paper, but our analysis methods can be extended to the case where $\alpha < 4$ with some additional technical computations.*

**Remark 2** (Noise of minibatch). *It follows from Petrov (1992) that if the gradient is estimated by a batch of i.i.d samples with batch size $N$, the $\alpha$-moment of noise has upper bound of:*

$$\mathbb{E}_{\{\xi_i\}\overset{i.i.d}{\sim}\mathcal{D}}\Big|\frac{1}{N}\sum_{i=1}^{N}\nabla F(x; \xi_i) - \nabla f(x)\Big|^\alpha \preceq c(\alpha)\big(\boldsymbol{\sigma}/\sqrt{N}\big)^\alpha, \tag{3.6}$$

*where $c(\alpha)$ is a problem-independent constant. It is easy to see that this bound is tight when the noise is Gaussian. Therefore, to get the rate for batch size $N$, we can just simply replace $\boldsymbol{\sigma}$ with $\boldsymbol{\sigma}/\sqrt{N}$ (up to a constant depending on $\alpha$) in the original convergence guarantee for batch size 1.*

---

[2]*e.g., if $\Omega \subset \{x : f(x) - f_* \le \Delta\}$, then $(L_0, L_1)$-smoothness would imply Assumption 2 for $L \asymp L_0 + L_1^2\Delta$. Note that we may not obtain the optimal dependence on $L_0, L_1$ in this way though.*

## 4 MAIN RESULTS

In this section, we provide our main results for *Local* Adam and its simplified version: *Local* SGDM. For the first time, we will be able to show the benefits of local iterations for the two algorithms, compared with their minibatch baselines in certain regime of $M, K, R$.

### 4.1 LOCAL SGDM

Before getting into *Local* Adam, we start with a simpler yet also important algorithm: *Local* SGD with momentum. Note that when $\beta_2 = 1, \lambda = 1$, Algorithm 1 will reduce to *Local* SGDM. We restate the complete version of *Local* SGDM in Algorithm 2 in Appendix C.

**Assumption 4** (Convexity). *There exists some set $\Omega \subset \mathbb{R}^d$ and constant $\mu \geq 0$ such that $f$ is $\mu$-strongly convex on $\Omega$, i.e., for any $x, y \in \Omega$,*

$$\langle \nabla f(x) - \nabla f(y), x - y \rangle \geq \mu \|x - y\|^2, \tag{4.1}$$

$$f(y) \geq f(x) + \langle \nabla f(x), y - x \rangle + \frac{\mu}{2}\|x - y\|^2. \tag{4.2}$$

Let $D_0 := \|x_0 - x_*\|$. Now we state the results for *Local* SGDM below. Notably, our results are the first convergence guarantee for distributed SGDM with local updates in (strongly) convex setting.

**Theorem 1** (Strongly convex, full version see Theorem C.4). *Let Assumption 1, 2, 3, 4 hold for $\Omega := \{\|x - x_*\| \leq \sqrt{3}D_0\}$ and $\mu > 0$. Further assume that $K \gtrsim \log \dfrac{MKR}{\delta}$, $1 - \beta_1 = \Omega(1)$ and $\|\boldsymbol{\sigma}\|_{2\alpha} d^{\frac{1}{2} - \frac{1}{2\alpha}} = \mathcal{O}(\sigma)$. Then with probability no less than $1 - \delta$, Local SGDM yields*

$$f(\hat{x}) - f_* \leq \exp\left(-\Theta\left(\frac{\mu K R}{L}\right)\right) + \tilde{\mathcal{O}}\left(\frac{\sigma^2}{\mu MKR} + \frac{L\sigma^2}{\mu^2 KR^2} + \frac{\sigma^2}{\mu}\left(\frac{L^{\frac{1}{2}}}{\mu^{\frac{1}{2}}KR}\right)^{\frac{2(\alpha-1)}{\alpha}}\right). \tag{4.3}$$

**Theorem 2** (Convex, full version see Theorem C.5). *Let Assumption 1, 2, 3, 4 hold for $\Omega := \{\|x - x_*\| \leq \sqrt{3}D_0\}$ and $\mu = 0$. Further assume that $K \gtrsim \log \dfrac{MKR}{\delta}$, $1 - \beta_1 = \Omega(1)$ and $\|\boldsymbol{\sigma}\|_{2\alpha} d^{\frac{1}{2} - \frac{1}{2\alpha}} = \mathcal{O}(\sigma)$. Then with probability no less than $1 - \delta$, Local SGDM yields*

$$f(\hat{x}) - f_* \leq \tilde{\mathcal{O}}\left(\frac{LD_0^2}{KR} + \frac{\sigma D_0}{\sqrt{MKR}} + \frac{L^{\frac{1}{3}}\sigma^{\frac{2}{3}}D_0^{\frac{4}{3}}}{K^{\frac{1}{3}}R^{\frac{2}{3}}} + D_0\left(\frac{(LD_0)^{\frac{1}{2}}\sigma^{\frac{\alpha}{\alpha-1}}}{KR}\right)^{\frac{2(\alpha-1)}{3\alpha-1}}\right). \tag{4.4}$$

**Remark 3** (Confidence level $\delta$). *$\delta$ does not appear in the bound since we have $\log \dfrac{1}{\delta}$ dependence.*

Our method can also be applied to *Minibath* SGDM (by substituting $M, K$ with 1 and $\sigma$ with $\dfrac{\sigma}{\sqrt{MK}}$; see Remark 2), whose convergence guarantee is

$$f(\hat{x}) - f_* \lesssim \begin{cases} \exp\left(-\Theta\left(\dfrac{\mu R}{L}\right)\right) + \tilde{\mathcal{O}}\left(\dfrac{\sigma^2}{\mu MKR}\right), & \text{if } \mu > 0, \\ \tilde{\mathcal{O}}\left(\dfrac{LD_0^2}{R} + \dfrac{\sigma D_0}{\sqrt{MKR}}\right), & \text{otherwise.} \end{cases} \tag{4.5}$$

This rate matches the well-known in-expectation lower bound on the convergence rate of *Minibatch* SGD (up to logarithmic factors). In fact, our analysis improves the state-of-the-art rate for strongly-convex SGDM (given in Liu et al. (2020b)), which has a stochastic term as $\tilde{\mathcal{O}}\left(\dfrac{L\sigma^2}{\mu^2 MKR}\right)$. In the convex setting, our rate is consistent with the state-of-the-art centralized in-expectation bound of SGDM in Sebbouh et al. (2021). Further notice that the last term in both (4.3) and (4.4) is due to the bias of gradient clipping and would be negligible as long as $K^{\alpha-2} \gtrsim \dfrac{\mu R^2}{L}$ or $K^{\frac{3\alpha-5}{2}} \gtrsim \dfrac{\sigma R^2}{LD_0}$. In this case, our guarantee for *Local* SGDM is aligned with the rate of *Local* SGD in Woodworth et al. (2020a); Khaled et al. (2020) up to logarithmic factor. Therefore, we can see the benefits of local iterations in the large $M$ and large $K$ regime compared to minibatch baseline.

We defer the complete version and detailed proof to Appendix C.

## 4.2 LOCAL ADAM

The convergence of Adam is much more difficult to prove. Reddi et al. (2019) pointed out that the original proof in Kingma & Ba (2014) in centralized convex setting was incorrect. Therefore, the convergence of Adam in for convex function is of independent interest and beyond our scope. Instead, we turn to consider Adam in the weakly convex setting.

**Assumption 5** (Weak convexity). *There exists constant $\tau > 0$ such that $f$ is $\tau$-weakly convex, i.e., for any $x, y \in \mathbb{R}^d$,*

$$\langle \nabla f(x) - \nabla f(y), x - y \rangle \geq -\tau \|x - y\|^2, \tag{4.6}$$

$$f(y) \geq f(x) + \langle \nabla f(x), y - x \rangle - \frac{\tau}{2}\|x - y\|^2, \ \nabla^2 f(x) \succeq -\tau I_d. \tag{4.7}$$

Note that $L$-smoothness implies that Assumption 5 always holds with $\tau = L$. Also note that here we assume the weak convexity holds in $\mathbb{R}^d$ for technical simplicity. Let $H_r = \mathbf{diag}(\sqrt{v_r + \lambda^2}) \succeq \lambda I_d$ and $\Delta := f(x_0) - f_*$. Furthermore, define an auxiliary sequence $\{z_{r,k}^m\}$ as:

$$z_{r,k+1}^m = \begin{cases} (x_{r,k+1}^m - \beta_1 x_{r,k}^m)/(1 - \beta_1) & \text{if } k \neq K - 1, \\ (x_{r,k+1}^m - \beta_1 \overline{x}_{r,k})/(1 - \beta_1) & \text{otherwise.} \end{cases} \tag{4.8}$$

Let $\overline{z}_{r,k} := \mathbb{E}_m[z_{r,k}^m]$. Now we state the main result of *Local* Adam below (see Theorem D.2 for more general results on Moreau envelope).

**Theorem 3** (Full version see Theorem D.3). *Let Assumption 1, 2, 3, 5 hold for $\Omega = \mathbf{conv}(\mathbf{B}_{R_0}(\Omega_0))$, where $\Omega_0 := \{f(x) - f_* \leq 4\Delta\}$ and $R_0 = \sqrt{\Delta/(80L)}$. Further assume $K \gtrsim \log(MKR/\delta)$, $1 - \beta_1 = \Omega(1)$, $\|\boldsymbol{\sigma}\|_{2\alpha} d^{\frac{1}{2} - \frac{1}{2\alpha}} = \mathcal{O}(\sigma)$ and $1 - \beta_2 = \tilde{\mathcal{O}}(K^{-3/2}R^{-1/2})$. Then with probability no less than $1 - \delta$, Local Adam yields*

$$\frac{\lambda}{KR} \sum_{r=0}^{R-1} \sum_{k=0}^{K-1} \|\nabla f(\overline{z}_{r,k})\|_{H_r^{-1}}^2 = \tilde{\mathcal{O}}\Big(\frac{\tau\Delta}{R} + \frac{L\Delta}{KR} + \sqrt{\frac{L\Delta\sigma^2}{MKR}} + \frac{(L\Delta\sigma)^{\frac{2}{3}}}{K^{\frac{1}{3}}R^{\frac{2}{3}}} + \Big(\frac{L\Delta\sigma^{\frac{\alpha}{\alpha-1}}}{KR}\Big)^{\frac{2(\alpha-1)}{3\alpha-2}}\Big). \tag{4.9}$$

The RHS of (4.9) consists of four parts. The first part is $\frac{\tau\Delta}{R} + \frac{L\Delta}{KR}$, which is the optimization term and determined by the upper bound of learning rate $\eta$. The second term is $\sqrt{\frac{L\Delta\sigma^2}{MKR}}$, corresponding to the standard statistical lower bound from $MKR$ stochastic gradients (Arjevani et al., 2023). The third component is $\frac{(L\Delta\sigma)^{\frac{2}{3}}}{K^{\frac{1}{3}}R^{\frac{2}{3}}}$, which is sourced from the discrepancy overhead of doing local iterations. And the last one, $\Big(\frac{L\Delta\sigma^{\frac{\alpha}{\alpha-1}}}{KR}\Big)^{\frac{2(\alpha-1)}{3\alpha-2}}$, is induced by the bias of clipped stochastic gradient and can be dominated when $K^{\frac{3\alpha-4}{2}} \gtrsim \sigma^2 R/(L\Delta)$.

Our analysis method can also be applied to *Minibatch* Adam (by substituting $M, K$ with 1 and $\sigma$ with $\sigma/\sqrt{MK}$; see Remark 2), and the convergence rate is

$$\tilde{\mathcal{O}}\Big(\frac{L\Delta}{R} + \sqrt{\frac{L\Delta\sigma^2}{MKR}}\Big), \tag{4.10}$$

aligned with (up to logarithmic factor) the state-of-the-art convergence guarantees for smooth weakly convex functions (Davis & Drusvyatskiy, 2019; Deng & Gao, 2021). Suppose $K^{\frac{3\alpha-4}{2}} \gtrsim \sigma^2 R/(L\Delta)$ and hence the last term in (4.9) would be dominated and negligible. Now we can observe the benefits of local iterations. Note that both (4.9) and (4.10) have the statistical lower bound $1/\sqrt{MKR}$. Hence when the statistical term dominates, both algorithms have similar worst-case rate. Once we leave the noise-dominated regime, then *Local* Adam converges faster than *Minibatch* Adam whenever $K \gtrsim \sigma^2 R/(L\Delta)$. And the gap will increase as $K$ grows until $K \asymp L/\tau$.

Therefore, we conclude that in the large $M$ and small $\tau$ regime, *Local* Adam would outperform *Minibatch* Adam. Since $f$ is close to convex function when $\tau$ is small, this is consistent with Woodworth et al. (2020a). Please see Appendix D.5 for more comparisons about Moreau envelop.

We defer further discussions on the choices of other important hyper-parameters including $\beta_1, \beta_2, \lambda$ to Appendix D.5. The complete proof is in Appendix D.

## 5 PROOF SKETCH

In this section, we show high-level ideas in our proofs. We only demonstrate the *Local* Adam here since *Local* SGDM is a special case of *Local* Adam ($\beta_2 = 1$) and has similar patterns.

As a common practice in the study of weakly convex function (Davis & Drusvyatskiy, 2019; Mai & Johansson, 2020), the norm of the gradient of the Moreau envelope can serve as a proxy for near-stationarity. Here we use a generalized Moreau envelope for adaptive algorithms, proposed by Alacaoglu et al. (2020). For any positive definite matrix $H$ and $\gamma > 0$ such that $\gamma^{-1} H \succeq \tau I_d$, define the Moreau envelope of $f$ as

$$f_\gamma^H(x) := \min_{y \in \mathbb{R}^d} f(y) + \frac{1}{2\gamma} \|x - y\|_H^2. \tag{5.1}$$

With some abuse of notation, we define $f_\gamma^\lambda(x) := f_\gamma^{\lambda I_d}(x) = f_{\gamma/\lambda}(x)$. The common convergence metric for weakly-convex function is correspondingly $\|\nabla f_\gamma^H(\cdot)\|_{H^{-1}}$, which can bound $\|\nabla f(\cdot)\|_{H^{-1}}$, as shown in the following lemma.

**Lemma 4** (Full version see Lemma D.4). *Let $z \in \Omega_0$ and $y := \arg\min_x f(x) + \frac{1}{2\gamma} \|x - z\|_H^2$ for some $H \succeq \lambda I_d$ and $L/\lambda \geq \gamma^{-1} \geq 2\tau/\lambda$. Then*

$$\nabla f_\gamma^H(z) = \nabla f(y) = H(z - y)/\gamma, \qquad \|\nabla f(z)\|_{H^{-1}} \leq 2\gamma L \|\nabla f_\gamma^H(z)\|_{H^{-1}}/\lambda. \tag{5.2}$$

In the rest of this section, we provide the proof sketch for general Moreau envelop.

For any integer $0 \leq t \leq T-1$, we define $r(t), k(t) \in \mathbb{N}$ such that $t = r(t)K + k(t)$ and $k(t) \leq K-1$. We will omit the dependence on $t$ and let $r = r(t), k = k(t)$ if not causing confusion. Further define

$$x_t^m := x_{r,k}^m, g_t^m := g_{r,k}^m, \widehat{g_t^m} := \widehat{g_{r,k}^m}, u_t^m = u_{r,k}^m, v_t^m = v_{r,k}^m, H_t^m := \mathbf{diag}(\sqrt{v_t^m + \lambda^2}) \tag{5.3}$$

Then Algorithm 1 is equivalent to the following update rule:

$$x_{t+1}^m = \begin{cases} x_t^m - \eta(H_t^m)^{-1} u_t^m & \text{if } t \bmod K \not\equiv -1, \\ \overline{x}_t - \eta\mathbb{E}_m[(H_t^m)^{-1} u_t^m] & \text{otherwise.} \end{cases} \tag{5.4}$$

Define an auxiliary sequence $\{z_t^m\}$ as:

$$z_{t+1}^m = \begin{cases} (x_{t+1}^m - \beta_1 x_t^m)/(1 - \beta_1) & \text{if } t \bmod K \not\equiv -1, \\ (x_{t+1}^m - \beta_1 \overline{x}_t)/(1 - \beta_1) & \text{otherwise.} \end{cases} \tag{5.5}$$

Let $y_t := \arg\min_y f(y) + \frac{1}{2\gamma} \|y - \overline{z}_t\|_{H_{r(t)}}^2$. Define filtration $\mathcal{F}_{-1} = \emptyset, \mathcal{F}_t := \sigma(\{g_{r,k}^m\}_m \cup \mathcal{F}_{t-1})$ and conditional expectation $\mathbb{E}_t[\cdot] = \mathbb{E}[\cdot|\mathcal{F}_t]$.

As standard practice in distributed optimization, our proof mainly contains two parts: **contraction** and **descent**. Here contraction involves showing that the iterates of local training at different workers will not diverge to different points. And decent involves showing that the objective value decreases at each iteration. Our strategy is to inductively prove that some probabilistic event $E_t \in \mathcal{F}_{t-1}$ holds with high probability, which are designed to ensure contraction and descent. And event $E_T$ can directly imply the upper bound in Theorem 3. In fact, event $E_t$ has the form of

$$E_t = \{\mathcal{A}_{j,i} \text{ holds for all } j \leq t - 1, i \in \{1, 2, 3, 4\}\}, \tag{5.6}$$

where $\mathcal{A}_{j,i} \in \mathcal{F}_j$ (defined later) is also some probabilistic event. As the components of $E_t$, each $\mathcal{A}_{j,i}$ is designed to ensure either contraction or descent. We will prove the high probability bound of these components in sequence.

## 5.1 BOUNDING THE TRAJECTORY WITH HIGH PROBABILITY

Similar to Sadiev et al. (2023), we only make assumptions on $f$ and noise in certain subset $\Omega \subset \mathbb{R}^d$. However, we are able to show that all the iterates will not leave $\Omega$ with high probability. Specifically, if it holds for all iterates before time $t$, using standard techniques for weakly convex optimization, we can upper bound the function value and Moreau envelope at $\overline{z}_{t+1}$ by

$$f_\gamma^{H_{r(t+1)}}(\overline{z}_{t+1}) \leq f_\gamma^\lambda(x_0) - \Omega(\eta) \sum_{j=0}^t \|\nabla f_\gamma^{H_{r(j)}}(\overline{z}_j)\|^2_{H_{r(j)}^{-1}} + \mathcal{O}(\eta^2) \underbrace{\sum_{j=0}^t \|\mathbb{E}_m[\nabla f(x_j^m) - \widehat{g_j^m}]\|^2}_{\text{stochastic noise}}$$

$$+ \mathcal{O}(\eta) \underbrace{\sum_{j=0}^t \|\nabla f(\overline{z}_j) - \mathbb{E}_m[\nabla f(x_j^m)]\|^2}_{\text{discrepancy}}$$

$$+ \mathcal{O}(\eta) \underbrace{\sum_{j=0}^t \left\langle \overline{z}_j - \eta H_{r(j)}^{-1}\nabla f(\overline{z}_j) - y_j, \mathbb{E}_m[\mathbb{E}_j[\widehat{g_j^m}] - \widehat{g_j^m}] \right\rangle}_{\text{martingale}}$$

$$+ \text{ higher order terms.}$$

$$(5.7)$$

To see that the last term is a martingale, note that $H_{r(j)}$ is independent of $\widehat{g_j^m}$ since the stochastic gradient $\widehat{g_j^m}$ is drawn during round $r$. Further note that $\mathbb{E}_j[\widehat{g_j^m}] - \widehat{g_j^m}$ is almost surely bounded thanks to clipping. Now (5.7) allows us to inductively bound $f_\gamma^{H_{r(j)}}(\overline{z}_j)$ and thus bound $\|\overline{z}_j - \eta H_{r(j)}^{-1}\nabla f(\overline{z}_j) - y_j\|$. After these preliminaries, we are able to apply Berstein's inequality (Bennett, 1962; Freedman, 1975) to control this martingale. Hence the Moreau envelope at $\overline{z}_{t+1}$ can be bounded by a constant with high probability. Combining this with contraction results below, we can show that all the iterates stay in $\Omega$ with high probability.

## 5.2 CONTRACTION

Next, we aim to show contraction, *i.e.*, $\|x_t^m - x_t^n\|$ will not diverge during local iterations with high probability. This property is crucial for showing the benefits of local updates in distributed optimization. However, different from Woodworth et al. (2020a); Khaled et al. (2020), the update of $x_t^m$ in Algorithm 1 is in the direction of $(H_t^m)^{-1}u_t^m$, which distorts the gradient by both exponential moving average (EMA) and coordinate-wise product. Thus, the weak monotonicity (4.6) can not be directly applied as in standard analysis of gradient descent. This will further impede contraction.

Our solution has two steps. Firstly, we try to diminish the negative effects of different denominators used in local iterations. Then we turn to deal with the EMA of past gradient in first order momentum.

**Lemma 5** (Informal). *Define probabilistic events*

$$\mathcal{A}_{t,1} := \left\{ \beta_2^{K/2} \preceq H_{r(t)}^{-1}H_t^m \preceq 1 + (1-\beta_2)B \text{ and for all } m \in [M] \right\}, \tag{5.8}$$

$$\mathcal{A}_{t,2} := \left\{ \|H_{r(t)}((H_t^m)^{-1} - (H_t^n)^{-1})\| \leq (1-\beta_2)B_1 \text{ for all } m,n \in [M] \right\}, \tag{5.9}$$

*where $B, B_1$ are some constants. Define $E_{t,1} := E_t \cap \mathcal{A}_{t,1}, E_{t,2} := E_{t,1} \cap \mathcal{A}_{t,2}$. For $B = \tilde{\mathcal{O}}(K), B_1 = \tilde{\mathcal{O}}(K)$, it holds that $\mathbb{P}(E_{t,1}) \geq \mathbb{P}(E_t) - \delta/(4T), \quad \mathbb{P}(E_{t,2}) \geq \mathbb{P}(E_{t,1}) - \delta/(4T)$.*

Event $\mathcal{A}_{t,1}$ implies the denominator of each worker during local iterations tends to be stagnant and close to the averaged one after communication. Event $\mathcal{A}_{t,2}$ suggests the denominator at each worker is close to each other. The key idea is to control the magnitude of $v_t^m = (1-\beta_2)\sum_{j=r(t)K}^t \beta_2^{t-j}\widehat{g_j^m}^2 + \beta_2^{k(t)+1}v_{r(t)}$. Since all the iterates stay in $\mathbf{conv}(\mathbf{B}_{R_0}(\Omega_0))$, the squared gradient $\nabla f(x_j^m)^2$ can be bounded. Besides, we can handle the martingale induced by $\widehat{g_j^m}^2 - \mathbb{E}_j[\widehat{g_j^m}^2]$ by Berstein's inequality. The remaining term $\mathbb{E}_j[\widehat{g_j^m}^2] - \nabla f(x_j^m)^2$ is controlled by the property of clipping operator.

Now that the denominator is relatively stagnant, the update of $x_t^m$ is approximately preconditioned by $H_{r(t)}$ for all $m$. Hence we can turn to handle the first order momentum. A vanilla idea is to do the following expansion:

$$\|x_{t+1}^m - x_{t+1}^n\|_{H_r}^2 \approx \|x_t^m - x_t^n\|_{H_r}^2 - 2\eta \langle x_t^m - x_t^n, u_t^m - u_t^n \rangle + \mathcal{O}(\eta^2). \quad (5.10)$$

By the definition of $u_t^m$, however, it would be influenced by noises from past stochastic gradients. In this way, $u_t^m - u_t^n$ is not independent of $x_t^m - x_t^n$ and thus it is difficult to construct a martingale and apply Berstein's inequality. This is the reason why we introduce the auxiliary sequence $\{z_t^m\}$ defined in (5.5). Fortunately, noticing that $x_t^m - x_t^n \in \mathbf{conv}(\{z_j^m - z_j^n\}_{j \le t})$, it suffices to show that $\|z_t^m - z_t^n\|$ will not get too large with high probability.

**Lemma 6** (Informal). *Define probabilistic event*

$$\mathcal{A}_{t,3} := \left\{ \|z_{t+1}^m - z_{t+1}^n\|_{H_r}^2 \le \frac{\eta^2 \sigma^2}{\lambda} KA, \sum_{j=rK}^{t} \|\widehat{g_j^m}\|^2 \le \frac{(1-\beta_1)^2 \sigma^2 A}{2^{12}(1-\beta_2)^2 B_1^2} \text{ for all } m, n \in [M] \right\},$$

$$(5.11)$$

*where $A$ is some constant. Define $E_{t,3} := E_{t,2} \cap \mathcal{A}_{t,3}$. For $A = \tilde{\mathcal{O}}(1)$ and $\eta = \tilde{\mathcal{O}}\left( \min\{1/(K\tau), 1/L\} \right)$, it holds that $\mathbb{P}(E_{t,3}) \ge \mathbb{P}(E_{t,2}) - \delta/(4T)$.*

Event $\mathcal{A}_{t,3}$ is the desired contraction property and can further imply that $\|x_{t+1}^m - x_{t+1}^n\|_{H_r}^2 \le \eta^2 \sigma^2 KA/\lambda$ when combined with event $E_t$. In fact, for $\{z_t^m\}$, we can do the following expansion:

$$\|z_{t+1}^m - z_{t+1}^n\|_{H_r}^2 \approx \|z_t^m - z_t^n\|_{H_r}^2 - 2\eta \langle z_t^m - z_t^n, \widehat{g_t^m} - \widehat{g_t^n} \rangle + \mathcal{O}(\eta^2). \quad (5.12)$$

Informally speaking, $\mathbb{E}_t[\widehat{g_t^m} - \widehat{g_t^n}]$ is roughly $\nabla f(x_t^m) - \nabla f(x_t^n)$, which is close to $\nabla f(z_t^m) - \nabla f(z_t^n)$ since $\|z_t^m - x_t^m\|^2 = \mathcal{O}(\|x_t^m - x_{t-1}^m\|^2) = \mathcal{O}(\eta^2)$. In this way, the middle term $\mathcal{O}(\eta)$ of RHS above can be turned to $-2\eta \langle z_t^m - z_t^n, \nabla f(z_t^m) - \nabla f(z_t^n) \rangle$, where the weak convexity can be applied. Finally we control the martingale induced by $\langle z_t^m - z_t^n, \widehat{g_t^m} - \widehat{g_t^n} - \mathbb{E}_t[\widehat{g_t^m} - \widehat{g_t^n}] \rangle$ through Bersteins's inequality.

## 5.3 DESCENT

Finally, we are ready to prove the descent lemma, which is the last component of $E_{t+1}$. Define

$$\mathcal{A}_{t,4} := \left\{ f_\gamma^{H_{r(t+1)}}(\bar{z}_{t+1}) - f_* + \frac{\eta}{12} \sum_{j=0}^{t} \|\nabla f_\gamma^{H_{r(j)}}(\bar{z}_j)\|_{H_{r(j)}^{-1}}^2 \le 2\Delta \right\}. \quad (5.13)$$

We proceed with (5.7) and control the stochastic noise term by subtracting its expectation to construct a martingale. As for the discrepancy overhead, we apply the upper bound of $\|x_j^m - x_j^n\|^2$, which is induced by the event $E_t$ and utilize the $\mathcal{O}(\eta^2)$ bound on $\|\bar{z}_j - \bar{x}_j\|^2$. Therefore, thanks to all the foundations above, we are able to bound each of these terms.

**Lemma 7** (Informal). *For sufficiently small $\eta$, it holds that $\mathbb{P}(E_{t+1}) \ge \mathbb{P}(E_{t,3}) - \delta/(4T)$.*

Therefore, we prove that $\mathbb{P}(E_{t+1}) \ge \mathbb{P}(E_t) - \delta/T$. And by induction rule, $\mathbb{P}(E_T) \ge 1 - \delta$. After carefully choosing the learning rate $\eta$, we complete the proof of Theorem 3.

## 6 CONCLUSION

In this paper, we prove the benefits of local updates within distributed adaptive methods to reduce communication complexity compared to their minibatch counterparts. We study *Local* SGDM and *Local* Adam under convex and weakly convex setting, respectively. We consider generalized smoothness assumption and gradient clipping, and develop a novel technique to show contraction during local updates. Future works may include improved analysis of *Local* Adam, benefits of local adaptive algorithms in non-convex setting, advantages over non-adaptive methods, etc.

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

## A  ADDITIONAL RELATED WORK

**Gradient clipping.** Pascanu et al. (2013) first proposed gradient clipping technique to address the issue of exploding gradient problem of deep neural networks. Since then, it has become standard practice in the training of language models (Gehring et al., 2017; Merity et al., 2017; Zhang et al., 2022a; Liu et al., 2023). Furthermore, from theoretical perspective, gradient clipping is also used for multiple purposes, including differential privacy (Abadi et al., 2016), distributed optimization (Karimireddy et al., 2021; Liu et al., 2022), heavy-tailed noise (Zhang et al., 2020).

**Generalized smoothness.** The generalized smoothness condition was initially proposed by (Zhang et al., 2019) to justify gradient clipping, and was called $(L_0, L_1)$-smoothness. The empirical evidence therein illustrated that the norm of Hessian matrix of language models depends linearly on the magnitude of gradient, contradicting the standard $L$-smoothness. A recent work (Li et al., 2024b) further generalized this condition to $\ell$-smoothness and proved convergence of classical SGD in this setting. Apart from bounding the Hessian through gradient, Sadiev et al. (2023) proposed to assume that the norm of Hessian is uniformly bounded in certain subset of whole space, in order to get high probability bounds for (accelerated) clipped-SGD. Gorbunov et al. (2023) further extended this setting to composite and distributed optimization without local updates. Here we follow the setting of (Sadiev et al., 2023) since $(L_0, L_1)$-smoothness would reduce to it in most cases. See Section 3.1 for details.

## B  TECHNICAL LEMMAS

**Lemma B.1** ((Bennett, 1962; Freedman, 1975)). *Let the sequence of random variables $\{X_i\}_{i \geq 1}$ form a martingale difference sequence, i.e. $\mathbb{E}[X_i | X_{i-1}, \cdots, X_1] = 0$ for all $i \geq 1$. Assume that conditional variances $\sigma_i^2 \stackrel{def}{=} \mathbb{E}[X_i^2 | X_{i-1}, \cdots, X_1]$ exist and are bounded and assume also that there exists deterministic constant $c > 0$ such that $|X_i| \leq c$ almost surely for all $i \geq 1$. Then for all $b > 0, V > 0$ and $n \geq 1$,*

$$\mathbb{P}\left\{ |\sum_{i=1}^{n} X_i| > b \text{ and } \sum_{i=1}^{n} \sigma_i^2 \leq V \right\} \leq 2 \exp\left( -\frac{b^2}{2V + 2cb/3} \right). \tag{B.1}$$

**Lemma B.2.** *Let $X$ be a random variable in $\mathbb{R}$ and $\tilde{X} := \mathbf{clip}(X, \rho)$, Then $\|\tilde{X} - \mathbb{E}\tilde{X}\| \leq 2\rho$. Moreover, if for some $\sigma > 0$ and $\alpha \geq 2$,*

$$\mathbb{E}[X] = x \in \mathbb{R}, \qquad \mathbb{E}|X - x|^\alpha \leq \sigma^\alpha, \tag{B.2}$$

*and $|x| \leq \frac{\rho}{2}$, $\rho \geq 3\sigma$, then*

$$|\mathbb{E}[\tilde{X}] - x| \leq \frac{(2\sigma)^\alpha}{\rho^{\alpha-1}}, \qquad \mathbb{E}|\tilde{X} - x|^\alpha \leq \sigma^\alpha, \qquad \mathbb{E}|\tilde{X} - \mathbb{E}[\tilde{X}]|^\alpha \leq (2\sigma)^\alpha. \tag{B.3}$$

*Proof.* The first claim is from (Sadiev et al., 2023) and we show the proof here for completeness. To start the proof, we introduce two indicator random variables. Let

$$\chi = \mathbb{I}_{\{X : |X| > \rho\}} = \begin{cases} 1, & \text{if } |X| > \rho, \\ 0, & \text{otherwise} \end{cases}, \quad \eta = \mathbb{I}_{\{X : |X - x| > \frac{\rho}{2}\}} = \begin{cases} 1, & \text{if } |X - x| > \frac{\rho}{2}, \\ 0, & \text{otherwise} \end{cases}. \tag{B.4}$$

Moreover, since $|X| \leq |x| + |X - x| \leq \frac{\rho}{2} + |X - x|$, we have $\chi \leq \eta$. Using that

$$\tilde{X} = \min\left\{ 1, \frac{\rho}{|X|} \right\} X = \chi \frac{\rho}{|X|} X + (1 - \chi) X, \tag{B.5}$$

we obtain

$$
\begin{aligned}
|\mathbb{E}[\tilde{X}] - x| &= \left| \mathbb{E}[X + \chi\left( \frac{\rho}{|X|} - 1 \right) X] - x \right| \\
&= \left| \mathbb{E}\left[ \chi\left( \frac{\rho}{|X|} - 1 \right) X \right] \right| \\
&= \mathbb{E}\left[ \chi\left( 1 - \frac{\rho}{|X|} \right) |X| \right].
\end{aligned}
\tag{B.6}
$$

Since $1 - \dfrac{\rho}{|X|} \in (0,1)$ when $\chi \neq 0$, we derive

$$
\begin{aligned}
|\mathbb{E}[\tilde{X}] - x| &\leq \mathbb{E}\left[\chi|X|\right] \\
&\leq \mathbb{E}\left[\eta|X|\right] \\
&\leq \mathbb{E}\left[\eta|X - x| + \eta|x|\right] \\
&\leq \left(\mathbb{E}\left[|X - x|^{\alpha}\right]\right)^{\frac{1}{\alpha}} \left(\mathbb{E}\left[\eta^{\frac{\alpha}{\alpha-1}}\right]\right)^{\frac{\alpha-1}{\alpha}} + |x|\mathbb{E}\left[\eta\right] \\
&\stackrel{\eta \in \{0,1\}}{\leq} \sigma \left(\mathbb{E}\left[\eta\right]\right)^{\frac{\alpha-1}{\alpha}} + \frac{\rho}{2}\mathbb{E}\left[\eta\right],
\end{aligned}
\tag{B.7}
$$

By Markov's inequality,

$$
\begin{aligned}
\mathbb{E}\left[\eta\right] &= \mathbb{P}\left\{|X - x|^{\alpha} > \frac{\rho^{\alpha}}{2^{\alpha}}\right\} \\
&\leq \frac{2^{\alpha}}{\rho^{\alpha}}\mathbb{E}\left[|X - x|^{\alpha}\right] \\
&\leq \left(\frac{2\sigma}{\rho}\right)^{\alpha}.
\end{aligned}
\tag{B.8}
$$

Thus, in combination with the previous chain of inequalities, we finally have

$$
|\mathbb{E}[\tilde{X}] - x| \leq \sigma\left(\frac{2\sigma}{\rho}\right)^{\alpha-1} + \frac{\rho}{2}\left(\frac{2\sigma}{\rho}\right)^{\alpha} = \frac{2^{\alpha}\sigma^{\alpha}}{\rho^{\alpha-1}}.
\tag{B.9}
$$

For the second part, since

$$
|\tilde{X} - x| = |\mathbf{clip}(X, \rho) - \mathbf{clip}(x, \rho)| \leq |X - x|,
\tag{B.10}
$$

hence $\mathbb{E}|\tilde{X} - x|^{\alpha} \leq \mathbb{E}|X - x|^{\alpha} \leq \sigma^{\alpha}$. By Jensen's inequality, we have for any $q \in (0,1)$,

$$
\begin{aligned}
\mathbb{E}|\tilde{X} - \mathbb{E}[\tilde{X}]|^{\alpha} &\leq q^{1-\alpha}\mathbb{E}|\tilde{X} - x|^{\alpha} + (1 - q)^{1-\alpha}|\mathbb{E}[\tilde{X}] - x|^{\alpha} \\
&\leq q^{1-\alpha}\sigma^{\alpha} + (1 - q)^{1-\alpha}\left(\frac{(2\sigma)^{\alpha}}{\rho^{\alpha-1}}\right)^{\alpha}.
\end{aligned}
\tag{B.11}
$$

Choose the optimal $q = \dfrac{\sigma}{\sigma + \frac{(2\sigma)^{\alpha}}{\rho^{\alpha-1}}}$ and we can conclude that

$$
\mathbb{E}|\tilde{X} - \mathbb{E}[\tilde{X}]|^{\alpha} \leq \left(\sigma + \frac{(2\sigma)^{\alpha}}{\rho^{\alpha-1}}\right)^{\alpha} \leq (2\sigma)^{\alpha}.
\tag{B.12}
$$

This completes the proof. $\qquad\square$

**Lemma B.3.** *For $M$ independent random vectors $X_1, \cdots, X_M \in \mathbb{R}^d$ such that $\mathbb{E}[X_m] = 0$, $\mathbb{E}[\|X_m\|^4] \leq \sigma^4$, the following holds*

$$
\mathbb{E}\left[\|\mathbb{E}_m X_m\|^2\right]^2 \leq \frac{4\sigma^4}{M^2}.
\tag{B.13}
$$

*Proof.* We prove by direct calculation as follows:

$$
\begin{aligned}
\mathbb{E}\left[\|\mathbb{E}_m X_m\|^2\right]^2 &\leq \mathbb{E}\left[\frac{1}{M^2}\sum_m \|X_m\|^2 + \frac{2}{M^2}\sum_{m<n}\langle X_m, X_n\rangle\right]^2 \\
&= \mathbb{E}\left[\frac{1}{M^2}\sum_m \|X_m\|^2\right]^2 + \mathbb{E}\left[\frac{2}{M^2}\sum_{m<n}\langle X_m, X_n\rangle\right]^2 \\
&\leq \frac{\sigma^4}{M^2} + \frac{4}{M^4}\mathbb{E}\sum_{m<n}\langle X_m, X_n\rangle^2 \\
&\leq \frac{4\sigma^4}{M^2}.
\end{aligned}
\tag{B.14}
$$

$\qquad\square$

**Lemma B.4.** *For any set* $\Omega \in \mathbb{R}^d$ *and* $r > 0$, *define* $\mathbf{B}_r(\Omega) := \left\{ x \in \mathbb{R}^d : \exists y \in \Omega, s.t., \|x - y\| \leq r \right\}$. *Then*

$$\mathbf{B}_r(\mathbf{conv}(\Omega)) = \mathbf{conv}(\mathbf{B}_r(\Omega)). \tag{B.15}$$

*Proof.* For any $x \in \mathbf{B}_r(\mathbf{conv}(\Omega))$, there exist $y_1, \cdots, y_N \in \Omega$ and $(\lambda_1, \cdots, \lambda_N) \in \Delta^N$ for some $N$, such that

$$\|x - y\| \leq r, \ y := \sum_{n=1}^{N} \lambda_n y_n. \tag{B.16}$$

Then $x = y + (x - y) = \sum_{n=1}^{N} \lambda_n(y_n + x - y) = \sum_{n=1}^{N} \lambda_n x_n$, where

$$x_n = y_n + x - y \in B_r(\Omega). \tag{B.17}$$

Hence $x \in \mathbf{conv}(\mathbf{B}_r(\Omega))$.

On the other hand, for any $x \in \mathbf{conv}(\mathbf{B}_r(\Omega))$, there exist $x_1, \cdots, x_N \in \mathbf{B}_r(\Omega), y_1, \cdots, y_N \in \Omega$ and $(\lambda_1, \cdots, \lambda_N) \in \Delta^N$, such that

$$x = \sum_{n=1}^{N} \lambda_n x_n, \|x_n - y_n\| \leq r. \tag{B.18}$$

Let $y := \sum_{n=1}^{N} \lambda_n y_n \in \mathbf{conv}(\Omega)$. Then $\|x - y\| \leq \sum_{n=1}^{N} \lambda_n \|x_n - y_n\| \leq r$ and thus $x \in \mathbf{B}_r(\mathbf{conv}(\Omega))$. $\square$

## C    PROOF OF LOCAL SGDM

We restate the *Local* SGDM algorithm here.

---
**Algorithm 2** *Local* SGDM
---
**Require:** initial model $x_0$, learning rate $\eta$, momentum $\beta_1 \in [0, 1)$
    Set $x_{0,0}^m = x_0$, $u_{0,-1}^m = 0$ for each worker $m \in [M]$
    **for** $r = 0, \cdots, R - 1$ **do**
        **for** each worker $m \in [M]$ in parallel **do**
            **for** $k = 0, \cdots, K - 1$ **do**
                $g_{r,k}^m = \nabla F(x_{r,k}^m; \xi_{r,k}^m), \ \widehat{g_{r,k}^m} = \mathbf{clip}(g_{r,k}^m, \rho)$         ▷ Compute clipped stochastic gradient
                $u_{r,k}^m = \beta_1 u_{r,k-1}^m + (1 - \beta_1)\widehat{g_{r,k}^m}$                 ▷ Update momentum
                $x_{r,k+1}^m = x_{r,k}^m - \eta u_{r,k}^m$                       ▷ Update model
            **end for**
        **end for**
        $x_{r+1,0}^m = \mathbb{E}_m[x_{r,K}^m], \ u_{r+1,-1}^m = \mathbb{E}_m[u_{r,K-1}^m]$         ▷ Communicate and average
    **end for**
---

### C.1    OVERVIEW AND MAIN THEOREM

For any integer $0 \leq t \leq T-1$, we define $r(t), k(t) \in \mathbb{N}$ such that $t = r(t)K + k(t)$ and $k(t) \leq K - 1$. We omit the dependence on $t$ and let $r = r(t), k = k(t)$ through out the proof if not causing confusion. Define $x_t^m := x_{r,k}^m, g_t^m := g_{r,k}^m, \widehat{g_t^m} := \widehat{g_{r,k}^m}, u_t^m = u_{r,k}^m$. Then Algorithm 2 is equivalent to the following update rule:

$$u_t^m = \begin{cases} \beta_1 u_{t-1}^m + (1 - \beta_1)\widehat{g_t^m} & \text{if } t \bmod K \not\equiv 0, \\ \beta_1 \overline{u}_{t-1} + (1 - \beta_1)\widehat{g_t^m} & \text{otherwise}, \end{cases} \tag{C.1}$$

$$x_{t+1}^m = \begin{cases} x_t^m - \eta u_t^m & \text{if } t \bmod K \not\equiv -1, \\ \overline{x}_t - \eta \overline{u}_t & \text{otherwise.} \end{cases} \tag{C.2}$$

Define an auxiliary sequence $\{z_t^m\}$ as:

$$z_{t+1}^m = \begin{cases} \dfrac{1}{1-\beta_1} x_{t+1}^m - \dfrac{\beta_1}{1-\beta_1} x_t^m & \text{if } t \bmod K \not\equiv -1, \\ \dfrac{1}{1-\beta_1} x_{t+1}^m - \dfrac{\beta_1}{1-\beta_1} \overline{x}_t & \text{otherwise.} \end{cases} \tag{C.3}$$

Define probabilistic events (see (C.12) for definition of some parameters)

$$\mathcal{A}_{t,1} := \left\{ \|z_{t+1}^m - z_{t+1}^n\|^2 \le \eta^2 \sigma^2 K A \text{ for all } m, n \in [M] \right\}, \tag{C.4}$$

$$\mathcal{A}_{t,2} := \left\{ \sum_{j=0}^{t} \frac{\eta}{2} (f(\overline{z}_j) - f_*)(1 - \frac{\eta\mu}{2})^{t-j} + \|\overline{z}_{t+1} - x_*\|^2 \le 2(1 - \frac{\eta\mu}{2})^{t+1} D_0^2 \right\}. \tag{C.5}$$

Besides, let

$$E_t := \{\mathcal{A}_{j,i} \text{ holds for all } j \le t-1, i \in \{1,2\}\}, \ E_{t,1} := E_t \cap \mathcal{A}_{t,1}. \tag{C.6}$$

Now we present two of our major lemmas, the first of which is to show contraction and the second is a descent lemma.

**Lemma C.1.** *Let* $A := \max\left\{ \dfrac{2^{10}\rho^2 d}{K\sigma^2} \log^2 \dfrac{MT}{\delta}, 2^9 \log \dfrac{MT}{\delta}, 2^{12} \dfrac{K\|2\boldsymbol{\sigma}\|_{2\alpha}^{2\alpha}}{\sigma^2 \rho^{2(\alpha-1)}} \right\}.$ *If* $\eta \le$ $\min\left\{ \dfrac{(1-\beta_1)^2}{2L}, \dfrac{D_0}{4\sigma\sqrt{KA}} \right\}$ *and* $\rho \ge \max\{3\sigma_\infty, 2G_\infty\}$, *then the following holds:*

$$\mathbb{P}(E_{t,1}) \ge \mathbb{P}(E_t) - \frac{\delta}{2T}. \tag{C.7}$$

**Lemma C.2.** *For any* $\varepsilon > 0$, *let*

$$\rho \ge \begin{cases} \max\left\{ \left( \dfrac{2^8\|2\boldsymbol{\sigma}\|_{2\alpha}^{2\alpha}}{\mu\varepsilon} \right)^{\frac{1}{2(\alpha-1)}}, 3\sigma_\infty, 2G_\infty \right\}, & \text{if } \mu > 0, \\ \max\left\{ \left( \dfrac{2^8 D_0 \|2\boldsymbol{\sigma}\|_{2\alpha}^{\alpha}}{\varepsilon} \right)^{\frac{1}{\alpha-1}}, 3\sigma_\infty, 2G_\infty \right\}, & \text{otherwise.} \end{cases} \tag{C.8}$$

$$\eta := \begin{cases} \dfrac{2}{\mu T} \log \dfrac{4\mu D_0^2}{\varepsilon}, & \text{if } \mu > 0, \\ \dfrac{4 D_0^2}{T\varepsilon}, & \text{otherwise.} \end{cases}$$

*If*

$$\eta \lesssim \begin{cases} \min\left\{ \dfrac{(1-\beta_1)^2}{L}, \dfrac{M\varepsilon}{\sigma^2 \log \frac{T}{\delta}}, \left( \dfrac{L\sigma^2 KA}{\varepsilon} \right)^{-1/2}, \dfrac{\sqrt{\varepsilon/\mu}}{\rho\sqrt{d} \log \frac{T}{\delta}} \right\}, & \text{if } \mu > 0, \\ \min\left\{ \dfrac{(1-\beta_1)^2}{L}, \dfrac{M\varepsilon}{\sigma^2 \log \frac{T}{\delta}}, \left( \dfrac{L\sigma^2 KA}{\varepsilon} \right)^{-1/2}, \dfrac{D_0}{\rho\sqrt{d} \log \frac{T}{\delta}} \right\}, & \text{otherwise,} \end{cases} \tag{C.9}$$

*where $A$ is defined in Lemma C.1, then the following holds*

$$\mathbb{P}(E_{t+1}) \ge \mathbb{P}(E_{t,1}) - \frac{\delta}{2T}. \tag{C.10}$$

The following is our main result, from which we will parse the implications in Theorems 1 and 2.

**Theorem C.3.** *Let Assumption 1, 2, 3, 4 hold for* $\Omega := \{\|x - x_*\| \le \sqrt{3}D_0\}$. *Further assume that for any* $x \in \Omega$, $\|\nabla f(x)\|_\infty \le G_\infty$. *Then with probability* $\ge 1-\delta$, *Local SGDM yields* $f(\hat{x}) - f_* \le \varepsilon$

*if*

$$T \gtrsim \begin{cases} \log \dfrac{\mu D_0^2}{\varepsilon} \left[ \dfrac{L}{(1-\beta_1)^2 \mu} + \dfrac{\sigma^2}{\mu M \varepsilon} \log \dfrac{T}{\delta} + \sqrt{\dfrac{L\sigma^2 KA}{\mu^2 \varepsilon}} + \dfrac{\rho \sqrt{d}}{\sqrt{\mu \varepsilon}} \log \dfrac{T}{\delta} \right], & \text{if } \mu > 0, \\[4mm] \dfrac{D_0^2}{\varepsilon} \left[ \dfrac{L}{(1-\beta_1)^2} + \dfrac{\sigma^2}{M \varepsilon} \log \dfrac{T}{\delta} + \sqrt{\dfrac{L\sigma^2 KA}{\varepsilon}} + \dfrac{\rho \sqrt{d}}{D_0} \log \dfrac{T}{\delta} \right], & \text{otherwise.} \end{cases}$$

(C.11)

*Here*

$$\rho \geq \begin{cases} \max \left\{ \left( \dfrac{2^8 \|2\boldsymbol{\sigma}\|_{2\alpha}^{2\alpha}}{\mu \varepsilon} \right)^{\frac{1}{2(\alpha-1)}}, 3\sigma_\infty, 2G_\infty \right\}, & \text{if } \mu > 0, \\[4mm] \max \left\{ \left( \dfrac{2^8 D_0 \|2\boldsymbol{\sigma}\|_{2\alpha}^{\alpha}}{\varepsilon} \right)^{\frac{1}{\alpha-1}}, 3\sigma_\infty, 2G_\infty \right\}, & \text{otherwise,} \end{cases}$$

$$A := \max \left\{ \frac{2^{10} \rho^2 d}{K\sigma^2} \log^2 \frac{MT}{\delta}, 2^9 \log \frac{MT}{\delta}, 2^{12} \frac{K\|2\boldsymbol{\sigma}\|_{2\alpha}^{2\alpha}}{\sigma^2 \rho^{2(\alpha-1)}} \right\},$$ (C.12)

$$\eta := \begin{cases} \dfrac{2}{\mu T} \log \dfrac{4\mu D_0^2}{\varepsilon}, & \text{if } \mu > 0, \\[4mm] \dfrac{4D_0^2}{T\varepsilon}, & \text{otherwise.} \end{cases}$$

*Proof.* We prove by induction that $\mathbb{P}(E_t) \geq 1 - \dfrac{t\delta}{T}$ for $t = 0, \cdots, T$.

When $t = 0$, this is trivial. Assume that the statement is true for some $t \leq T - 1$. We aim to prove that $\mathbb{P}(E_{t+1}) \geq 1 - \dfrac{(t+1)\delta}{T}$. It is easy to verify the conditions in Lemma C.1, C.2 once (C.11) and (C.12) hold. Hence we have

$$\mathbb{P}(E_{t+1}) \geq \mathbb{P}(E_t) - 2 \cdot \frac{\delta}{2T} \geq 1 - \frac{(t+1)\delta}{T}. \tag{C.13}$$

Therefore by induction rule, $\mathbb{P}(E_T) \geq 1 - \delta$ and this implies by event $\mathcal{A}_{T,2}$ that

$$\sum_{j=0}^{T-1} \frac{\eta}{2} (f(\bar{z}_j) - f_*) \left( 1 - \frac{\eta\mu}{2} \right)^{T-j} \leq 2 \left( 1 - \frac{\eta\mu}{2} \right)^T D_0^2. \tag{C.14}$$

Let $\hat{x} := \dfrac{\eta\mu \sum_{j=0}^{T-1} (1 - \frac{\eta\mu}{2})^{T-j} \bar{z}_j}{2(1 - (1 - \frac{\eta\mu}{2})^T)}$. By convexity, we have

$$f(\hat{x}) - f_* \leq \frac{2(1 - \frac{\eta\mu}{2})^T \mu D_0^2}{1 - (1 - \frac{\eta\mu}{2})^T}. \tag{C.15}$$

(1) **Case** $\mu > 0$.

$$f(\hat{x}) - f_* \leq \frac{2(1 - \frac{\eta\mu}{2})^T \mu D_0^2}{1 - (1 - \frac{\eta\mu}{2})^T} \leq 4(1 - \frac{\eta\mu}{2})^T \mu D_0^2 \leq 4e^{-\eta\mu T/2} \mu D_0^2 = \varepsilon. \tag{C.16}$$

(2) **Case** $\mu = 0$.

$$f(\hat{x}) - f_* \leq \frac{2(1 - \frac{\eta\mu}{2})^T \mu D_0^2}{1 - (1 - \frac{\eta\mu}{2})^T} = \frac{4D_0^2}{\eta T} = \varepsilon. \tag{C.17}$$

$\square$

We now state and prove the implications of Theorem C.3 which yield the results stated in the main body of our paper.

**Theorem C.4** (Complete version of Theorem 1). *Under the conditions of Theorem C.3 and $\mu > 0$, assume $1 - \beta_1 = \Omega(1)$, $\left(\frac{\|\boldsymbol{\sigma}\|_{2\alpha}^{2\alpha}}{\mu\varepsilon}\right)^{\frac{1}{2(\alpha-1)}} \gtrsim G_\infty \vee \sigma_\infty$, and $K \gtrsim \log \frac{MT}{\delta} \left(\frac{\|\boldsymbol{\sigma}\|_{2\alpha} d^{\frac{1}{2} - \frac{1}{2\alpha}}}{\sigma}\right)^{\frac{2\alpha}{\alpha-2}}$. Then with probability no less than $1 - \delta$, Local SGDM with optimal $\eta, \rho$ yields $f(\hat{x}) - f_* \leq \varepsilon$, if*

$$T \gtrsim \log \frac{\mu D_0^2}{\varepsilon} \left[\frac{L}{\mu} + \frac{\sigma^2}{\mu M \varepsilon} \log \frac{T}{\delta} + \sqrt{\frac{L\sigma^2 K \log \frac{MT}{\delta}}{\mu^2 \varepsilon}} + \sqrt{\frac{Ld}{\mu^2 \varepsilon}} \log \frac{MT}{\delta} \left(\frac{\|\boldsymbol{\sigma}\|_{2\alpha}^{2\alpha}}{\mu\varepsilon}\right)^{\frac{1}{2(\alpha-1)}}\right]. \tag{C.18}$$

*And equivalently, let $\kappa := L/\mu$,*

$$f(\hat{x}) - f_* \lesssim \exp\left(-\Theta\left(\frac{\mu K R}{L}\right)\right) + \frac{\sigma^2 \log(MKR)}{\mu M K R} \log \frac{KR}{\delta}$$

$$+ \frac{L\sigma^2 \log^2(KR)}{\mu^2 KR^2} \log \frac{MKR}{\delta} + \frac{\|\boldsymbol{\sigma}\|_{2\alpha}^2 (\kappa d)^{\frac{\alpha-1}{\alpha}}}{\mu} \left(\frac{\log \frac{MKR}{\delta}}{KR}\right)^{\frac{2(\alpha-1)}{\alpha}}. \tag{C.19}$$

*Proof.* Plug the definition of $A$ in (C.11),

$$T \gtrsim \log \frac{\mu D_0^2}{\varepsilon} \left[\frac{L}{\mu} + \frac{\sigma^2}{\mu M \varepsilon} \log \frac{T}{\delta} + \sqrt{\frac{L\sigma^2 K \log \frac{MT}{\delta}}{\mu^2 \varepsilon}} + \frac{\rho\sqrt{d}}{\sqrt{\mu\varepsilon}} \log \frac{T}{\delta}\right]$$

$$+ \log \frac{\mu D_0^2}{\varepsilon} \sqrt{\frac{LK}{\mu^2 \varepsilon}} \sqrt{\frac{\rho^2 d}{K} \log^2 \frac{MT}{\delta} + \frac{K\|2\boldsymbol{\sigma}\|_{2\alpha}^{2\alpha}}{\rho^{2(\alpha-1)}}}$$

$$\asymp \log \frac{\mu D_0^2}{\varepsilon} \left[\frac{L}{\mu} + \frac{\sigma^2}{\mu M \varepsilon} \log \frac{T}{\delta} + \sqrt{\frac{L\sigma^2 K \log \frac{MT}{\delta}}{\mu^2 \varepsilon}}\right] \tag{C.20}$$

$$+ \log \frac{\mu D_0^2}{\varepsilon} \sqrt{\frac{LK}{\mu^2 \varepsilon}} \sqrt{\frac{\rho^2 d}{K} \log^2 \frac{MT}{\delta} + \frac{K\|2\boldsymbol{\sigma}\|_{2\alpha}^{2\alpha}}{\rho^{2(\alpha-1)}}}.$$

Hence the optimal $\rho$ is given by

$$\rho \asymp \max\left\{\|\boldsymbol{\sigma}\|_{2\alpha}\left(\frac{K}{\sqrt{d}\log \frac{MT}{\delta}}\right)^{1/\alpha}, \left(\frac{\|\boldsymbol{\sigma}\|_{2\alpha}^{2\alpha}}{\mu\varepsilon}\right)^{\frac{1}{2(\alpha-1)}}, \sigma_\infty, G_\infty\right\}. \tag{C.21}$$

Note that $\left(\frac{\|\boldsymbol{\sigma}\|_{2\alpha}^{2\alpha}}{\mu\varepsilon}\right)^{\frac{1}{2(\alpha-1)}} \gtrsim G_\infty \vee \sigma_\infty$ and this implies

$$T \gtrsim \log \frac{\mu D_0^2}{\varepsilon} \left[\frac{L}{\mu} + \frac{\sigma^2}{\mu M \varepsilon} \log \frac{T}{\delta} + \sqrt{\frac{L\sigma^2 K \log \frac{MT}{\delta}}{\mu^2 \varepsilon}}\right]$$

$$+ \log \frac{\mu D_0^2}{\varepsilon} \sqrt{\frac{L}{\mu^2 \varepsilon} \cdot \left[\|\boldsymbol{\sigma}\|_{2\alpha}^2 K^{\frac{2}{\alpha}} \left(d \log^2 \frac{MT}{\delta}\right)^{1 - \frac{1}{\alpha}} + \left(\frac{\|\boldsymbol{\sigma}\|_{2\alpha}^{2\alpha}}{\mu\varepsilon}\right)^{\frac{1}{(\alpha-1)}} d \log^2 \frac{MT}{\delta}\right]}$$

$$\asymp \log \frac{\mu D_0^2}{\varepsilon} \left[\frac{L}{\mu} + \frac{\sigma^2}{\mu M \varepsilon} \log \frac{T}{\delta} + \sqrt{\frac{L\sigma^2 K \log \frac{MT}{\delta}}{\mu^2 \varepsilon}} + \sqrt{\frac{Ld}{\mu^2 \varepsilon}} \log \frac{MT}{\delta} \left(\frac{\|\boldsymbol{\sigma}\|_{2\alpha}^{2\alpha}}{\mu\varepsilon}\right)^{\frac{1}{2(\alpha-1)}}\right]. \tag{C.22}$$

In the last equation we use $K \gtrsim \log \frac{MT}{\delta} \left(\frac{\|\boldsymbol{\sigma}\|_{2\alpha} d^{\frac{1}{2} - \frac{1}{2\alpha}}}{\sigma}\right)^{\frac{2\alpha}{\alpha-2}}$. This completes the proof. $\quad\square$

**Theorem C.5** (Complete version of Theorem 2). *Under the conditions of Theorem C.3 and $\mu = 0$, assume $1 - \beta_1 = \Omega(1)$, $\left( \frac{D_0 \|\boldsymbol{\sigma}\|_{2\alpha}^{\alpha}}{\varepsilon} \right)^{\frac{1}{\alpha-1}} \gtrsim G_\infty \vee \sigma_\infty$, and $K \gtrsim \log \frac{MT}{\delta} \left( \frac{\|\boldsymbol{\sigma}\|_{2\alpha} d^{\frac{1}{2} - \frac{1}{2\alpha}}}{\sigma} \right)^{\frac{2\alpha}{\alpha-2}}$. Then with probability no less than $1 - \delta$, Local SGDM with optimal $\eta, \rho$ yields $f(\hat{x}) - f_* \leq \varepsilon$ if*

$$T \gtrsim \frac{D_0^2}{\varepsilon} \left[ L + \frac{\sigma^2}{M\varepsilon} \log \frac{T}{\delta} + \sqrt{\frac{L\sigma^2 K \log \frac{MT}{\delta}}{\varepsilon}} + \sqrt{\frac{dL}{\varepsilon}} \left( \frac{D_0 \|\boldsymbol{\sigma}\|_{2\alpha}^{\alpha}}{\varepsilon} \right)^{\frac{1}{\alpha-1}} \log \frac{MT}{\delta} \right]. \quad \text{(C.23)}$$

*And equivalently,*

$$f(\hat{x}) - f_* \lesssim \frac{LD_0^2}{KR} + \frac{\sigma D_0}{\sqrt{MKR}} \log^{\frac{1}{2}} \frac{KR}{\delta}$$

$$+ \frac{L^{\frac{1}{3}} \sigma^{\frac{2}{3}} D_0^{\frac{4}{3}}}{K^{\frac{1}{3}} R^{\frac{2}{3}}} \log^{\frac{1}{3}} \frac{MKR}{\delta} + \left( \|\boldsymbol{\sigma}\|_{2\alpha}^{\frac{2\alpha}{\alpha-1}} dLD_0 \right)^{\frac{\alpha-1}{3\alpha-1}} D_0 \left( \frac{\log \frac{MKR}{\delta}}{KR} \right)^{\frac{2(\alpha-1)}{3\alpha-1}}. \quad \text{(C.24)}$$

*Proof.* Plug the definition of $A$ in (C.11),

$$T \gtrsim \frac{D_0^2}{\varepsilon} \left[ L + \frac{\sigma^2}{M\varepsilon} \log \frac{T}{\delta} + \sqrt{\frac{L\sigma^2 K \log \frac{MT}{\delta}}{\varepsilon}} + \frac{\rho \sqrt{d}}{D_0} \log \frac{T}{\delta} \right]$$

$$+ \frac{D_0^2}{\varepsilon} \sqrt{\frac{LK}{\varepsilon}} \sqrt{\frac{\rho^2 d}{K} \log^2 \frac{MT}{\delta} + \frac{K \|2\boldsymbol{\sigma}\|_{2\alpha}^{2\alpha}}{\rho^{2(\alpha-1)}}}$$

$$\asymp \frac{D_0^2}{\varepsilon} \left[ L + \frac{\sigma^2}{M\varepsilon} \log \frac{T}{\delta} + \sqrt{\frac{L\sigma^2 K \log \frac{MT}{\delta}}{\varepsilon}} + \sqrt{\frac{LK}{\varepsilon}} \sqrt{\frac{\rho^2 d}{K} \log^2 \frac{MT}{\delta} + \frac{K \|2\boldsymbol{\sigma}\|_{2\alpha}^{2\alpha}}{\rho^{2(\alpha-1)}}} \right]. \quad \text{(C.25)}$$

Hence the optimal $\rho$ is given by

$$\rho \asymp \max \left\{ \|\boldsymbol{\sigma}\|_{2\alpha} \left( \frac{K}{\sqrt{d} \log \frac{MT}{\delta}} \right)^{1/\alpha}, \left( \frac{D_0 \|\boldsymbol{\sigma}\|_{2\alpha}^{\alpha}}{\varepsilon} \right)^{\frac{1}{\alpha-1}}, \sigma_\infty, G_\infty \right\}. \quad \text{(C.26)}$$

Note that $\left( \frac{D_0 \|\boldsymbol{\sigma}\|_{2\alpha}^{\alpha}}{\varepsilon} \right)^{\frac{1}{\alpha-1}} \gtrsim G_\infty \vee \sigma_\infty$ and this implies

$$T \gtrsim \frac{D_0^2}{\varepsilon} \left[ L + \frac{\sigma^2}{M\varepsilon} \log \frac{T}{\delta} + \sqrt{\frac{L\sigma^2 K \log \frac{MT}{\delta}}{\varepsilon}} \right]$$

$$+ \frac{D_0^2}{\varepsilon} \sqrt{\frac{L}{\varepsilon} \cdot \left[ \|\boldsymbol{\sigma}\|_{2\alpha}^2 K^{\frac{2}{\alpha}} \left( d \log^2 \frac{MT}{\delta} \right)^{1 - \frac{1}{\alpha}} + \left( \frac{D_0 \|\boldsymbol{\sigma}\|_{2\alpha}^{\alpha}}{\varepsilon} \right)^{\frac{2}{\alpha-1}} d \log^2 \frac{MT}{\delta} \right]} \quad \text{(C.27)}$$

$$\asymp \frac{D_0^2}{\varepsilon} \left[ L + \frac{\sigma^2}{M\varepsilon} \log \frac{T}{\delta} + \sqrt{\frac{L\sigma^2 K \log \frac{MT}{\delta}}{\varepsilon}} + \sqrt{\frac{dL}{\varepsilon}} \left( \frac{D_0 \|\boldsymbol{\sigma}\|_{2\alpha}^{\alpha}}{\varepsilon} \right)^{\frac{1}{\alpha-1}} \log \frac{MT}{\delta} \right].$$

In the last equation we use $K \gtrsim \log \frac{MT}{\delta} \left( \frac{\|\boldsymbol{\sigma}\|_{2\alpha} d^{\frac{1}{2} - \frac{1}{2\alpha}}}{\sigma} \right)^{\frac{2\alpha}{\alpha-2}}$. Solve $\varepsilon$ and we get the upper bound of $f(\hat{x}) - f_*$. This completes the proof. $\qquad \square$

## C.2 Preliminaries

In this subsection, we show that event $E_t$ implies all the iterates remain in certain area, so that we can apply all kinds of properties of $f$ afterwards.

**Lemma C.6.** *If $\eta\sigma\sqrt{KA} \leq (\sqrt{3} - \sqrt{2})D_0$, Event $E_t$ implies that for all $j \leq t, m \in [M]$, we have $x_j^m, \overline{x}_j, z_j^m, \overline{z}_j \in \Omega$. And $\|x_j^m - x_j^n\| \leq \eta\sigma\sqrt{KA}$ for all $m, n$.*

*Proof.* Event $E_t$ implies that for all $j \leq t$,

$$\|\overline{z}_j - x_*\| \leq \sqrt{2}D_0, \ \|z_j^m - z_j^n\| \leq \eta\sigma\sqrt{KA} \leq (\sqrt{3} - \sqrt{2})D_0. \tag{C.28}$$

Hence $\overline{z}_j \in \Omega, \|z_j^m - x_*\| \leq \sqrt{3}D_0$ and $z_j^m \in \Omega$. Also, notice that $\overline{x}_j \in \mathbf{conv}\{\overline{z}_i\}_{i \leq j}$ and $x_j^m - x_j^n \in \mathbf{conv}\{z_i^m - z_i^n\}_{i \leq j}$. We have

$$\|\overline{x}_j - x_*\| \leq \sqrt{2}D_0, \ \|x_j^m - x_j^n\| \leq \eta\sigma\sqrt{KA}, \ \|x_j^m - \overline{x}_j\| \leq \eta\sigma\sqrt{KA} \leq (\sqrt{3} - \sqrt{2})D_0. \tag{C.29}$$

Therefore $x_j^m, \overline{x}_j \in \Omega$. This completes the proof. $\qquad\square$

## C.3 Proof of Contraction Lemma C.1

In this subsection, we aim to show contraction, *i.e.*, $\|x_t^m - x_t^n\|$ won't be too large during local iterations with high probability. This property is crucial for showing the benefits of local updates in distributed optimization. However, different from (Woodworth et al., 2020a; Khaled et al., 2020), the update of $x_t^m$ is in the direction of momentum $u_t^m$, which incorporates information from all past gradient. Therefore, we cannot directly apply $\langle x_t^m - x_t^n, \mathbb{E}_t[u_t^m - u_t^n]\rangle \geq 0$. Fortunately, noticing that $x_t^m - x_t^n \in \mathbf{conv}(\{z_j^m - z_j^n\}_{j \leq t})$, it suffices to show that $\|z_t^m - z_t^n\|$ won't get too large with high probability. Besides, the update rule of $z_t^m$ is much easier to handle.

*Proof.* First note that by the upper bound of $\eta$, Lemma C.6 holds. Since $z_{t+1}^m = z_t^m - \eta\widehat{g_t^m}$,

$$\|z_{t+1}^m - z_{t+1}^n\|^2 = \|z_t^m - z_t^n\|^2 - 2\eta\left\langle z_t^m - z_t^n, \widehat{g_t^m} - \widehat{g_t^n}\right\rangle + \eta^2\|\widehat{g_t^m} - \widehat{g_t^n}\|^2$$
$$\leq \|z_t^m - z_t^n\|^2 - 2\eta\langle z_t^m - z_t^n, \nabla f(x_t^m) - \nabla f(x_t^n)\rangle + 2\eta^2\|\nabla f(x_t^m) - \nabla f(x_t^n)\|^2$$
$$+ 2\eta\left\langle z_t^m - z_t^n, \nabla f(x_t^m) - \nabla f(x_t^n) - \widehat{g_t^m} + \widehat{g_t^n}\right\rangle + 2\eta^2\|\nabla f(x_t^m) - \nabla f(x_t^n) - \widehat{g_t^m} + \widehat{g_t^n}\|^2. \tag{C.30}$$

Event $E_t$ implies $z_t^m, x_t^m \in \Omega$ and thus by $\forall x, y \in \Omega, \langle x - y, \nabla f(x) - \nabla f(y)\rangle \geq \frac{1}{L}\|\nabla f(x) - \nabla f(y)\|^2$,

$$\langle z_t^m - z_t^n, \nabla f(x_t^m) - \nabla f(x_t^n)\rangle = \langle x_t^m - x_t^n, \nabla f(x_t^m) - \nabla f(x_t^n)\rangle + \langle z_t^m - z_t^n - (x_t^m - x_t^n), \nabla f(x_t^m) - \nabla f(x_t^n)\rangle$$
$$\geq \langle x_t^m - x_t^n, \nabla f(x_t^m) - \nabla f(x_t^n)\rangle$$
$$- \left[L\|z_t^m - z_t^n - (x_t^m - x_t^n)\|^2 + \frac{1}{4L}\|\nabla f(x_t^m) - \nabla f(x_t^n)\|^2\right]$$
$$\geq \frac{3}{4L}\|\nabla f(x_t^m) - \nabla f(x_t^n)\|^2 - L\|z_t^m - z_t^n - (x_t^m - x_t^n)\|^2. \tag{C.31}$$

Therefore, for the second and third term in the RHS of (C.30),

$$-2\eta\langle z_t^m - z_t^n, \nabla f(x_t^m) - \nabla f(x_t^n)\rangle + 2\eta^2\|\nabla f(x_t^m) - \nabla f(x_t^n)\|^2$$
$$\leq -\frac{\eta}{L}\|\nabla f(x_t^m) - \nabla f(x_t^n)\|^2 + 2\eta L\|z_t^m - z_t^n - (x_t^m - x_t^n)\|^2. \tag{C.32}$$

By the update rule,

$$\|z_t^m - z_t^n - (x_t^m - x_t^n)\|^2 = \left(\frac{\eta\beta_1}{1 - \beta_1}\right)^2\|u_{t-1}^m - u_{t-1}^n\|^2$$
$$\leq \left(\frac{\eta\beta_1}{1 - \beta_1}\right)^2\left\|(1 - \beta_1)\sum_{j=rK}^{t-1}\beta_1^{t-j-1}[\widehat{g_k^m} - \widehat{g_k^n}]\right\|^2$$
$$\leq \frac{2(\eta\beta_1)^2}{1 - \beta_1}\sum_{j=rK}^{t-1}\beta_1^{t-j-1}\left[\|\nabla f(x_j^m) - \nabla f(x_j^n)\|^2 + \|\widehat{g_j^m} - \widehat{g_j^n} - \nabla f(x_j^m) + \nabla f(x_j^n)\|^2\right]. \tag{C.33}$$

Let $S_t := \sum_{j=rK}^{t} \beta_1^{t-j} \|\nabla f(x_j^m) - \nabla f(x_j^n)\|^2$. We further get

$$\text{LHS of (C.32)} \leq -\frac{\eta}{L}(S_t - \beta_1 S_{t-1}) + \frac{4\eta L(\eta\beta_1)^2}{1-\beta_1}\left[S_{t-1} + \sum_{j=rK}^{t-1} \beta_1^{t-j-1}[\|\widehat{g_j^m} - \widehat{g_j^n} - \nabla f(x_j^m) + \nabla f(x_j^n)\|^2]\right]$$

$$= -\frac{\eta}{L}(S_t - S_{t-1}) + \frac{4\eta L(\eta\beta_1)^2}{1-\beta_1}\left[\sum_{j=rK}^{t-1} \beta_1^{t-j-1}[\|\widehat{g_j^m} - \widehat{g_j^n} - \nabla f(x_j^m) + \nabla f(x_j^n)\|^2]\right]$$

$$\tag{C.34}$$

Then plug in (C.30),

$$\|z_{t+1}^m - z_{t+1}^n\|^2 \leq \|z_t^m - z_t^n\|^2 - \frac{\eta}{L}(S_t - S_{t-1})$$

$$+ \frac{4\eta L(\eta\beta_1)^2}{1-\beta_1}\left[\sum_{j=rK}^{t-1} \beta_1^{t-j-1}[\|\widehat{g_j^m} - \widehat{g_j^n} - \nabla f(x_j^m) + \nabla f(x_j^n)\|^2]\right]$$

$$+ 2\eta\left\langle z_t^m - z_t^n, \nabla f(x_t^m) - \nabla f(x_t^n) - \widehat{g_t^m} + \widehat{g_t^n}\right\rangle + 2\eta^2\|\widehat{g_t^m} - \widehat{g_t^n} - \nabla f(x_t^m) + \nabla f(x_t^n)\|^2.$$

$$\tag{C.35}$$

Notice that this recursive bound holds for any $rK \leq i \leq t$. Unroll it and recalculate the coefficients using $\eta L \leq (1-\beta_1)^2/2$,

$$\|z_{t+1}^m - z_{t+1}^n\|^2 + \frac{\eta}{L}S_t \leq \sum_{j=rK}^{t} 2\eta\left\langle z_j^m - z_j^n, \nabla f(x_j^m) - \nabla f(x_j^n) - \widehat{g_j^m} + \widehat{g_j^n}\right\rangle$$

$$+ \sum_{j=rK}^{t} 4\eta^2\|\nabla f(x_j^m) - \nabla f(x_j^n) - \widehat{g_j^m} + \widehat{g_j^n}\|^2$$

$$\leq \underbrace{\sum_{j=rK}^{t} 2\eta\left\langle z_j^m - z_j^n, \mathbb{E}_j[\widehat{g_j^m} - \widehat{g_j^n}] - [\widehat{g_j^m} - \widehat{g_j^n}]\right\rangle}_{\text{①: martingale}}$$

$$+ \underbrace{\sum_{j=rK}^{t} 2\eta\left\langle z_j^m - z_j^n, \nabla f(x_j^m) - \nabla f(x_j^n) - \mathbb{E}_j[\widehat{g_j^m} - \widehat{g_j^n}]\right\rangle}_{\text{②: clipping bias}}$$

$$+ \underbrace{\sum_{j=rK}^{t} 4\eta^2\left[\|\nabla f(x_j^m) - \nabla f(x_j^n) - \widehat{g_j^m} + \widehat{g_j^n}\|^2 - \mathbb{E}_j[\|\nabla f(x_j^m) - \nabla f(x_j^n) - [\widehat{g_j^m} - \widehat{g_j^n}]\|^2]\right]}_{\text{③: martingale}}$$

$$+ 4\eta^2 K \cdot 2\sigma^2.$$

$$\tag{C.36}$$

For ①, define

$$\zeta_j^{m,n} = \begin{cases} 2\eta\left\langle z_j^m - z_j^n, \mathbb{E}_j[\widehat{g_j^m} - \widehat{g_j^n}] - [\widehat{g_j^m} - \widehat{g_j^n}]\right\rangle, & \text{if event } E_j \text{ holds,} \\ 0, & \text{otherwise.} \end{cases} \tag{C.37}$$

Then since event $E_j$ implies $\|z_j^m - z_j^n\| \leq \eta\sigma\sqrt{KA}$,

$$|\zeta_j^{m,n}| \leq 2\eta \cdot \eta\sigma\sqrt{KA} \cdot 2\rho\sqrt{d} = 4\eta^2\sigma\rho\sqrt{dKA} \stackrel{def}{=} c, \tag{C.38}$$

$$\text{Var}_j(\zeta_j^{m,n}) \leq 4\eta^2 \cdot \eta^2\sigma^2 KA \cdot 2\sigma^2 = 8\eta^4\sigma^4 KA. \tag{C.39}$$

Let $b = \frac{1}{4}\eta^2\sigma^2 KA$, $V = 8\eta^4\sigma^4 K^2 A$. By Lemma B.1, $|\sum_{j=0}^{t} \zeta_j^{m,n}| \leq b$ with probability no less than

$$1 - 2\exp\left(\frac{b^2}{2V + 2cb/3}\right) \geq 1 - \frac{\delta}{4M^2 T}. \tag{C.40}$$

For ②,

$$|②| \leq 2\eta K \cdot \eta\sigma\sqrt{KA} \cdot 2\frac{\|2\boldsymbol{\sigma}\|_{2\alpha}^\alpha}{\rho^{(\alpha-1)}} \leq \frac{1}{4}\eta^2\sigma^2 KA. \tag{C.41}$$

For ③, define

$$\theta_j^{m,n} = \begin{cases} 4\eta^2\left[\|\nabla f(x_j^m) - \nabla f(x_j^n) - \widehat{g_j^m} + \widehat{g_j^n}\|^2 - \mathbb{E}_j[\|\nabla f(x_j^m) - \nabla f(x_j^n) - [\widehat{g_j^m} - \widehat{g_j^n}]\|^2]\right], & \text{if event } E_j \text{ holds,} \\ 0, & \text{otherwise.} \end{cases} \tag{C.42}$$

Then,

$$|\theta_j^{m,n}| \leq 4\eta^2 \cdot 4\rho^2 d = 16\eta^2\rho^2 d \stackrel{def}{=} c, \tag{C.43}$$

$$\text{Var}_j(\theta_j^{m,n}) \leq 16\eta^4 \cdot \mathbb{E}_j[\|\nabla f(x_j^m) - \nabla f(x_j^n) - [\widehat{g_j^m} - \widehat{g_j^n}]\|^2]^2 \leq 64\eta^4\sigma^4. \tag{C.44}$$

Let $b = \frac{1}{4}\eta^2\sigma^2 KA$, $V = 64K\eta^4\sigma^4$. By Lemma B.1, $|\sum_{j=0}^{t} \theta_j^{m,n}| \leq b$ with probability no less than

$$1 - 2\exp\left(\frac{b^2}{2V + 2cb/3}\right) \geq 1 - \frac{\delta}{4M^2 T}. \tag{C.45}$$

Combine ①, ②, ③ and thus we can conclude that with probability no less than $\mathbb{P}(E_t) - 2 \cdot \frac{\delta}{4T}$, event $E_t$ holds and $\|z_{t+1}^m - z_{t+1}^n\|^2 \leq \eta^2\sigma^2 KA$ for all $m, n$. This completes the proof. $\qquad\square$

## C.4 PROOF OF DESCENT LEMMA C.2

Now we are ready to state the main descent lemma of *Local* SGDM.

*Proof.* Again, note that by the upper bound of $\eta$, Lemma C.6 holds. Under event $E_t$,

$$\|\overline{z}_{t+1} - x_*\|^2 = \|\overline{z}_t - x_*\|^2 - 2\eta\left\langle\overline{z}_t - x_*, \mathbb{E}_m[\widehat{g_t^m}]\right\rangle + \eta^2\|\mathbb{E}_m[\widehat{g_t^m}]\|^2$$

$$\leq \|\overline{z}_t - x_*\|^2 - 2\eta\left\langle\overline{z}_t - x_*, \mathbb{E}_m[\nabla f(x_t^m)]\right\rangle - 2\eta\left\langle\overline{z}_t - x_*, \mathbb{E}_m[\widehat{g_t^m} - \nabla f(x_t^m)]\right\rangle$$

$$+ 2\eta^2\|\mathbb{E}_m[\widehat{g_t^m} - \nabla f(x_t^m)]\|^2 + 2\eta^2\|\mathbb{E}_m[\nabla f(x_t^m)]\|^2. \tag{C.46}$$

Since $x_t^m, \overline{x}_t, \overline{z}_t \in \Omega$, for the second term,

$$\langle\overline{z}_t - x_*, \mathbb{E}_m[\nabla f(x_t^m)]\rangle = \langle\overline{x}_t - x_*, \mathbb{E}_m[\nabla f(x_t^m)]\rangle + \langle\overline{z}_t - \overline{x}_t, \mathbb{E}_m[\nabla f(x_t^m)]\rangle$$

$$= \mathbb{E}_m\left[\langle\overline{x}_t - x_t^m, \nabla f(x_t^m)\rangle + \langle x_t^m - x_*, \nabla f(x_t^m)\rangle\right]$$

$$+ \langle\overline{z}_t - \overline{x}_t, \nabla f(\overline{x}_t)\rangle + \langle\overline{z}_t - \overline{x}_t, \mathbb{E}_m[\nabla f(x_t^m) - \nabla f(\overline{x}_t)]\rangle. \tag{C.47}$$

By smoothness,

$$\mathbb{E}_m\left[\langle\overline{x}_t - x_t^m, \nabla f(x_t^m)\rangle\right] \geq -L\mathbb{E}_m[\|x_t^m - \overline{x}_t\|^2], \tag{C.48}$$

$$f(\overline{z}_t) \leq f(\overline{x}_t) + \langle\overline{z}_t - \overline{x}_t, \nabla f(\overline{x}_t)\rangle + \frac{L}{2}\|\overline{x}_t - \overline{z}_t\|^2. \tag{C.49}$$

By $\mu$-strong convexity,

$$\mathbb{E}_m\left[\langle x_t^m - x_*, \nabla f(x_t^m)\rangle\right] \geq \mathbb{E}_m[f(x_t^m) - f_* + \frac{\mu}{2}\|x_t^m - x_*\|^2]$$

$$\geq f(\overline{x}_t) - f_* + \frac{\mu}{2}\|\overline{x}_t - x_*\|^2. \tag{C.50}$$

Therefore,

$$\langle \overline{z}_t - x_*, \mathbb{E}_m[\nabla f(x_t^m)] \rangle = \langle \overline{x}_t - x_*, \mathbb{E}_m[\nabla f(x_t^m)] \rangle + \langle \overline{z}_t - \overline{x}_t, \mathbb{E}_m[\nabla f(x_t^m)] \rangle$$

$$\overset{(C.48),(C.50)}{\geq} f(\overline{x}_t) - f_* + \frac{\mu}{2}\|\overline{x}_t - x_*\|^2 - L\mathbb{E}_m[\|x_t^m - \overline{x}_t\|^2]$$
$$+ \langle \overline{z}_t - \overline{x}_t, \nabla f(\overline{x}_t) \rangle + \langle \overline{z}_t - \overline{x}_t, \mathbb{E}_m[\nabla f(x_t^m) - \nabla f(\overline{x}_t)] \rangle$$

$$\overset{(C.49),\ \text{AM-GM}}{\geq} f(\overline{z}_t) - f_* + \frac{\mu}{2}\|\overline{x}_t - x_*\|^2 - \frac{L}{2}\|\overline{z}_t - \overline{x}_t\|^2 - L\mathbb{E}_m[\|x_t^m - \overline{x}_t\|^2]$$
$$- \frac{L}{2}\left(\|\overline{z}_t - \overline{x}_t\|^2 + \mathbb{E}_m[\|x_t^m - \overline{x}_t\|^2]\right)$$

$$\overset{\text{AM-GM}}{\geq} f(\overline{z}_t) - f_* + \frac{\mu}{4}\|\overline{z}_t - x_*\|^2 - \frac{3L}{2}\left(\|\overline{z}_t - \overline{x}_t\|^2 + \mathbb{E}_m[\|x_t^m - \overline{x}_t\|^2]\right).$$
$$\tag{C.51}$$

For the last term in (C.46),

$$2\eta^2\|\mathbb{E}_m[\nabla f(x_t^m)]\|^2 \leq 6\eta^2\left[L^2\|x_t^m - \overline{x}_t\|^2 + L^2\|\overline{x}_t - \overline{z}_t\|^2 + \|\nabla f(\overline{z}_t)\|^2\right]$$
$$\leq 6\eta^2\left[L^2\|x_t^m - \overline{x}_t\|^2 + L^2\|\overline{x}_t - \overline{z}_t\|^2 + \frac{1}{2L}(f(\overline{z}_t) - f_*)\right]$$
$$\tag{C.52}$$

Combine all these inequalities plugging in (C.46) and notice that $\eta \leq \frac{1}{6L}$,

$$\|\overline{z}_{t+1} - x_*\|^2 \leq (1 - \frac{\eta\mu}{2})\|\overline{z}_t - x_*\|^2 - \eta(f(\overline{z}_t) - f_*) + 4\eta L\left[\|\overline{z}_t - \overline{x}_t\|^2 + \mathbb{E}_m[\|x_t^m - \overline{x}_t\|^2]\right]$$
$$- 2\eta\left\langle \overline{z}_t - x_*, \mathbb{E}_m[\widehat{g_t^m} - \nabla f(x_t^m)] \right\rangle + 2\eta^2\|\mathbb{E}_m[\widehat{g_t^m} - \nabla f(x_t^m)]\|^2.$$
$$\tag{C.53}$$

Define $\Lambda_t := \sum_{j=0}^{t-1} a_{t,j}\|\overline{x}_j - \overline{x}_{j+1}\|^2$, where $a_{t,j} := \beta_1^{t-j-1}(t - j + \frac{\beta_1}{1 - \beta_1})$. By Lemma C.7,

we plug (C.85) in the above inequality and compute $(C.53) + \frac{2^8(\eta L)^3\beta_1^2}{(1-\beta_1)^4} \times (C.84)$. Now let

$\Phi_t := \|\overline{z}_t - x_*\|^2 + \frac{2^8(\eta L)^3\beta_1^2}{(1-\beta_1)^4}\Lambda_{t-1}$. Hence we obtain

$$\Phi_{t+1} \leq (1 - \frac{\eta\mu}{2})\Phi_t - \eta(f(\overline{z}_t) - f_*) + 4\eta L\left[\mathbb{E}_m[\|x_t^m - \overline{x}_t\|^2] + 64\left(\frac{\eta\beta_1}{1-\beta_1}\right)^2\|\nabla f(\overline{z}_t)\|^2\right]$$
$$+ 32\eta L\left(\frac{\eta\beta_1}{1-\beta_1}\right)^2\left[(1-\beta_1)\sum_{j=0}^{t-1}\beta_1^{t-j-1}\left[2L^2\mathbb{E}_m[\|x_j^m - \overline{x}_j\|^2] + \|\mathbb{E}_m[\widehat{g_j^m} - \nabla f(x_j^m)]\|^2\right]\right]$$
$$- 2\eta\left\langle \overline{z}_t - x_*, \mathbb{E}_m[\widehat{g_t^m} - \nabla f(x_t^m)] \right\rangle + 2\eta^2\|\mathbb{E}_m[\widehat{g_t^m} - \nabla f(x_t^m)]\|^2$$

$$\leq (1 - \frac{\eta\mu}{2})\Phi_t - \frac{\eta}{2}(f(\overline{z}_t) - f_*) + 4\eta L\mathbb{E}_m[\|x_t^m - \overline{x}_t\|^2]$$
$$+ 32\eta L\left(\frac{\eta\beta_1}{1-\beta_1}\right)^2\left[(1-\beta_1)\sum_{j=0}^{t-1}\beta_1^{t-j-1}\left[2L^2\mathbb{E}_m[\|x_j^m - \overline{x}_j\|^2] + \|\mathbb{E}_m[\widehat{g_j^m} - \nabla f(x_j^m)]\|^2\right]\right]$$
$$- 2\eta\left\langle \overline{z}_t - x_*, \mathbb{E}_m[\widehat{g_t^m} - \nabla f(x_t^m)] \right\rangle + 2\eta^2\|\mathbb{E}_m[\widehat{g_t^m} - \nabla f(x_t^m)]\|^2$$

$$\leq (1 - \frac{\eta\mu}{2})\Phi_t - \frac{\eta}{2}(f(\overline{z}_t) - f_*) + 16\eta L \cdot \eta^2\sigma^2 KA$$
$$+ 32\eta L\left(\frac{\eta\beta_1}{1-\beta_1}\right)^2\left[(1-\beta_1)\sum_{j=0}^{t-1}\beta_1^{t-j-1}\|\mathbb{E}_m[\widehat{g_j^m} - \nabla f(x_j^m)]\|^2\right]$$
$$- 2\eta\left\langle \overline{z}_t - x_*, \mathbb{E}_m[\widehat{g_t^m} - \nabla f(x_t^m)] \right\rangle + 2\eta^2\|\mathbb{E}_m[\widehat{g_t^m} - \nabla f(x_t^m)]\|^2.$$
$$\tag{C.54}$$

Here in the second inequality we use $\|\nabla f(\bar{z}_t)\|^2 \leq 2L(f(\bar{z}_t) - f_*)$. In the last inequality, we apply contraction results implied by event $E_{t,1}$.

Unroll this recursive bound and re-calculate the coefficients,

$$\sum_{j=0}^{t} \frac{\eta}{2}(f(\bar{z}_j) - f_*)(1 - \frac{\eta\mu}{2})^{t-j} + \Phi_{t+1} \leq (1 - \frac{\eta\mu}{2})^{t+1}\Phi_0 + \frac{32\eta^2 L\sigma^2 KA}{\mu}$$
$$- 2\eta\sum_{j=0}^{t}(1 - \frac{\eta\mu}{2})^{t-j}\left\langle \bar{z}_j - x_*, \mathbb{E}_m[\widehat{g_j^m} - \nabla f(x_j^m)]\right\rangle$$
$$+ 4\eta^2\sum_{j=0}^{t}(1 - \frac{\eta\mu}{2})^{t-j}\|\mathbb{E}_m[\widehat{g_j^m} - \nabla f(x_j^m)]\|^2$$

(C.55)

Simplify $\Phi_{t+1}$ term,

$$\sum_{j=0}^{t} \frac{\eta}{2}(f(\bar{z}_j) - f_*)(1 - \frac{\eta\mu}{2})^{t-j} + \|\bar{z}_{t+1} - x_*\|^2 \leq (1 - \frac{\eta\mu}{2})^{t+1}\|x_0 - x_*\|^2 + \frac{32\eta^2 L\sigma^2 KA}{\mu}$$
$$\underbrace{- 2\eta\sum_{j=0}^{t}(1 - \frac{\eta\mu}{2})^{t-j}\left\langle \bar{z}_j - x_*, \mathbb{E}_m[\widehat{g_j^m} - \mathbb{E}_j[\widehat{g_j^m}]]\right\rangle}_{\text{①: martingale}}$$
$$\underbrace{- 2\eta\sum_{j=0}^{t}(1 - \frac{\eta\mu}{2})^{t-j}\left\langle \bar{z}_j - x_*, \mathbb{E}_m[\mathbb{E}_j[\widehat{g_j^m}] - \nabla f(x_j^m)]\right\rangle}_{\text{②: clipping bias}}$$
$$+ 4\eta^2\sum_{j=0}^{t}(1 - \frac{\eta\mu}{2})^{t-j}\|\mathbb{E}_m[\widehat{g_j^m} - \nabla f(x_j^m)]\|^2.$$

(C.56)

For the last term,

$$4\eta^2\sum_{j=0}^{t}(1 - \frac{\eta\mu}{2})^{t-j}\|\mathbb{E}_m[\widehat{g_j^m} - \nabla f(x_j^m)]\|^2 \leq \underbrace{8\eta^2\sum_{j=0}^{t}(1 - \frac{\eta\mu}{2})^{t-j}\left[\|\mathbb{E}_m[\widehat{g_j^m} - \mathbb{E}_j[\widehat{g_j^m}]]\|^2 - \mathbb{E}_j[\|\mathbb{E}_m[\widehat{g_j^m} - \mathbb{E}_j[\widehat{g_j^m}]]\|^2]\right]}_{\text{③: martingale}}$$
$$+ \underbrace{8\eta^2\sum_{j=0}^{t}(1 - \frac{\eta\mu}{2})^{t-j}\mathbb{E}_j[\|\mathbb{E}_m[\widehat{g_j^m} - \mathbb{E}_j[\widehat{g_j^m}]]\|^2]}_{\text{Lemma B.2}}$$
$$+ \underbrace{8\eta^2\sum_{j=0}^{t}(1 - \frac{\eta\mu}{2})^{t-j}\|\mathbb{E}_m[\mathbb{E}_j[\widehat{g_j^m}] - \nabla f(x_j^m)]\|^2,}_{\text{④: clipping bias}}$$

(C.57)

we finally get

$$\sum_{j=0}^{t} \frac{\eta}{2}(f(\bar{z}_j) - f_*)(1 - \frac{\eta\mu}{2})^{t-j} + \|\bar{z}_{t+1} - x_*\|^2 \leq (1 - \frac{\eta\mu}{2})^{t+1}D_0^2 + 32\left[\eta LKA + \frac{1}{M}\right]\frac{\eta\sigma^2}{\mu}$$
$$+ ① + ② + ③ + ④.$$

(C.58)

(1) **Case** $\mu > 0$.

For ①, define

$$\zeta_j = \begin{cases} -2\eta(1 - \frac{\eta\mu}{2})^{t-j} \left\langle \overline{z}_j - x_*, \mathbb{E}_m[\widehat{g_j^m} - \mathbb{E}_j[\widehat{g_j^m}]] \right\rangle, & \text{if event } E_j \text{ holds,} \\ 0, & \text{otherwise.} \end{cases} \tag{C.59}$$

Then since event $E_j$ implies $\|\overline{z}_j - x_*\| \leq \sqrt{2}(1 - \frac{\eta\mu}{2})^{j/2}D_0$,

$$|\zeta_j| \leq 2\eta \cdot \sqrt{2}(1 - \frac{\eta\mu}{2})^{t/2}D_0 \cdot 2\rho\sqrt{d} = 4(1 - \frac{\eta\mu}{2})^{t/2}\eta\rho\sqrt{2d}D_0 \stackrel{def}{=} c, \tag{C.60}$$

$$\text{Var}_j(\zeta_j) \leq 4\eta^2(1 - \frac{\eta\mu}{2})^{2(t-j)} \cdot 2(1 - \frac{\eta\mu}{2})^j D_0^2 \cdot \frac{\sigma^2}{M} = 8(1 - \frac{\eta\mu}{2})^{2t-j}\frac{\eta^2 D_0^2 \sigma^2}{M}. \tag{C.61}$$

Let $b = \frac{(1 - \frac{\eta\mu}{2})^{t+1}D_0^2}{5}$, $V = 16(1 - \frac{\eta\mu}{2})^t \frac{\eta D_0^2 \sigma^2}{\mu M}$. By Lemma B.1, $|\sum_{j=0}^t \zeta_j| \leq b$ with probability no less than

$$1 - 2\exp\left(\frac{b^2}{2V + 2cb/3}\right) \geq 1 - \frac{\delta}{4T}. \tag{C.62}$$

For ②, since by Lemma B.2,

$$\|\mathbb{E}_j[\widehat{g_j^m} - \nabla f(x_j^m)]\|^2 \leq \frac{\|2\boldsymbol{\sigma}\|_{2\alpha}^{2\alpha}}{\rho^{2(\alpha-1)}}, \tag{C.63}$$

event $E_t$ implies that

$$\begin{aligned} |②| &\leq 2\eta \sum_{j=0}^t (1 - \frac{\eta\mu}{2})^{t-j} \cdot \sqrt{2}(1 - \frac{\eta\mu}{2})^{j/2}D_0 \cdot \frac{\|2\boldsymbol{\sigma}\|_{2\alpha}^{\alpha}}{\rho^{\alpha-1}} \\ &\leq 4\sqrt{2}(1 - \frac{\eta\mu}{2})^{t/2}\frac{D_0\|2\boldsymbol{\sigma}\|_{2\alpha}^{\alpha}}{\mu\rho^{\alpha-1}} \\ &\leq \frac{(1 - \frac{\eta\mu}{2})^{t+1}D_0^2}{5}. \end{aligned} \tag{C.64}$$

Here we use the definition of $\eta$ and conditions of $\rho$ in (C.12).

For ③, define

$$\theta_j = \begin{cases} 8\eta^2(1 - \frac{\eta\mu}{2})^{t-j} \left[\|\mathbb{E}_m[\widehat{g_j^m} - \mathbb{E}_j[\widehat{g_j^m}]]\|^2 - \mathbb{E}_j[\|\mathbb{E}_m[\widehat{g_j^m} - \mathbb{E}_j[\widehat{g_j^m}]]\|^2]\right], & \text{if event } E_j \text{ holds,} \\ 0, & \text{otherwise.} \end{cases} \tag{C.65}$$

Then

$$|\theta_j| \leq 8\eta^2 \cdot 4\rho^2 d = 32\eta^2\rho^2 d \stackrel{def}{=} c, \tag{C.66}$$

$$\text{Var}_j(\theta_j) \leq 64\eta^4(1 - \frac{\eta\mu}{2})^{2(t-j)} \cdot \mathbb{E}_j[\|\mathbb{E}_m[\widehat{g_j^m} - \mathbb{E}_j[\widehat{g_j^m}]]\|^2]^2 \stackrel{\text{Lemma B.3}}{\leq} 64\eta^4(1 - \frac{\eta\mu}{2})^{2(t-j)} \cdot \frac{4(2\sigma)^4}{M^2}. \tag{C.67}$$

Let $b = \frac{(1 - \frac{\eta\mu}{2})^{t+1}D_0^2}{5}$, $V = \frac{2^{13}\eta^3\sigma^4}{\mu M^2}$. By Lemma B.1, $|\sum_{j=0}^t \theta_j| \leq b$ with probability no less than

$$1 - 2\exp\left(\frac{b^2}{2V + 2cb/3}\right) \geq 1 - \frac{\delta}{4T}. \tag{C.68}$$

For ④, by Lemma B.2,

$$|④| \leq \frac{16\eta}{\mu} \cdot \frac{\|2\boldsymbol{\sigma}\|_{2\alpha}^{2\alpha}}{\rho^{2(\alpha-1)}} \leq \frac{(1 - \frac{\eta\mu}{2})^{t+1}D_0^2}{5}. \tag{C.69}$$

Combine the above claims, with probability no less than $\mathbb{P}(E_{t,1}) - 2 \cdot \dfrac{\delta}{4T}$, we have $|① + ② + ③ + ④| \leq \dfrac{4}{5}(1 - \dfrac{\eta\mu}{2})^{t+1}D_0^2$. By (C.58), these implies

$$
\sum_{j=0}^{t} \frac{\eta}{2}(f(\overline{z}_j) - f_*)(1 - \frac{\eta\mu}{2})^{t-j} + \|\overline{z}_{t+1} - x_*\|^2 \leq (1 - \frac{\eta\mu}{2})^{t+1}D_0^2 + 32\left[\eta LKA + \frac{1}{M}\right]\frac{\eta\sigma^2}{\mu}
$$
$$
+ \frac{4}{5}(1 - \frac{\eta\mu}{2})^{t+1}D_0^2
$$
$$
\leq 2(1 - \frac{\eta\mu}{2})^{t+1}D_0^2.
$$
(C.70)

Therefore, we conclude that $\mathbb{P}(E_{t+1}) \geq \mathbb{P}(E_{t,1}) - \dfrac{\delta}{2T}$.

(2) **Case** $\mu = 0$.

In this case, (C.58) reduces to

$$
\frac{\eta}{2}\sum_{j=0}^{t}(f(\overline{z}_j) - f_*) + \|\overline{z}_{t+1} - x_*\|^2 \leq D_0^2 + 16\left[\eta LKA + \frac{1}{M}\right]\eta^2\sigma^2(t+1) + ① + ② + ③ + ④. \quad \text{(C.71)}
$$

For ①, define

$$
\zeta_j = \begin{cases} -2\eta\left\langle \overline{z}_j - x_*, \mathbb{E}_m[\widehat{g_j^m} - \mathbb{E}_j[\widehat{g_j^m}]]\right\rangle, & \text{if event } E_j \text{ holds,} \\ 0, & \text{otherwise.} \end{cases} \quad \text{(C.72)}
$$

Then since event $E_j$ implies $\|\overline{z}_j - x_*\| \leq \sqrt{2}D_0$,

$$
|\zeta_j| \leq 2\eta \cdot \sqrt{2}D_0 \cdot 2\rho\sqrt{d} = 4\eta\rho\sqrt{2d}D_0 \overset{def}{=} c, \quad \text{(C.73)}
$$

$$
\text{Var}_j(\zeta_j) \leq 4\eta^2 \cdot 2D_0^2 \cdot \frac{\sigma^2}{M} = \frac{8\eta^2 D_0^2\sigma^2}{M}. \quad \text{(C.74)}
$$

Let $b = \dfrac{D_0^2}{5}$, $V = \dfrac{8\eta^2 D_0^2\sigma^2 T}{M}$. By Lemma B.1, $|\sum_{j=0}^{t}\zeta_j| \leq b$ with probability no less than

$$
1 - 2\exp\left(\frac{b^2}{2V + 2cb/3}\right) \geq 1 - \frac{\delta}{4T}. \quad \text{(C.75)}
$$

For ②, since by Lemma B.2,

$$
\|\mathbb{E}_j[\widehat{g_j^m} - \nabla f(x_j^m)]\|^2 \leq \frac{\|2\boldsymbol{\sigma}\|_{2\alpha}^{2\alpha}}{\rho^{2(\alpha-1)}}, \quad \text{(C.76)}
$$

event $E_t$ implies that

$$
|②| \leq 2\eta(t+1) \cdot \sqrt{2}D_0 \cdot \frac{\|2\boldsymbol{\sigma}\|_{2\alpha}^{\alpha}}{\rho^{(\alpha-1)}} \leq \frac{D_0^2}{5}. \quad \text{(C.77)}
$$

Here we again use definitions and conditions in (C.12).

For ③, define

$$
\theta_j = \begin{cases} 8\eta^2\left[\|\mathbb{E}_m[\widehat{g_j^m} - \mathbb{E}_j[\widehat{g_j^m}]]\|^2 - \mathbb{E}_j[\|\mathbb{E}_m[\widehat{g_j^m} - \mathbb{E}_j[\widehat{g_j^m}]]\|^2]\right], & \text{if event } E_j \text{ holds,} \\ 0, & \text{otherwise.} \end{cases} \quad \text{(C.78)}
$$

Then

$$
|\theta_j| \leq 8\eta^2 \cdot 4\rho^2 d = 32\eta^2\rho^2 d \overset{def}{=} c, \quad \text{(C.79)}
$$

$$
\text{Var}_j(\theta_j) \leq 64\eta^4 \cdot \mathbb{E}_j[\|\mathbb{E}_m[\widehat{g_j^m} - \mathbb{E}_j[\widehat{g_j^m}]]\|^2]^2 \overset{\text{Lemma B.3}}{\leq} 64\eta^4 \cdot \frac{4(2\sigma)^4}{M^2}. \quad \text{(C.80)}
$$

Let $b = \dfrac{D_0^2}{5}$, $V = \dfrac{2^{12}\eta^4\sigma^4}{M^2}$. By Lemma B.1, $|\sum\limits_{j=0}^{t}\theta_j| \leq b$ with probability no less than

$$1 - 2\exp\left(\frac{b^2}{2V + 2cb/3}\right) \geq 1 - \frac{\delta}{4T}. \tag{C.81}$$

For ④, by Lemma B.2,

$$|④| \leq 8\eta^2(t+1) \cdot \frac{\|2\boldsymbol{\sigma}\|_{2\alpha}^{2\alpha}}{\rho^{2(\alpha-1)}} \leq \frac{D_0^2}{5}. \tag{C.82}$$

Combine the above claims, with probability no less than $\mathbb{P}(E_{t,1}) - 2\cdot\dfrac{\delta}{4T}$, we have $|①+②+③+④| \leq \dfrac{4}{5}D_0^2$. By (C.58), these implies

$$\frac{\eta}{2}\sum_{j=0}^{t}(f(\overline{z}_j) - f_*) + \|\overline{z}_{t+1} - x_*\|^2 \leq D_0^2 + 16\left[\eta LKA + \frac{1}{M}\right]\eta^2\sigma^2(t+1) + \frac{4}{5}D_0^2 \tag{C.83}$$

$$\leq 2D_0^2.$$

Therefore, we conclude that $\mathbb{P}(E_{t+1}) \geq \mathbb{P}(E_{t,1}) - \dfrac{\delta}{2T}$.

$\square$

**Lemma C.7.** *Let* $\Lambda_t := \sum\limits_{j=0}^{t-1} a_{t,j}\|\overline{x}_j - \overline{x}_{j+1}\|^2$, *where* $a_{t,j} := \beta_1^{t-j-1}(t - j + \dfrac{\beta_1}{1-\beta_1})$. *Under the conditions in Lemma C.2, then the following holds:*

$$\Lambda_t \leq \left(1 - \frac{(1-\beta_1)^2}{2}\right)\Lambda_{t-1} + \frac{32\eta^2}{1-\beta_1}\|\nabla f(\overline{z}_t)\|^2$$

$$+ 4\eta^2\sum_{j=0}^{t-1}\beta_1^{t-j-1}\left[2L^2\mathbb{E}_m[\|x_j^m - \overline{x}_j\|^2] + \|\mathbb{E}_m[\widehat{g_j^m} - \nabla f(x_j^m)]\|^2\right]. \tag{C.84}$$

$$\|\overline{z}_t - \overline{x}_t\|^2 \leq \left(\frac{\eta\beta_1}{1-\beta_1}\right)^2\left[16L^2\Lambda_{t-1} + 32\|\nabla f(\overline{z}_t)\|^2\right]$$

$$+ \frac{4(\eta\beta_1)^2}{1-\beta_1}\sum_{j=0}^{t-1}\beta_1^{t-j-1}\left[2L^2\mathbb{E}_m[\|x_j^m - \overline{x}_j\|^2] + \|\mathbb{E}_m[\widehat{g_j^m} - \nabla f(x_j^m)]\|^2\right]. \tag{C.85}$$

*Proof.* By definition, $\|\overline{z}_t - \overline{x}_t\|^2 = \left(\dfrac{\beta_1}{1-\beta_1}\right)^2\|\overline{x}_t - \overline{x}_{t-1}\|^2$ and

$$\|\overline{x}_t - \overline{x}_{t-1}\|^2 = \eta^2\|\overline{u}_{t-1}\|^2$$

$$= \eta^2\left\|(1-\beta_1)\sum_{j=0}^{t-1}\beta_1^{t-j-1}\mathbb{E}_m[\widehat{g_j^m}]\right\|^2$$

$$\leq 2\eta^2\left[\left\|(1-\beta_1)\sum_{j=0}^{t-1}\beta_1^{t-j-1}\mathbb{E}_m[\nabla f(x_j^m)]\right\|^2 + \left\|(1-\beta_1)\sum_{j=0}^{t-1}\beta_1^{t-j-1}\mathbb{E}_m[\widehat{g_j^m} - \nabla f(x_j^m)]\right\|^2\right]$$

$$\leq 4\eta^2\left\|(1-\beta_1)\sum_{j=0}^{t-1}\beta_1^{t-j-1}\nabla f(\overline{x}_j)\right\|^2$$

$$+ 2\eta^2(1-\beta_1)\sum_{j=0}^{t-1}\beta_1^{t-j-1}\left[2L^2\mathbb{E}_m[\|x_j^m - \overline{x}_j\|^2] + \|\mathbb{E}_m[\widehat{g_j^m} - \nabla f(x_j^m)]\|^2\right]. \tag{C.86}$$

Note that

$$
\left\|(1-\beta_1)\sum_{j=0}^{t-1}\beta_1^{t-j-1}\nabla f(\overline{x}_j)\right\|^2 \leq 2\left\|(1-\beta_1)\sum_{j=0}^{t-1}\beta_1^{t-j-1}[\nabla f(\overline{x}_j)-\nabla f(\overline{x}_t)]\right\|^2 + 2\|\nabla f(\overline{x}_t)\|^2
$$

$$
\leq 2(1-\beta_1)\sum_{j=0}^{t-1}\beta_1^{t-j-1}L^2\|\overline{x}_j-\overline{x}_t\|^2 + 2\|\nabla f(\overline{x}_t)\|^2
$$

$$
\leq 2(1-\beta_1)\sum_{j=0}^{t-1}\beta_1^{t-j-1}L^2\cdot(t-j)\sum_{i=j}^{t-1}[\|\overline{x}_i-\overline{x}_{i+1}\|^2] + 2\|\nabla f(\overline{x}_t)\|^2
$$

$$
\leq 2L^2\sum_{j=0}^{t-1}a_{t,j}\|\overline{x}_j-\overline{x}_{j+1}\|^2 + 4\|\nabla f(\overline{z}_t)\|^2 + 4L^2\|\overline{x}_t-\overline{z}_t\|^2
$$

$$
\leq 2L^2\sum_{j=0}^{t-2}a_{t-1,j}\|\overline{x}_j-\overline{x}_{j+1}\|^2 + 4\|\nabla f(\overline{z}_t)\|^2 + \frac{4L^2}{(1-\beta_1)^2}\|\overline{x}_t-\overline{x}_{t-1}\|^2
$$

(C.87)

Here $a_{t,j} = \beta_1^{t-j-1}(t-j+\frac{\beta_1}{1-\beta_1})$. For $j \leq t-2$, we have $a_{t,j} \leq \beta_1(2-\beta_1)a_{t-1,j}$. Since $\Lambda_t = \sum_{j=0}^{t-1}a_{t,j}\|\overline{x}_j-\overline{x}_{j+1}\|^2$, we can conclude that

$$
\|\overline{x}_t-\overline{x}_{t-1}\|^2 \leq 16\eta^2 L^2\Lambda_{t-1} + 32\eta^2\|\nabla f(\overline{z}_t)\|^2
$$
$$
+ 4\eta^2(1-\beta_1)\sum_{j=0}^{t-1}\beta_1^{t-j-1}\left[2L^2\mathbb{E}_m[\|x_j^m-\overline{x}_j\|^2] + \|\mathbb{E}_m[\widehat{g_j^m}-\nabla f(x_j^m)]\|^2\right],
$$

(C.88)

which implies (C.85). We complete the proof by plugging the above inequality in

$$
\Lambda_t \leq \beta_1(2-\beta_1)\Lambda_{t-1} + \frac{1}{1-\beta_1}\|\overline{x}_t-\overline{x}_{t-1}\|^2. \tag{C.89}
$$

$\square$

## C.5 FURTHER DISCUSSION

**Coordinate-wise clipping and global clipping.** Lemma B.2 can be easily extended to $\mathbb{R}^d$, similar to Sadiev et al. (2023, Lemma 5.1). Therefore, our results can be easily generalized to global clipping operator $\mathbf{clip}_g(X, \rho_g) := \min\left\{1, \frac{\rho_g}{\|X\|}\right\}X$ with threshold $\rho_g := \rho\sqrt{d}$. We omit the details in this paper. Readers may also wonder why our Theorem C.4 and Theorem C.5 depend on **poly**$(d)$. However, if we assume $\|\boldsymbol{\sigma}\|_{2\alpha}d^{\frac{1}{2}-\frac{1}{2\alpha}} = \mathcal{O}(\sigma)$, both of which are of order $\mathcal{O}(d^{\frac{1}{2}})$, then our convergence guarantee will not depend on **poly**$(d)$ explicitly. Zhang et al. (2020, Corollary 7) claims that coordinate-wise clipping has better dependence on dimension $d$. But they simply upper bound $\mathbb{E}_{\xi\sim\mathcal{D}}\|\nabla F(x,\xi)\|^\alpha$ by $d^{\alpha/2}\mathbb{E}_{\xi\sim\mathcal{D}}\|\nabla F(x,\xi)\|_\alpha^\alpha$, which is too pessimistic. In fact, if we assume $\mathbb{E}_{\xi\sim\mathcal{D}}\|\nabla F(x,\xi)\|^\alpha = \mathcal{O}(d^{\alpha/2-1}\mathbb{E}_{\xi\sim\mathcal{D}}\|\nabla F(x,\xi)\|_\alpha^\alpha)$, both of which are of order $\mathcal{O}(d^{\frac{\alpha}{2}})$, then there is still no difference between coordinate-wise clipping and global clipping in their setting.

**Prior works on distributed SGDM with local updates.** There are many works on *Local* SGDM in distributed setting. Liu et al. (2020a) studies *Local* SGDM in convex setting and rely on some strong assumptions to show convergence. Xu et al. (2021) analyze *Local* SGDM with bounded gradient assumption and the use a global momentum parameter during local iterations. Yu et al. (2019) considers non-convex *Local* SGDM but is only able to prove linear speedup. Wang et al. (2019); Cheng et al. (2023) also study non-convex problem and use momentum to handle heterogeneity in federated learning. All these works fail to show the benefits of local iterations compared to minibatch baseline.

# D PROOF OF LOCAL ADAM

## D.1 OVERVIEW AND MAIN THEOREM

For any integer $0 \le t \le T-1$, we define $r(t), k(t) \in \mathbb{N}$ such that $t = r(t)K + k(t)$ and $k(t) \le K-1$. We omit the dependence on $t$ and let $r = r(t), k = k(t)$ through out the proof if not causing confusion. Define $x_t^m := x_{r,k}^m, g_t^m := g_{r,k}^m, \widehat{g_t^m} := \widehat{g_{r,k}^m}, u_t^m = u_{r,k}^m$. Then Algorithm 2 is equivalent to the following update rule:

$$u_t^m = \begin{cases} \beta_1 u_{t-1}^m + (1-\beta_1)\widehat{g_t^m} & \text{if } t \bmod K \not\equiv 0, \\ \beta_1 \bar{u}_{t-1} + (1-\beta_1)\widehat{g_t^m} & \text{otherwise,} \end{cases} \tag{D.1}$$

$$v_t^m = \begin{cases} \beta_2 v_{t-1}^m + (1-\beta_2)\widehat{g_t^m}^2 & \text{if } t \bmod K \not\equiv 0, \\ \beta_2 \bar{v}_{t-1} + (1-\beta_2)\widehat{g_t^m}^2 & \text{otherwise,} \end{cases} \tag{D.2}$$

$$x_{t+1}^m = \begin{cases} x_t^m - \eta(H_t^m)^{-1}u_t^m & \text{if } t \bmod K \not\equiv -1, \\ \bar{x}_t - \eta \mathbb{E}_m[(H_t^m)^{-1}u_t^m] & \text{otherwise.} \end{cases} \tag{D.3}$$

Define an auxiliary sequence $\{z_t^m\}$ as:

$$z_{t+1}^m = \begin{cases} \dfrac{1}{1-\beta_1}x_{t+1}^m - \dfrac{\beta_1}{1-\beta_1}x_t^m & \text{if } t \bmod K \not\equiv -1, \\ \dfrac{1}{1-\beta_1}x_{t+1}^m - \dfrac{\beta_1}{1-\beta_1}\bar{x}_t & \text{otherwise.} \end{cases} \tag{D.4}$$

Let

$$e_t^m := \frac{\beta_1}{1-\beta_1}(I_d - H_t^m(H_{t-1}^m)^{-1})u_{t-1}^m. \tag{D.5}$$

Then the definition of $\{z_t^m\}$ implies

$$\begin{aligned} z_{t+1}^m - z_t^m &= -\frac{\eta(H_t^m)^{-1}u_t^m}{1-\beta_1} + \frac{\eta\beta_1(H_{t-1}^m)^{-1}u_{t-1}^m}{1-\beta_1} \\ &= -\frac{\eta\beta_1}{1-\beta_1}[(H_t^m)^{-1} - (H_{t-1}^m)^{-1}]u_{t-1}^m - \eta(H_t^m)^{-1}\widehat{g_t^m} \\ &=: -\eta(H_t^m)^{-1}(\widehat{g_t^m} + e_t^m). \end{aligned} \tag{D.6}$$

Finally, let $y_t := \arg\min_y f(y) + \frac{1}{2\gamma}\|y - \bar{z}_t\|_{H_{r(t)}}^2$.

Define probabilistic events (see (D.15) for definition of some parameters)

$$\mathcal{A}_{t,1} := \left\{ \beta_2^{K/2} \preceq H_{r(t)}^{-1}H_t^m \preceq 1 + (1-\beta_2)B \text{ and for all } m \in [M] \right\}, \tag{D.7}$$

$$\mathcal{A}_{t,2} := \left\{ \|H_{r(t)}((H_t^m)^{-1} - (H_t^n)^{-1})\| \le (1-\beta_2)B_1 \text{ for all } m, n \in [M] \right\}, \tag{D.8}$$

$$\mathcal{A}_{t,3} := \left\{ \|z_{t+1}^m - z_{t+1}^n\|_{H_r}^2 \le \frac{\eta^2\sigma^2}{\lambda}KA, \sum_{j=rK}^t \|\widehat{g_j^m}\|^2 \le \frac{(1-\beta_1)^2\sigma^2 A}{2^{12}(1-\beta_2)^2 B_1^2} \text{ for all } m, n \in [M] \right\}, \tag{D.9}$$

$$\mathcal{A}_{t,4} := \left\{ f_\gamma^{H_{r(t+1)}}(\bar{z}_{t+1}) - \min f_\gamma^\lambda + \frac{\eta}{12}\sum_{j=0}^t \|\nabla f_\gamma^{H_{r(j)}}(\bar{z}_j)\|_{H_{r(j)}^{-1}}^2 \le 2\Delta \right\}. \tag{D.10}$$

Here $\Delta := f_\gamma^\lambda(x_0) - \min f_\gamma^\lambda$. Besides, let

$$E_t := \{\mathcal{A}_{j,i} \text{ holds for all } j \le t-1, i \in \{1, 2, 3, 4\}\}, \tag{D.11}$$

$$E_{t,1} := E_t \cap \mathcal{A}_{t,1}, E_{t,2} := E_{t,1} \cap \mathcal{A}_{t,2}, E_{t,3} := E_{t,2} \cap \mathcal{A}_{t,3}. \tag{D.12}$$

**Theorem D.1.** *For $L/\lambda \geq \gamma^{-1} \geq 2\tau/\lambda$, let Assumption 1, 2, 3, 5 hold for $\Omega = \mathbf{conv}(\mathbf{B}_{R_0}(\Omega_0))$, where $\Omega_0 := \{f_\gamma^\lambda(x) - \min f_\gamma^\lambda \leq 2\Delta\}$, $\Delta = f_\gamma^\lambda(x_0) - \min f_\gamma^\lambda$ and $R_0 = \sqrt{\dfrac{\Delta\gamma}{160\lambda}}$. Further assume that for any $x \in \Omega$, $\|\nabla f(x)\| \leq G$, $\|\nabla f(x)\|_\infty \leq G_\infty$, and*

$$1 - \beta_2 \lesssim \min\left\{\frac{1-\beta_1}{K^{1/2}B_1}\frac{(1-\beta_1)\sigma\sqrt{A}}{K^{1/2}B_1 G}, \frac{\eta}{\gamma B}, \frac{1-\beta_1}{K^{1/2}B}, \frac{1}{K}\right\}. \tag{D.13}$$

*If $\eta = \dfrac{24\lambda\Delta}{\varepsilon T}$, then with probability no less than $1 - \delta$, Local Adam yields*
$$\frac{\lambda}{KR}\sum_{r=0}^{R-1}\sum_{k=0}^{K-1}\|\nabla f_\gamma^{H_r}(\overline{z}_{r,k})\|_{H_r^{-1}}^2 \leq \varepsilon \text{ if}$$

$$T \gtrsim \frac{\lambda\Delta\sigma^2}{\gamma M\varepsilon^2}\log^{\frac{1}{2}}\frac{T}{\delta} + \frac{\Delta}{\varepsilon}\cdot\sqrt{\frac{L^2\sigma^2 KA}{\min\{\varepsilon,\sigma_\infty^2/G_\infty\}}} + \frac{L\Delta}{(1-\beta_1)^2\varepsilon} + \frac{K\tau\Delta}{\varepsilon} + \frac{\sqrt{L\Delta\rho^2 d\log\frac{T}{\delta}}}{(\sqrt{\beta_2}-\beta_1)\varepsilon}. \tag{D.14}$$

*Here*

$$\rho \geq \max\left\{\left(\frac{2^6\|2\boldsymbol{\sigma}\|_{2\alpha}^{2\alpha}}{\varepsilon}\right)^{\frac{1}{2(\alpha-1)}}, 3\sigma_\infty, 2G_\infty\right\},$$
$$B := \max\left\{\frac{6K(G_\infty^2 + \sigma_\infty^2)}{\lambda^2}, \frac{16\rho^2}{\lambda^2}\log\frac{dMT}{\delta}, 2^6\frac{\sqrt{K}(G_\infty + \sigma_\infty)\sigma_\infty}{\lambda^2}\log^{1/2}\frac{dMT}{\delta}\right\},$$
$$B_1 := \max\left\{\frac{16K\sigma_\infty^2}{\lambda^2}, \frac{16\rho^2}{\lambda^2}\log\frac{dMT}{\delta}, 2^6\frac{\sqrt{K}(G_\infty + \sigma_\infty)\sigma_\infty}{\lambda^2}\log^{1/2}\frac{dMT}{\delta}\right\},$$
$$A := \max\left\{\frac{2^{20}\rho^2 d}{K\sigma^2}\log\frac{MT}{\delta}, 2^{20}\log^2\frac{MT}{\delta}, \frac{2^8 K\|2\boldsymbol{\sigma}\|_{2\alpha}^{2\alpha}}{\sigma^2\rho^{2(\alpha-1)}}\right\}. \tag{D.15}$$

*Proof.* We prove by induction that $\mathbb{P}(E_t) \geq 1 - \dfrac{t\delta}{T}$ for $t = 0, \cdots, T$.

When $t = 0$, this is trivial. Assume that the statement is true for some $t \leq T - 1$. We aim to prove that $\mathbb{P}(E_{t+1}) \geq 1 - \dfrac{(t+1)\delta}{T}$. By Lemma D.8, D.9, D.10, D.11, we have

$$\mathbb{P}(E_{t+1}) \geq \mathbb{P}(E_t) - 4\cdot\frac{\delta}{4T} \geq 1 - \frac{(t+1)\delta}{T}. \tag{D.16}$$

Therefore by induction rule, $\mathbb{P}(E_T) \geq 1 - \delta$ and this implies

$$\frac{\lambda}{T}\sum_{t=0}^{T-1}\|\nabla f_\gamma^{H_{r(t)}}(\overline{z}_t)\|_{H_{r(t)}^{-1}}^2 \leq \frac{24\Delta\lambda}{\eta T} = \varepsilon. \tag{D.17}$$

Now we verify the conditions in all the lemmas. In Lemma D.7,

$$\frac{\eta}{\lambda} \lesssim \sqrt{\frac{\Delta\gamma}{\lambda\sigma^2 KA}} \Longleftarrow T \gtrsim \frac{\sigma}{\varepsilon}\sqrt{L\Delta KA}. \tag{D.18}$$

In Lemma D.9,

$$\frac{\eta}{\lambda} \lesssim \frac{\sigma_\infty^2}{G_\infty L\sigma\sqrt{KA}} \Longleftarrow T \gtrsim \frac{\Delta}{\varepsilon}\cdot\sqrt{\frac{L^2\sigma^2 KA}{\sigma_\infty^2/G_\infty}}. \tag{D.19}$$

In Lemma D.10,

$$\frac{\eta}{\lambda} \lesssim \min\left\{\frac{1}{K\tau}, \frac{(1-\beta_1)^2}{L}\right\} \Longleftarrow T \gtrsim \frac{L\Delta}{(1-\beta_1)^2\varepsilon} + \frac{K\tau\Delta}{\varepsilon}. \tag{D.20}$$

In Lemma D.11, by noticing that $\frac{24\Delta\lambda}{\eta T} = \varepsilon$, (D.113) is equivalent to $\rho \gtrsim \left( \frac{\|2\boldsymbol{\sigma}\|_{2\alpha}^{2\alpha}}{\varepsilon} \right)^{\frac{1}{2(\alpha-1)}}$ and

$$\frac{\eta}{\lambda} \lesssim \min \left\{ \frac{(1-\beta_1)^2}{L}, \frac{M\gamma\varepsilon}{\lambda\sigma^2 \log^{1/2} \frac{T}{\delta}}, \left( \frac{L^2\sigma^2 KA}{\varepsilon} \right)^{-1/2}, \frac{M\Delta}{\sigma^2 \log \frac{T}{\delta}}, \sqrt{\frac{\gamma\Delta}{\lambda\rho^2 d \log \frac{T}{\delta}}}, \frac{\sqrt{T}\varepsilon(\sqrt{\beta_2}-\beta_1)}{L\rho\sqrt{d} \log^{1/2} \frac{T}{\delta}} \right\},$$
(D.21)

which can be ensured as long as

$$T \gtrsim \max \left\{ \frac{L\Delta}{(1-\beta_1)^2\varepsilon}, \frac{\lambda\Delta\sigma^2}{\gamma M\varepsilon^2} \log^{\frac{1}{2}} \frac{T}{\delta}, \frac{\Delta}{\varepsilon} \cdot \sqrt{\frac{L^2\sigma^2 KA}{\varepsilon}}, \frac{\sqrt{L\Delta\rho^2 d \log \frac{T}{\delta}}}{(\sqrt{\beta_2}-\beta_1)\varepsilon} \right\}.$$
(D.22)

Here we use the fact that $\gamma \geq \frac{\lambda}{L}$. Therefore we can conclude that all the lemmas hold if

$$T \gtrsim \frac{\lambda\Delta\sigma^2}{\gamma M\varepsilon^2} \log^{\frac{1}{2}} \frac{T}{\delta} + \frac{\Delta}{\varepsilon} \cdot \sqrt{\frac{L^2\sigma^2 KA}{\min\{\varepsilon, \sigma_\infty^2/G_\infty\}}} + \frac{L\Delta}{(1-\beta_1)^2\varepsilon} + \frac{K\tau\Delta}{\varepsilon} + \frac{\sqrt{L\Delta\rho^2 d \log \frac{T}{\delta}}}{\varepsilon}.$$
(D.23)

Finally, we verify the upper bound of $1-\beta_2$ in Lemma D.9, D.10 and D.11 as:

$$1-\beta_2 \lesssim \min \left\{ \frac{1-\beta_1}{K^{1/2}B_1} \frac{(1-\beta_1)\sigma\sqrt{A}}{K^{1/2}B_1 G}, \frac{\eta}{\gamma B}, \frac{1-\beta_1}{K^{1/2}B}, \frac{1}{K} \right\}.$$
(D.24)

$\square$

**Theorem D.2.** *Under the conditions of Theorem D.1, assume $1-\beta_1 = \Omega(1)$ and*

$$1-\beta_2 = \tilde{\mathcal{O}} \left( \frac{1}{K^{3/2}R^{1/2}} \right), \quad \left( \frac{\|\boldsymbol{\sigma}\|_{2\alpha}^{2\alpha}}{\varepsilon} \right)^{\frac{1}{2(\alpha-1)}} \gtrsim G_\infty \vee \sigma_\infty, \varepsilon \lesssim \frac{\sigma_\infty^2}{G_\infty},$$
$$K \gtrsim \log \frac{MT}{\delta} \left( \frac{\|\boldsymbol{\sigma}\|_{2\alpha} d^{\frac{1}{2}-\frac{1}{2\alpha}}}{\sigma} \right)^{\frac{2\alpha}{\alpha-2}}.$$
(D.25)

*Then with probability no less than $1 - \delta$, Local Adam with optimal $\eta, \rho$ yields $\frac{\lambda}{KR} \sum_{r=0}^{R-1} \sum_{k=0}^{K-1} \|\nabla f_\gamma^{H_r}(\overline{z}_{r,k})\|_{H_r^{-1}}^2 \leq \varepsilon$ if*

$$T \gtrsim \frac{\lambda\Delta\sigma^2}{\gamma M\varepsilon^2} \log^{\frac{1}{2}} \frac{T}{\delta} + \frac{L\Delta}{\varepsilon^{\frac{3}{2}}} \cdot \sqrt{\sigma^2 K \log \frac{MT}{\delta}} + \frac{(L+K\tau)\Delta}{\varepsilon} + \frac{L\Delta}{\varepsilon^{\frac{3}{2}}} \left( \frac{\|\boldsymbol{\sigma}\|_{2\alpha}^{2\alpha}}{\varepsilon} \right)^{\frac{1}{2(\alpha-1)}} d^{\frac{1}{2}} \log \frac{MT}{\delta}.$$
(D.26)

*And equivalently,*

$$\frac{\lambda}{KR} \sum_{r=0}^{R-1} \sum_{k=0}^{K-1} \|\nabla f_\gamma^{H_r}(\overline{z}_{r,k})\|_{H_r^{-1}}^2 \lesssim \frac{\tau\Delta}{R} + \frac{L\Delta}{KR} + \sqrt{\frac{\lambda\Delta\sigma^2}{\gamma MKR}} \log^{\frac{1}{4}} \frac{KR}{\delta}$$

$$+ \frac{(L\Delta\sigma)^{\frac{2}{3}}}{K^{\frac{1}{3}}R^{\frac{2}{3}}} \log^{\frac{1}{3}} \frac{MKR}{\delta} + \left( \|\boldsymbol{\sigma}\|_{2\alpha} d^{\frac{1}{2}-\frac{1}{2\alpha}} \right)^{\frac{2\alpha}{3\alpha-2}} \left( \frac{L\Delta \log \frac{MKR}{\delta}}{KR} \right)^{\frac{2(\alpha-1)}{3\alpha-2}}.$$
(D.27)

*Proof.* Plug the definition of $A$ in (D.14),

$$
\begin{aligned}
T &\gtrsim \frac{\lambda \Delta \sigma^2}{\gamma M \varepsilon^2} \log^{\frac{1}{2}} \frac{T}{\delta} + \frac{\Delta}{\varepsilon} \cdot \sqrt{\frac{L^2 \sigma^2 K \log \frac{MT}{\delta}}{\varepsilon}} + \frac{(L+K\tau)\Delta}{\varepsilon} + \frac{\sqrt{L\Delta \rho^2 d \log \frac{T}{\delta}}}{\varepsilon} \\
&\quad + \frac{\Delta}{\varepsilon} \cdot \sqrt{\frac{L^2 K}{\varepsilon}} \sqrt{\frac{d \log^2 \frac{MT}{\delta}}{K} \rho^2 + K \|\boldsymbol{\sigma}\|_{2\alpha}^{2\alpha} \cdot \rho^{2(1-\alpha)}} \\
&\asymp \frac{\lambda \Delta \sigma^2}{\gamma M \varepsilon^2} \log^{\frac{1}{2}} \frac{T}{\delta} + \frac{\Delta}{\varepsilon} \cdot \sqrt{\frac{L^2 \sigma^2 K \log \frac{MT}{\delta}}{\varepsilon}} + \frac{(L+K\tau)\Delta}{\varepsilon} \\
&\quad + \frac{\Delta}{\varepsilon} \cdot \sqrt{\frac{L^2 K}{\varepsilon}} \sqrt{\frac{d \log^2 \frac{MT}{\delta}}{K} \rho^2 + K \|\boldsymbol{\sigma}\|_{2\alpha}^{2\alpha} \cdot \rho^{2(1-\alpha)}}.
\end{aligned}
\tag{D.28}
$$

Hence the optimal $\rho$ is given by

$$
\rho \asymp \max \left\{ \|\boldsymbol{\sigma}\|_{2\alpha} \left( \frac{K}{\sqrt{d} \log \frac{MT}{\delta}} \right)^{1/\alpha}, \left( \frac{\|\boldsymbol{\sigma}\|_{2\alpha}^{2\alpha}}{\varepsilon} \right)^{\frac{1}{2(\alpha-1)}}, \sigma_\infty, G_\infty \right\}.
\tag{D.29}
$$

Note that $\left( \frac{\|\boldsymbol{\sigma}\|_{2\alpha}^{2\alpha}}{\varepsilon} \right)^{\frac{1}{2(\alpha-1)}} \gtrsim G_\infty \vee \sigma_\infty$ and this implies

$$
\begin{aligned}
T &\gtrsim \frac{\lambda \Delta \sigma^2}{\gamma M \varepsilon^2} \log^{\frac{1}{2}} \frac{T}{\delta} + \frac{\Delta}{\varepsilon} \cdot \sqrt{\frac{L^2 \sigma^2 K \log \frac{MT}{\delta}}{\varepsilon}} + \frac{(L+K\tau)\Delta}{\varepsilon} \\
&\quad + \frac{L\Delta}{\varepsilon^{\frac{3}{2}}} \left[ \|\boldsymbol{\sigma}\|_{2\alpha} d^{\frac{1}{2}-\frac{1}{2\alpha}} K^{\frac{1}{\alpha}} \log^{1-\frac{1}{\alpha}} \frac{MT}{\delta} + \left( \frac{\|\boldsymbol{\sigma}\|_{2\alpha}^{2\alpha}}{\varepsilon} \right)^{\frac{1}{2(\alpha-1)}} d^{\frac{1}{2}} \log \frac{MT}{\delta} \right] \\
&\asymp \frac{\lambda \Delta \sigma^2}{\gamma M \varepsilon^2} \log^{\frac{1}{2}} \frac{T}{\delta} + \frac{L\Delta}{\varepsilon^{\frac{3}{2}}} \cdot \sqrt{\sigma^2 K \log \frac{MT}{\delta}} + \frac{(L+K\tau)\Delta}{\varepsilon} + \frac{L\Delta}{\varepsilon^{\frac{3}{2}}} \left( \frac{\|\boldsymbol{\sigma}\|_{2\alpha}^{2\alpha}}{\varepsilon} \right)^{\frac{1}{2(\alpha-1)}} d^{\frac{1}{2}} \log \frac{MT}{\delta}.
\end{aligned}
\tag{D.30}
$$

In the last equation we use $K \gtrsim \log \frac{MT}{\delta} \left( \frac{\|\boldsymbol{\sigma}\|_{2\alpha} d^{\frac{1}{2}-\frac{1}{2\alpha}}}{\sigma} \right)^{\frac{2\alpha}{\alpha-2}}$. Solve $\varepsilon$ and we get the upper

bound of $\frac{\lambda}{KR} \sum_{r=0}^{R-1} \sum_{k=0}^{K-1} \|\nabla f_\gamma^{H_r}(\overline{z}_{r,k})\|_{H_r^{-1}}^2$.

Further note that $A = \tilde{\mathcal{O}}(1), B = \tilde{\mathcal{O}}(K), B_1 = \tilde{\mathcal{O}}(K), \eta = \tilde{\mathcal{O}}(1/\sqrt{T})$ and we can get the upper bound of $1 - \beta_2$ as:

$$
1 - \beta_2 = \tilde{\mathcal{O}} \left( \frac{1}{K^{3/2} R^{1/2}} \right).
\tag{D.31}
$$

This completes the proof. $\qquad\square$

**Theorem D.3** (Complete version of Theorem 3). *Under the conditions of Theorem D.2, let $\gamma = \frac{\lambda}{L}$ and thus $\Omega_0 \subset \{x : f(x) - f_* \leq 4(f(x_0) - f_*)\}, \Delta \asymp f(x_0) - f_*$. Then with probability no less than $1 - \delta$, Local Adam with optimal $\eta, \rho$ yields $\frac{\lambda}{KR} \sum_{r=0}^{R-1} \sum_{k=0}^{K-1} \|\nabla f(\overline{z}_{r,k})\|_{H_r^{-1}}^2 \leq \varepsilon$ if*

$$
T \gtrsim \frac{L\Delta \sigma^2}{M \varepsilon^2} \log^{\frac{1}{2}} \frac{T}{\delta} + \frac{L\Delta}{\varepsilon^{\frac{3}{2}}} \cdot \sqrt{\sigma^2 K \log \frac{MT}{\delta}} + \frac{(L+K\tau)\Delta}{\varepsilon} + \frac{L\Delta}{\varepsilon^{\frac{3}{2}}} \left( \frac{\|\boldsymbol{\sigma}\|_{2\alpha}^{2\alpha}}{\varepsilon} \right)^{\frac{1}{2(\alpha-1)}} d^{\frac{1}{2}} \log \frac{MT}{\delta}.
\tag{D.32}
$$

*And equivalently,*

$$\frac{\lambda}{KR}\sum_{r=0}^{R-1}\sum_{k=0}^{K-1}\|\nabla f(\overline{z}_{r,k})\|_{H_r^{-1}}^2 \lesssim \frac{\tau\Delta}{R} + \frac{L\Delta}{KR} + \sqrt{\frac{L\Delta\sigma^2}{MKR}}\log^{\frac{1}{4}}\frac{KR}{\delta}$$

$$+ \frac{(L\Delta\sigma)^{\frac{2}{3}}}{K^{\frac{1}{3}}R^{\frac{2}{3}}}\log^{\frac{1}{3}}\frac{MKR}{\delta} + \left(\|\boldsymbol{\sigma}\|_{2\alpha}d^{\frac{1}{2}-\frac{1}{2\alpha}}\right)^{\frac{2\alpha}{3\alpha-2}}\left(\frac{L\Delta\log\frac{MKR}{\delta}}{KR}\right)^{\frac{2(\alpha-1)}{3\alpha-2}}.$$

$$(D.33)$$

*Further, if* $1-\beta_2 \lesssim \dfrac{G_\infty^2+\sigma_\infty^2}{\rho^2\log\frac{dR}{\delta}}$, *where $\rho$ is defineded in (D.29), then with probability no less than* $1-2\delta$,

$$\frac{1}{KR}\sum_{r=0}^{R-1}\sum_{k=0}^{K-1}\|\nabla f(\overline{z}_{r,k})\|^2 \lesssim \left(1+\frac{G_\infty+\sigma_\infty}{\lambda}\right)\left[\frac{\tau\Delta}{R} + \frac{L\Delta}{KR} + \sqrt{\frac{L\Delta\sigma^2}{MKR}}\log^{\frac{1}{4}}\frac{KR}{\delta} + \frac{(L\Delta\sigma)^{\frac{2}{3}}}{K^{\frac{1}{3}}R^{\frac{2}{3}}}\log^{\frac{1}{3}}\frac{MKR}{\delta}\right.$$

$$\left.+ \left(\|\boldsymbol{\sigma}\|_{2\alpha}d^{\frac{1}{2}-\frac{1}{2\alpha}}\right)^{\frac{2\alpha}{3\alpha-2}}\left(\frac{L\Delta\log\frac{MKR}{\delta}}{KR}\right)^{\frac{2(\alpha-1)}{3\alpha-2}}\right].$$

$$(D.34)$$

*Proof.* By Lemma D.6, we have $\Omega_0 \subset \{x : f(x) - f_* \leq 4(f(x_0) - f_*)\}, \Delta \asymp f(x_0) - f_*$. By Lemma D.4, we have $\|\nabla f(\overline{z}_{r,k})\|_{H_r^{-1}} \leq 2\|\nabla f_\gamma^{H_r}(\overline{z}_{r,k})\|_{H_r^{-1}}$. Therefore, the bound for $T$ in Theorem D.2 will reduce to (D.32). Solve $\varepsilon$ and we get the upper bound of $\frac{\lambda}{KR}\sum_{r=0}^{R-1}\sum_{k=0}^{K-1}\|\nabla f(\overline{z}_{r,k})\|_{H_r^{-1}}^2$.

Now we turn to bound $\|H_r\|$. Note that $H_{r+1} = \mathbf{diag}(\sqrt{v_{r+1}+\lambda^2})$ and

$$[v_{r+1}]_i = (1-\beta_2)\sum_{j=0}^{rK-1}\beta_2^{rK-j-1}\mathbb{E}_m[\widehat{g_j^m}]_i^2$$

$$= (1-\beta_2)\sum_{j=0}^{rK-1}\beta_2^{rK-j-1}\left(\mathbb{E}_m\left[[\widehat{g_j^m}]_i^2 - \mathbb{E}_j[\widehat{g_j^m}]_i^2\right] + \mathbb{E}_m\mathbb{E}_j[\widehat{g_j^m}]_i^2\right) \quad (D.35)$$

$$\leq (1-\beta_2)\sum_{j=0}^{rK-1}\beta_2^{rK-j-1}\mathbb{E}_m\left[[\widehat{g_j^m}]_i^2 - \mathbb{E}_j[\widehat{g_j^m}]_i^2\right] + \sigma_\infty^2 + 3G_\infty^2,$$

where the last inequality is due to Lemma B.2. Define

$$[\theta_j]_i = \begin{cases} (1-\beta_2)\beta_2^{rK-j-1}\mathbb{E}_m\left[[\widehat{g_j^m}]_i^2 - \mathbb{E}_j[\widehat{g_j^m}]_i^2\right], & \text{if event } E_j \text{ holds,} \\ 0, & \text{otherwise.} \end{cases} \quad (D.36)$$

Further note that

$$|[\theta_j]_i| \leq (1-\beta_2)\rho^2 \overset{def}{=} c, \quad (D.37)$$

$$\text{Var}_j([\theta_j]_i) \leq \frac{(1-\beta_2)^2\beta_2^{2(rK-j-1)}}{M}\mathbb{E}_m\mathbb{E}_j\left[[\widehat{g_j^m}]_i^2 - \mathbb{E}_j[\widehat{g_j^m}]_i^2\right]^2$$

$$\leq \frac{(1-\beta_2)^2\beta_2^{2(rK-j-1)}}{M}\mathbb{E}_m\mathbb{E}_j\left[[\widehat{g_j^m}]_i^2 - [\nabla f(x_j^m)]_i^2\right]^2 \quad (D.38)$$

$$\leq \frac{(1-\beta_2)^2\beta_2^{2(rK-j-1)}}{M}(2\sigma_\infty^4 + 8\sigma_\infty^2 G_\infty^2).$$

Let $b = G_\infty^2 + 3\sigma_\infty^2$, $V = \dfrac{2(1-\beta_2)\sigma_\infty^2(\sigma_\infty^2 + 4G_\infty^2)}{M}$. If $1 - \beta_2 \lesssim \dfrac{G_\infty^2 + \sigma_\infty^2}{\rho^2 \log \frac{dR}{\delta}}$, then by Lemma

B.1, we have $|\sum\limits_{j=0}^{rK-1} [\theta_j]_i| \leq b$ with probability no less than

$$1 - 2\exp\left(-\frac{b^2}{2V + 2cb/3}\right) \geq 1 - \frac{\delta}{dR}, \tag{D.39}$$

which implies $[H_r]_{i,i} \leq \lambda + 2G_\infty + 2\sigma_\infty$. Therefore, we have

$$\mathbb{P}\{E_T \text{ and } \|H_r\| \leq \lambda + 2G_\infty + 2\sigma_\infty \text{ for all } r \leq R\} \geq 1 - 2\delta. \tag{D.40}$$

And thus

$$\frac{1}{KR} \sum_{r=0}^{R-1} \sum_{k=0}^{K-1} \|\nabla f(\overline{z}_{r,k})\|^2 \lesssim \left(1 + \frac{G_\infty + \sigma_\infty}{\lambda}\right) \left[\frac{\tau\Delta}{R} + \frac{L\Delta}{KR} + \sqrt{\frac{L\Delta\sigma^2}{MKR}} \log^{\frac{1}{4}} \frac{T}{\delta} + \frac{(L\Delta\sigma)^{\frac{2}{3}}}{K^{\frac{1}{3}} R^{\frac{2}{3}}} \log^{\frac{1}{3}} \frac{MKR}{\delta}\right.$$

$$\left. + \left(\|\boldsymbol{\sigma}\|_{2\alpha} d^{\frac{1}{2} - \frac{1}{2\alpha}}\right)^{\frac{2\alpha}{3\alpha - 2}} \left(\frac{L\Delta \log \frac{MKR}{\delta}}{KR}\right)^{\frac{2(\alpha-1)}{3\alpha-2}}\right]. \tag{D.41}$$

$\square$

## D.2 PRELIMINARIES

We start with theoretical properties of weakly convex function and Moreau envelop, which are repeatedly used in our proof.

**Lemma D.4.** *Let $z \in \mathbb{R}^d$ and $y = y(z) := \arg\min\limits_x f(x) + \dfrac{1}{2\gamma}\|x - z\|_H^2$ for some $H \succeq \lambda I_d$ and $L/\lambda \geq \gamma^{-1} \geq 2\tau/\lambda$. Then*

$$\nabla f_\gamma^H(z) = \nabla f(y) = \frac{H(z-y)}{\gamma}. \tag{D.42}$$

*If further assume $f_\gamma^H(z) - \min f_\gamma^\lambda \leq 2\Delta$, $0 \leq \eta \leq \dfrac{\lambda}{L}$, then $z, y \in \Omega_0$, and*

$$\|\nabla f(z)\|_{H^{-1}} \leq \frac{2\gamma L}{\lambda} \|\nabla f_\gamma^H(z)\|_{H^{-1}}, \tag{D.43}$$

$$\|H(z-y) - \eta\nabla f(z)\|_{H^{-1}} \leq \gamma\|\nabla f(y)\|_{H^{-1}}. \tag{D.44}$$

$$\|\nabla f_\gamma^H(z)\|_{H^{-1}}^2 \leq \frac{2}{\gamma}(f_\gamma^H(z) - \min f_\gamma^\lambda). \tag{D.45}$$

*Proof.* Since $y$ is the minimizer,

$$0 = \nabla_y \left[f(y) + \frac{1}{2\gamma}\|y - z\|_H^2\right] = \nabla f(y) + \frac{H(y-z)}{\gamma}, \tag{D.46}$$

and note that

$$\nabla f_\gamma^H(z) = \nabla_z \left[f(y(z)) + \frac{1}{2\gamma}\|y(z) - z\|_H^2\right] = \frac{H(z-y)}{\gamma}. \tag{D.47}$$

If $f_\gamma^H(z) - \min f_\gamma^\lambda \leq 2\Delta$, then $f_\gamma^\lambda(z) \leq f_\gamma^H(z)$ and

$$f_\gamma^\lambda(y) \leq f_\gamma^H(y) \leq f(y) \leq f_\gamma^H(z) \leq f(z), \tag{D.48}$$

which implies $y, z \in \Omega_0$.

By mean value theorem, there exists a symmetric matrix $-\tau I_d \preceq H_g \preceq LI_d$, such that

$$\nabla f(z) - \nabla f(y) = H_g(z-y) = \gamma H_g H^{-1} \nabla f(y). \tag{D.49}$$

Hence,

$$\|\nabla f(z) - \nabla f(y)\|_{H^{-1}} \leq \gamma \|H^{-1}\nabla f(y)\|_{H_g H^{-1} H_g} \leq \frac{\gamma L}{\lambda}\|\nabla f_\gamma^H(z)\|_{H^{-1}}. \tag{D.50}$$

$$\|\nabla f(z)\|_{H^{-1}} \leq (1 + \frac{\gamma L}{\lambda})\|\nabla f_\gamma^H(z)\|_{H^{-1}} \leq \frac{2\gamma L}{\lambda}\|\nabla f_\gamma^H(z)\|_{H^{-1}}. \tag{D.51}$$

Also,

$$H(z - y) - \eta \nabla f(z) = (\gamma I_d - \eta(I_d + \gamma H_g H^{-1}))\nabla f(y) =: \gamma \Lambda \nabla f(y). \tag{D.52}$$

By noticing that

$$-I_d \preceq H^{-1/2}\Lambda H^{1/2} = I_d - \eta\gamma^{-1} - \eta H^{-1/2}H_g H^{-1/2} \preceq I_d, \tag{D.53}$$

we have $\|H(z - y) - \eta \nabla f(z)\|_{H^{-1}} \leq \gamma \|\nabla f(y)\|_{H^{-1}}$.

Last,

$$\min f_\gamma^\lambda \leq f_\gamma^\lambda(y) \leq f(y) = f_\gamma^H(z) - \frac{1}{2\gamma}\|y - z\|_H^2 = f_\gamma^H(z) - \frac{\gamma}{2}\|\nabla f_\gamma^H(z)\|_{H^{-1}}^2. \tag{D.54}$$

This completes the proof. $\qquad\square$

**Lemma D.5.** *If $x, y \in \Omega$, then*

$$-\langle x - y, \nabla f(x) - \nabla f(y)\rangle + \frac{1}{L}\|\nabla f(x) - \nabla f(y)\|^2 \leq 2\tau\|x - y\|^2. \tag{D.55}$$

*Proof.* By mean value theorem, there exists a symmetric matrix $-\tau I_d \preceq H \preceq L I_d$, such that

$$\nabla f(x) - \nabla f(y) = H(x - y). \tag{D.56}$$

Therefore,

$$\begin{aligned} -\langle x - y, \nabla f(x) - \nabla f(y)\rangle + \frac{1}{L}\|\nabla f(x) - \nabla f(y)\|^2 &= (x - y)^T(-H + \frac{H^2}{L})(x - y) \\ &\leq (\tau + \frac{\tau^2}{L})\|x - y\|^2 \\ &\leq 2\tau\|x - y\|^2. \end{aligned} \tag{D.57}$$

$\qquad\square$

**Lemma D.6.** *If $\gamma = \frac{\lambda}{L}$, then for $z \in \Omega_0$, it holds that $\dfrac{f(z) - f_*}{2} \leq f_{1/L}(z) - f_* \leq f(z) - f_*$.*

*Proof.* By definition of Moreau envelop, the second inequality is trivial. Let $y = \arg\min_x f(x) + \frac{L}{2}\|x - z\|^2$. Note that $x \to f(x) + \frac{L}{2}\|x - z\|^2$ is $2L$-smooth. Then we have

$$f(z) \leq f(y) + \frac{L}{2}\|y - z\|^2 + L\|y - z\|^2 = f_{1/L}(z) + L\|y - z\|^2. \tag{D.58}$$

Furthermore, by Lemma D.4

$$\frac{L}{2}\|y - z\|^2 = \frac{1}{2L}\|\nabla f(y)\|^2 \leq f(y) - f_*. \tag{D.59}$$

Therefore, $f(z) - f_* \leq f_{1/L}(z) - f_* + L\|y - z\|^2 \leq 2(f_{1/L}(z) - f_*)$. $\qquad\square$

Next, we show that event $E_t$ implies all the iterates remain in certain area.

**Lemma D.7.** *If $\dfrac{\eta\sigma}{\lambda}\sqrt{KA} \leq \sqrt{\dfrac{\Delta\gamma}{160\lambda}}$, then event $E_t$ implies that for all $j \leq t, m \in [M]$, we have $\overline{z}_j \in \Omega_0, x_j^m, \overline{x}_j, z_j^m \in \Omega$. And $\|x_j^m - x_j^n\| \leq \dfrac{\eta\sigma}{\lambda}\sqrt{KA}$ for all $m, n$.*

*Proof.* Event $E_t$ implies that for all $j \leq t$,

$$f_\gamma^\lambda(\overline{z}_j) - \min f_\gamma^\lambda \leq 2\Delta, \; \|z_j^m - z_j^n\| \leq \frac{\eta\sigma}{\lambda}\sqrt{KA} \leq \sqrt{\frac{\Delta\gamma}{160\lambda}}. \tag{D.60}$$

Hence $\overline{z}_j \in \Omega_0, \|z_j^m - \overline{z}_j\| \leq \frac{\eta\sigma}{\lambda}\sqrt{KA}$ and $z_j^m \in \mathbf{B}_{R_0}(\Omega_0) \subset \Omega$. Also, notice that $\overline{x}_j \in \mathbf{conv}\{\overline{z}_i\}_{i\leq j} \subset \mathbf{conv}(\Omega_0) \subset \Omega$ and $x_j^m - x_j^n \in \mathbf{conv}\{z_i^m - z_i^n\}_{i\leq j}$. We have

$$\|x_j^m - x_j^n\| \leq \frac{\eta\sigma}{\lambda}\sqrt{KA}, \; \|x_j^m - \overline{x}_j\| \leq \frac{\eta\sigma}{\lambda}\sqrt{KA} \leq \sqrt{\frac{\Delta\gamma}{160\lambda}}. \tag{D.61}$$

Therefore by Lemma B.4, $x_j^m \in \mathbf{B}_{R_0}(\mathbf{conv}(\Omega_0)) = \Omega$. $\qquad\square$

The following lemma shows that the second order momentum $v_t^m$ does not change too much from $v_{r(t)}$ during local training with high probability, which is also repeatedly used in our proof.

**Lemma D.8.** *Let* $B := \max\left\{\dfrac{6K(G_\infty^2 + \sigma_\infty^2)}{\lambda^2}, \dfrac{16\rho^2}{\lambda^2}\log\dfrac{dMT}{\delta}, 2^6\dfrac{\sqrt{K}(G_\infty + \sigma_\infty)\sigma_\infty}{\lambda^2}\log^{1/2}\dfrac{dMT}{\delta}\right\}.$
*If* $\rho \geq \max\{3\sigma_\infty, 2G_\infty\}$, *then the following holds*

$$\mathbb{P}(E_{t,1}) \geq \mathbb{P}(E_t) - \frac{\delta}{4T}. \tag{D.62}$$

*Proof.* Let $t = rK + k$. By the update rule of local Adam, we have

$$v_t^m = \beta_2^{k+1}v_r + (1 - \beta_2)\sum_{j=rK}^{t}\beta_2^{t-j}\widehat{g_j^m} \odot \widehat{g_j^m} \succeq \beta_2^K v_r, \tag{D.63}$$

and hence

$$H_t^m = \mathbf{diag}(\sqrt{v_t^m + \lambda^2}) \succeq \beta_2^{K/2}\mathbf{diag}(\sqrt{v_r + \lambda^2}) = \beta_2^{K/2}H_r. \tag{D.64}$$

For the upper bound, for any index $i \in [d]$, by Lemma B.2,

$$\mathbb{E}_j[\widehat{g_j^m}]_i^2 \leq \sigma_i^2 + [\mathbb{E}_j[\widehat{g_j^m}]_i]^2 \leq \sigma_\infty^2 + 3G_\infty^2. \tag{D.65}$$

Therefore,

$$[v_t^m]_i \leq [v_r]_i + (1 - \beta_2)K(\sigma_\infty^2 + 3G_\infty^2) + (1 - \beta_2)\sum_{j=rK}^{t}\left[[\widehat{g_j^m}]_i^2 - \mathbb{E}_j[\widehat{g_j^m}]_i^2\right]. \tag{D.66}$$

Define

$$[\theta_j^m]_i = \begin{cases} [\widehat{g_j^m}]_i^2 - \mathbb{E}_j[\widehat{g_j^m}]_i^2, & \text{if event } E_j \text{ holds,} \\ 0, & \text{otherwise.} \end{cases} \tag{D.67}$$

Event $E_t$ implies $[\theta_j^m]_i = [\widehat{g_j^m}]_i^2 - \mathbb{E}_j[\widehat{g_j^m}]_i^2$. Further note that $|[\theta_j^m]_i| \leq \rho^2 \overset{def}{=} c$,

$$\begin{aligned} \mathrm{Var}_j([\theta_j^m]_i) &\leq \mathbb{E}_j\left[[\widehat{g_j^m}]_i^2 - [\nabla f(x_j^m)]_i^2\right]^2 \\ &= \mathbb{E}_j\left[[\widehat{g_j^m}]_i - [\nabla f(x_j^m)]_i\right]^2\left[[\widehat{g_j^m}]_i - [\nabla f(x_j^m)]_i + 2[\nabla f(x_j^m)]_i\right]^2 \\ &\overset{\text{AM-GM}}{\leq} 2\mathbb{E}_j\left[[\widehat{g_j^m}]_i - [\nabla f(x_j^m)]_i\right]^4 + 8\mathbb{E}_j\left[[\widehat{g_j^m}]_i - [\nabla f(x_j^m)]_i\right]^2[\nabla f(x_j^m)]_i^2 \\ &\overset{\text{Lemma B.2}}{\leq} 2\sigma_\infty^4 + 8\sigma_\infty^2 G_\infty^2. \end{aligned} \tag{D.68}$$

Let $b = B\lambda^2/2, V = 2K\sigma_\infty^2(\sigma_\infty^2 + 4G_\infty^2)$. Applying Lemma B.1, we have $\left|\sum_{j=rK}^{t}[\theta_j^m]_i\right| \leq b$ with probability no less than

$$1 - 2\exp\left(-\frac{b^2}{2V + 2cb/3}\right) \geq 1 - \frac{\delta}{4dMT}, \tag{D.69}$$

which implies with probability no less than $1 - \dfrac{\delta}{4T}$, for any $m \in [M]$,

$$v_t^m \preceq v_r + (1 - \beta_2) K (\sigma_\infty^2 + 3G_\infty^2) + (1 - \beta_2) B \lambda^2 / 2 \preceq v_r + (1 - \beta_2) B \lambda^2. \qquad \text{(D.70)}$$

and thus

$$H_t^m \preceq \sqrt{1 + (1 - \beta_2) B} H_r. \qquad \text{(D.71)}$$

$\square$

### D.3  PROOF OF CONTRACTION

In this subsection, we aim to show contraction, *i.e.*, $\|x_t^m - x_t^n\|$ will not get too large during local iterations with high probability. However, since the update of $x_t^m$ involves the coupling of both first order momentum and second order momentum, it is much harder than showing the contraction of *Local* SGDM. Our solution below is in two folds.

We begin with showing contraction of the second order momentum in some sense.

**Lemma D.9.** *Let* $B_1 := \max \left\{ \dfrac{16K\sigma_\infty^2}{\lambda^2}, \dfrac{16\rho^2}{\lambda^2} \log \dfrac{dMT}{\delta}, 2^6 \dfrac{\sqrt{K}(G_\infty + \sigma_\infty)\sigma_\infty}{\lambda^2} \log^{1/2} \dfrac{dMT}{\delta} \right\}$
*and* $1 - \beta_2 \leq \dfrac{1}{4K}$. *If* $\rho \geq \max\{3\sigma_\infty, 2G_\infty\}$, $\dfrac{\eta L \sigma}{\lambda} \sqrt{KA} G_\infty \leq 2\sigma_\infty^2$, *then the following holds:*

$$\mathbb{P}(E_{t,2}) \geq \mathbb{P}(E_{t,1}) - \dfrac{\delta}{4T} \qquad \text{(D.72)}$$

*Proof.* Event $E_{t,1}$ implies for all $j \leq t$, $x_j^m, x_j^n \in \Omega$ and for any index $i \in [d]$,

$$
\begin{aligned}
\left| [v_t^m - v_t^n]_i \right| &= \left| (1 - \beta_2) \sum_{j=rK}^{t} \beta_2^{t-j} \left[ [\widehat{g_j^m}]_i^2 - [\widehat{g_j^n}]_i^2 \right] \right| \\
&\leq \left| (1 - \beta_2) \sum_{j=rK}^{t} \beta_2^{t-j} \left[ [\widehat{g_j^m}]_i^2 - [\widehat{g_j^n}]_i^2 - \mathbb{E}_j \left[ [\widehat{g_j^m}]_i^2 - [\widehat{g_j^n}]_i^2 \right] \right] \right| \\
&\quad + \left| (1 - \beta_2) \sum_{j=rK}^{t} \beta_2^{t-j} \left[ \mathbb{E}_j \left[ [\widehat{g_j^m}]_i^2 - [\widehat{g_j^n}]_i^2 \right] - \left[ [\nabla f(x_j^m)]_i^2 - [\nabla f(x_j^n)]_i^2 \right] \right] \right| \\
&\quad + \left| (1 - \beta_2) \sum_{j=rK}^{t} \beta_2^{t-j} \left[ [\nabla f(x_j^m)]_i^2 - [\nabla f(x_j^n)]_i^2 \right] \right| \\
&\leq \left| (1 - \beta_2) \sum_{j=rK}^{t} \beta_2^{t-j} \left[ [\widehat{g_j^m}]_i^2 - [\widehat{g_j^n}]_i^2 - \mathbb{E}_j \left[ [\widehat{g_j^m}]_i^2 - [\widehat{g_j^n}]_i^2 \right] \right] \right| \\
&\quad + (1 - \beta_2) K \cdot 4\sigma_\infty^2 + (1 - \beta_2) K \cdot 2G_\infty \dfrac{\eta L \sigma}{\lambda} \sqrt{KA} \\
&\leq \left| (1 - \beta_2) \sum_{j=rK}^{t} \beta_2^{t-j} \left[ [\widehat{g_j^m}]_i^2 - [\widehat{g_j^n}]_i^2 - \mathbb{E}_j \left[ [\widehat{g_j^m}]_i^2 - [\widehat{g_j^n}]_i^2 \right] \right] \right| + 8(1 - \beta_2) K \cdot \sigma_\infty^2.
\end{aligned}
$$

$$\text{(D.73)}$$

Here in the second inequality we apply Lemma B.2 and contraction results implied by $E_{t,1}$.

Define

$$[\Xi_j^{m,n}]_i = \begin{cases} \beta_2^{t-j} \left[ [\widehat{g_j^m}]_i^2 - [\widehat{g_j^n}]_i^2 - \mathbb{E}_j \left[ [\widehat{g_j^m}]_i^2 - [\widehat{g_j^n}]_i^2 \right] \right], & \text{if event } E_j \text{ holds,} \\ 0, & \text{otherwise.} \end{cases} \qquad \text{(D.74)}$$

Then we have

$$\left| [\Xi_j^{m,n}]_i \right| \leq 2\rho^2 \overset{def}{=} c, \qquad \text{(D.75)}$$

$$
\begin{aligned}
\mathrm{Var}_j([\Xi_j^{m,n}]_i) &\leq 2\mathbb{E}_j\left[[\widehat{g_j^m}]_i^2 - \mathbb{E}_j[\widehat{g_j^m}]_i^2\right]^2 \\
&\leq 2\mathbb{E}_j\left[[\widehat{g_j^m}]_i^2 - [\nabla f(x_j^m)]_i^2\right]^2 \\
&\leq 4\mathbb{E}_j\left[[\widehat{g_j^m}]_i - [\nabla f(x_j^m)]_i\right]^2 \cdot \left[\left[[\widehat{g_j^m}]_i - [\nabla f(x_j^m)]_i\right]^2 + 4[\nabla f(x_j^m)]_i^2\right] \\
&\overset{\text{Lemma B.2}}{\leq} 4\sigma_\infty^4 + 16\sigma_\infty^2 G_\infty^2.
\end{aligned}
\tag{D.76}
$$

Let $b = B_1\lambda^2/2$, $V = 4K\sigma_\infty^2(\sigma_\infty^2 + 4G_\infty^2)$ and by Lemma B.1, we have $\left|\sum_{j=rK}^{t}[\Xi_j^{m,n}]_i\right| \leq b$ with probability no less than

$$
1 - 2\exp\left(\frac{b^2}{2V + 2cb/3}\right) \geq 1 - \frac{\delta}{4dM^2T}.
\tag{D.77}
$$

This implies with probability no less than $1 - \frac{\delta}{4M^2T}$,

$$
\left|v_t^m - v_t^n\right| \preceq (1-\beta_2)B_1\lambda^2/2 + 8(1-\beta_2)K \cdot \sigma_\infty^2 \preceq (1-\beta_2)B_1\lambda^2.
\tag{D.78}
$$

Combine this inequality and event $E_{t,1}$,

$$
\begin{aligned}
\left|\frac{H_r}{H_t^m} - \frac{H_r}{H_t^n}\right| &= \frac{\sqrt{v_r + \lambda^2}|v_t^n - v_t^m|}{\sqrt{v_t^m + \lambda^2}\sqrt{v_t^n + \lambda^2}(\sqrt{v_t^m + \lambda^2} + \sqrt{v_t^n + \lambda^2})} \\
&\preceq (1-\beta_2)B_1\frac{\sqrt{v_r + \lambda^2}}{(\sqrt{v_t^m + \lambda^2} + \sqrt{v_t^n + \lambda^2})} \\
&\preceq (1-\beta_2)B_1.
\end{aligned}
\tag{D.79}
$$

The last inequality is due to event $E_{t,1}$ and $1 - \beta_2 \leq \frac{1}{4K}$. We can conclude that under event $E_{t,1}$, with probability no less than $1 - \frac{\delta}{4T}$, the inequality above holds for any $m, n \in [M]$, which implies $\mathbb{P}(E_{t,2}) \geq \mathbb{P}(E_{t,1}) - \frac{\delta}{4T}$. $\qquad\square$

Now we are ready to prove contraction of $z_t^m$.

**Lemma D.10.** *Let* $A := \max\left\{\frac{2^{20}\rho^2 d}{K\sigma^2}\log\frac{MT}{\delta}, 2^{20}\log\frac{MT}{\delta}, \frac{2^8 K\|2\boldsymbol{\sigma}\|_{2\alpha}^{2\alpha}}{\sigma^2\rho^{2(\alpha-1)}}\right\}$. *If* $\eta \leq \min\left\{\frac{\lambda}{60K\tau}, \frac{(1-\beta_1)^2\lambda}{64L}\right\}$, $\rho \geq \max\{3\sigma_\infty, 2G_\infty\}$, *and*

$$
(1-\beta_2)K^{1/2} \leq \min\left\{\frac{(1-\beta_1)}{4B_1}, \frac{(1-\beta_1)\sigma}{2^{12}B_1 G}\sqrt{A}, \frac{1-\beta_1}{4B}\right\},
\tag{D.80}
$$

*then the following holds:*

$$
\mathbb{P}(E_{t,3}) \geq \mathbb{P}(E_{t,2}) - \frac{\delta}{4T}.
\tag{D.81}
$$

*Proof.* If $t \bmod K \equiv -1$, then $z_{t+1}^m = z_{t+1}^n$ for all $m, n$ and the claim is trivial. Below we assume that $t \bmod K \not\equiv -1$. The update rules implies

$$
\begin{aligned}
\|z_{t+1}^m - z_{t+1}^n\|_{H_r}^2 &\overset{(D.6)}{=} \|z_t^m - z_t^n\|_{H_r}^2 - 2\eta\left\langle z_t^m - z_t^n, (H_t^m)^{-1}(\widehat{g_t^m} + e_t^m) - (H_t^n)^{-1}(\widehat{g_t^n} + e_t^n)\right\rangle_{H_r} \\
&\quad + \eta^2 \underbrace{\left\|(H_t^m)^{-1}(\widehat{g_t^m} + e_t^m) - (H_t^n)^{-1}(\widehat{g_t^n} + e_t^n)\right\|_{H_r}^2}_{\textcircled{1}}.
\end{aligned}
\tag{D.82}
$$

Note that the first order term is

$$
\left\langle z_t^m - z_t^n, (H_t^m)^{-1}(\widehat{g_t^m} + e_t^m) - (H_t^n)^{-1}(\widehat{g_t^n} + e_t^n)\right\rangle_{H_r}
$$
$$
= \left\langle z_t^m - z_t^n, \nabla f(x_t^m) - \nabla f(x_t^n)\right\rangle
$$
$$
+ \left\langle z_t^m - z_t^n, \widehat{g_t^m} - \widehat{g_t^n} - \nabla f(x_t^m) + \nabla f(x_t^n)\right\rangle
$$
$$
+ \underbrace{\left\langle z_t^m - z_t^n, (H_t^m)^{-1}e_t^m - (H_t^n)^{-1}e_t^n\right\rangle_{H_r}}_{②}
$$
$$
+ \underbrace{\left\langle z_t^m - z_t^n, (H_r(H_t^m)^{-1} - I_d)\widehat{g_t^m} - (H_r(H_t^n)^{-1} - I_d)\widehat{g_t^n}\right\rangle}_{③}.
$$

(D.83)

And for the first term above,

$$
\langle z_t^m - z_t^n, \nabla f(x_t^m) - \nabla f(x_t^n)\rangle = \langle x_t^m - x_t^n, \nabla f(x_t^m) - \nabla f(x_t^n)\rangle
$$
$$
+ \langle z_t^m - z_t^n - (x_t^m - x_t^n), \nabla f(x_t^m) - \nabla f(x_t^n)\rangle
$$
$$
\geq \langle x_t^m - x_t^n, \nabla f(x_t^m) - \nabla f(x_t^n)\rangle
$$
$$
- \frac{L}{\lambda}\|(z_t^m - z_t^n) - (x_t^m - x_t^n)\|_{H_r}^2 - \frac{\lambda}{4L}\|\nabla f(x_t^m) - \nabla f(x_t^n)\|_{H_r^{-1}}^2
$$

(D.84)

By definition of $\{z_t^m\}$ and event $E_{t,2}$,

$$
\|(z_t^m - z_t^n) - (x_t^m - x_t^n)\|_{H_r}^2 = \left(\frac{\eta\beta_1}{1-\beta_1}\right)^2\|(H_t^m)^{-1}u_t^m - (H_t^n)^{-1}u_t^n\|_{H_r}^2
$$
$$
\leq 2\left(\frac{\eta\beta_1}{1-\beta_1}\right)^2\left[\|((H_t^m)^{-1} - (H_t^n)^{-1})u_t^m\|_{H_r}^2 + \|(H_t^n)^{-1}(u_t^m - u_t^n)\|_{H_r^{-1}}^2\right]
$$
$$
\overset{\mathcal{A}_{t,1},\mathcal{A}_{t,2}}{\leq} 2\left(\frac{\eta\beta_1}{1-\beta_1}\right)^2\left[[(1-\beta_2)B_1]^2\|u_t^m\|_{H_r^{-1}}^2 + 4\|u_t^m - u_t^n\|_{H_r^{-1}}^2\right].
$$

(D.85)

Besides,

$$
① \leq \underbrace{4\|(H_t^m)^{-1}e_t^m - (H_t^n)^{-1}e_t^n\|_{H_r}^2}_{(*)} + \underbrace{4\|(H_r(H_t^m)^{-1} - I_d)\widehat{g_t^m} - (H_r(H_t^n)^{-1} - I_d)\widehat{g_t^n}\|_{H_r^{-1}}^2}_{(**)}
$$
$$
+ 4\|\widehat{g_t^m} - \widehat{g_t^n} - \nabla f(x_t^m) + \nabla f(x_t^n)\|_{H_r^{-1}}^2 + 4\|\nabla f(x_t^m) - \nabla f(x_t^n)\|_{H_r^{-1}}^2,
$$

(D.86)

$$
|②| \leq \frac{1}{8\eta K}\|z_t^m - z_t^n\|_{H_r}^2 + 2\eta K \cdot (*).
$$

(D.87)

$$
|③| \leq \frac{1}{8\eta K}\|z_t^m - z_t^n\|_{H_r}^2 + 2\eta K \cdot (**).
$$

(D.88)

$$
(*) \overset{(D.5)}{=} \left(\frac{\beta_1}{1-\beta_1}\right)^2\|\left[(H_t^m)^{-1} - (H_{t-1}^m)^{-1}\right]u_t^m - \left[(H_t^n)^{-1} - (H_{t-1}^n)^{-1}\right]u_t^n\|_{H_r}^2
$$
$$
\leq 2\left(\frac{\beta_1}{1-\beta_1}\right)^2\left[\|\left[(H_t^m)^{-1} - (H_{t-1}^m)^{-1} - (H_t^n)^{-1} + (H_{t-1}^n)^{-1}\right]u_t^m\|_{H_r}^2 \right.
$$
$$
\left. + \|\left[(H_t^n)^{-1} - (H_{t-1}^n)^{-1}\right](u_t^m - u_t^n)\|_{H_r}^2\right]
$$
$$
\overset{\mathcal{A}_{t,1},\mathcal{A}_{t,2}}{\leq} 2\left(\frac{\beta_1}{1-\beta_1}\right)^2\left[4[(1-\beta_2)B_1]^2\|u_t^m\|_{H_r^{-1}}^2 + 4[(1-\beta_2)B]^2\|(u_t^m - u_t^n)\|_{H_r^{-1}}^2\right]
$$
$$
= 8\left(\frac{\beta_1(1-\beta_2)}{1-\beta_1}\right)^2\left[B_1^2\|u_t^m\|_{H_r^{-1}}^2 + B^2\|(u_t^m - u_t^n)\|_{H_r^{-1}}^2\right]
$$

(D.89)

$$(**) \leq 2 \left[ \left\| H_r((H_t^m)^{-1} - (H_t^n)^{-1})\widehat{g_t^m} \right\|_{H_r^{-1}}^2 + \left\| (H_r(H_t^n)^{-1} - I_d)(\widehat{g_t^m} - \widehat{g_t^n}) \right\|_{H_r^{-1}}^2 \right]$$

$$\overset{\mathcal{A}_{t,1}, \mathcal{A}_{t,2}}{\leq} 2 \left[ [(1-\beta_2)B_1]^2 \|\widehat{g_t^m}\|_{H_r^{-1}}^2 + [(1-\beta_2)B]^2 \|\widehat{g_t^m} - \widehat{g_t^m}\|_{H_r^{-1}}^2 \right]$$

$$\leq 2(1-\beta_2)^2 \left[ B_1^2 \|\widehat{g_t^m}\|_{H_r^{-1}}^2 + 2B^2 \left( \|\widehat{g_t^m} - \widehat{g_t^m} - \nabla f(x_t^m) + \nabla f(x_t^n)\|_{H_r^{-1}}^2 + \|\nabla f(x_t^m) - \nabla f(x_t^n)\|_{H_r^{-1}}^2 \right) \right]$$

(D.90)

Here we repeatedly apply $\|H_r(H_t^n)^{-1} - I_d\| \leq (1-\beta_2)B$ and $\|H_r((H_t^m)^{-1} - (H_t^n)^{-1})\| \leq (1-\beta_2)B_1$ by event $E_{t,2}$. Plug in (D.82),

$$\|z_{t+1}^m - z_{t+1}^n\|_{H_r}^2 \leq \|z_t^m - z_t^n\|_{H_r}^2 \overset{(D.83)}{\underbrace{- 2\eta \left\langle z_t^m - z_t^n, \widehat{g_t^m} - \widehat{g_t^n} - \nabla f(x_t^m) + \nabla f(x_t^n) \right\rangle}} \overset{(D.84)}{- 2\eta \langle x_t^m - x_t^n, \nabla f(x_t^m) - \nabla f(}$$

$$\underbrace{}_{(***)}$$

$$\overset{(D.84)}{+} 2\eta \left[ \frac{L}{\lambda} \|(z_t^m - z_t^n) - (x_t^m - x_t^n)\|_{H_r}^2 + \frac{\lambda}{4L} \|\nabla f(x_t^m) - \nabla f(x_t^n)\|_{H_r^{-1}}^2 \right]$$

$$\overset{(D.84)}{-} 2\eta \cdot (\text{②} + \text{③}) + \eta^2 \cdot \text{①}$$

$$\leq \|z_t^m - z_t^n\|_{H_r}^2 + (***) + 2\eta \left[ \frac{L}{\lambda} \|(z_t^m - z_t^n) - (x_t^m - x_t^n)\|_{H_r}^2 + \frac{\lambda}{4L} \|\nabla f(x_t^m) - \nabla f(x_t^n)\|_{H_r^{-1}}^2 \right]$$

$$+ 2\eta \left[ \frac{1}{4\eta K} \|z_t^m - z_t^n\|_{H_r}^2 \overset{(D.87)}{+} 2\eta K \cdot (*) + 2\eta K \cdot (**) \right]$$

$$\overset{(D.86)}{+} 4\eta^2 \left[ (*) + (**) + \|\widehat{g_t^m} - \widehat{g_t^n} - \nabla f(x_t^m) + \nabla f(x_t^n)\|_{H_r^{-1}}^2 + \|\nabla f(x_t^m) - \nabla f(x_t^n)\|_{H_r^{-1}}^2 \right]$$

$$\leq (1 + \frac{1}{2K})\|z_t^m - z_t^n\|_{H_r}^2 + (***) + \frac{2\eta L}{\lambda} \|(z_t^m - z_t^n) - (x_t^m - x_t^n)\|_{H_r}^2$$

$$+ (\frac{\eta}{2L} + \frac{4\eta^2}{\lambda})\|\nabla f(x_t^m) - \nabla f(x_t^n)\|^2 + 4\eta^2 \underbrace{\|\widehat{g_t^m} - \widehat{g_t^m} - \nabla f(x_t^m) + \nabla f(x_t^n)\|_{H_r^{-1}}^2}_{(\sharp)}$$

$$+ 8\eta^2 K \left( (*) + (**) \right)$$

$$\leq (1 + \frac{1}{2K})\|z_t^m - z_t^n\|_{H_r}^2 \underbrace{- 2\eta \langle x_t^m - x_t^n, \nabla f(x_t^m) - \nabla f(x_t^n) \rangle + \frac{\eta}{L} \|\nabla f(x_t^m) - \nabla f(x_t^n)\|^2}_{(\sharp\sharp)}$$

$$- 2\eta \left\langle z_t^m - z_t^n, \widehat{g_t^m} - \widehat{g_t^n} - \nabla f(x_t^m) + \nabla f(x_t^n) \right\rangle + 8\eta^2 \cdot (\sharp)$$

$$\overset{(D.85)}{+} \frac{4\eta L}{\lambda} \left( \frac{\eta\beta_1}{1-\beta_1} \right)^2 \left[ [(1-\beta_2)B_1]^2 \|u_t^m\|_{H_r^{-1}}^2 + 4\|u_t^m - u_t^n\|_{H_r^{-1}}^2 \right]$$

$$\overset{(D.89)}{+} 64\eta^2 K \left( \frac{\beta_1(1-\beta_2)}{1-\beta_1} \right)^2 \left[ B_1^2 \|u_t^m\|_{H_r^{-1}}^2 + B^2 \|(u_t^m - u_t^n)\|_{H_r^{-1}}^2 \right] \overset{(D.90)}{+} 16\eta^2 K(1-\beta_2)^2 B_1^2 \|\widehat{g_t^m}\|_{H}$$

$$\leq (1 + \frac{1}{2K})\|z_t^m - z_t^n\|_{H_r}^2 + (\sharp\sharp) + 8\eta^2 \cdot (\sharp)$$

$$- 2\eta \left\langle z_t^m - z_t^n, \widehat{g_t^m} - \widehat{g_t^n} - \mathbb{E}_t[\widehat{g_t^m} - \widehat{g_t^n}] \right\rangle - 2\eta \left\langle z_t^m - z_t^n, \mathbb{E}_t[\widehat{g_t^m} - \widehat{g_t^n}] - \nabla f(x_t^m) + \nabla f(x_t^n) \right\rangle$$

$$+ \underbrace{24\eta^2 \|u_t^m - u_t^n\|_{H_r^{-1}}^2 + 65\eta^2 K \left( \frac{\beta_1(1-\beta_2)}{1-\beta_1} \right)^2 B_1^2 \|u_t^m\|_{H_r^{-1}}^2 + 16\eta^2 K(1-\beta_2)^2 B_1^2 \|\widehat{g_t^m}\|_{H_r^{-1}}^2}_{(\sharp\sharp\sharp)}$$

$$\leq (1 + \frac{1}{K})\|z_t^m - z_t^n\|_{H_r}^2 + (\sharp\sharp) + 8\eta^2 \cdot (\sharp) - 2\eta \left\langle z_t^m - z_t^n, \widehat{g_t^m} - \widehat{g_t^n} - \mathbb{E}_t[\widehat{g_t^m} - \widehat{g_t^n}] \right\rangle$$

$$+ (\sharp\sharp\sharp) \overset{\text{Lemma B.2}}{+} \frac{8\eta^2 K}{\lambda} \cdot \frac{\|2\boldsymbol{\sigma}\|_{2\alpha}^{2\alpha}}{\rho^{2(\alpha-1)}}.$$

(D.91)

In the second to last inequality we apply $8K(1-\beta_2)^2 B^2 \le (1-\beta_1)^2$ and $\frac{\eta L}{\lambda} \le (1-\beta_1)^2$. Also notice that by definition of $\{u_t^m\}$,

$$u_t^m = (1-\beta_1)\sum_{j=rK}^{t}\beta_1^{t-j}\widehat{g_j^m} + \beta_1^{t-rK+1}u_r, \tag{D.92}$$

which implies

$$\|u_t^m\|_{H_r^{-1}}^2 \le (1-\beta_1)\sum_{j=rK}^{t}\beta_1^{t-j}\|\widehat{g_j^m}\|_{H_r^{-1}}^2 + \beta_1^{t-rK+1}\|u_r\|_{H_r^{-1}}^2. \tag{D.93}$$

$$\|u_t^m - u_t^n\|_{H_r^{-1}}^2 \le (1-\beta_1)\sum_{j=rK}^{t}\beta_1^{t-j}\|\widehat{g_j^m}-\widehat{g_j^n}\|_{H_r^{-1}}^2$$
$$\le 2(1-\beta_1)\sum_{j=rK}^{t}\beta_1^{t-j}\left[\|\nabla f(x_j^m)-\nabla f(x_j^n)\|_{H_r^{-1}}^2 + \|\widehat{g_j^m}-\widehat{g_j^n}-[\nabla f(x_j^m)-\nabla f(x_j^n)]\|_{H_r^{-1}}^2\right]. \tag{D.94}$$

And thus

$$\sum_{j=rK}^{t}\|u_j^m-u_j^n\|_{H_r^{-1}}^2 \le 2\sum_{j=rK}^{t}\left[\|\nabla f(x_j^m)-\nabla f(x_j^n)\|_{H_r^{-1}}^2 + \|\widehat{g_j^m}-\widehat{g_j^n}-[\nabla f(x_j^m)-\nabla f(x_j^n)]\|_{H_r^{-1}}^2\right]. \tag{D.95}$$

Unroll the recursive bound (D.91) and note that $(1+\frac{1}{K})^K \le 3$,

$$\|z_{t+1}^m - z_{t+1}^n\|_{H_r}^2 \le \underbrace{-\sum_{j=rK}^{t}2\eta(1+\frac{1}{K})^{t-j}\left\langle z_j^m-z_j^n, \widehat{g_j^m}-\widehat{g_j^n}-\mathbb{E}_j[\widehat{g_j^m}-\widehat{g_j^n}]\right\rangle}_{\text{①: martingale}}$$
$$+\sum_{j=rK}^{t}(1+\frac{1}{K})^{t-j}\left[-2\eta\langle x_j^m-x_j^n, \nabla f(x_j^m)-\nabla f(x_j^n)\rangle + \frac{\eta}{L}\|\nabla f(x_j^m)-\nabla f(x_j^n)\|^2\right]$$
$$+24\sum_{j=rK}^{t}\eta^2\|\widehat{g_j^m}-\widehat{g_j^n}-\nabla f(x_j^m)+\nabla f(x_j^n)\|_{H_r^{-1}}^2 + 72\eta^2\sum_{j=rK}^{t}\|u_j^m-u_j^n\|_{H_r^{-1}}^2$$
$$+195\eta^2 K\frac{(1-\beta_2)^2 B_1^2}{(1-\beta_1)^3}\|u_r\|_{H_r^{-1}}^2 + 48\eta^2 K\left(\frac{1-\beta_2}{1-\beta_1}\right)^2 B_1^2\sum_{j=rK}^{t}\|\widehat{g_j^m}\|_{H_r^{-1}}^2 + \frac{24\eta^2 K^2}{\lambda}\cdot\frac{\|2\boldsymbol{\sigma}\|_{2\alpha}^{2\alpha}}{\rho^{2(\alpha-1)}}$$
$$\overset{(D.95)}{\le}① + \sum_{j=rK}^{t}(1+\frac{1}{K})^{t-j}\left[-2\eta\langle x_j^m-x_j^n, \nabla f(x_j^m)-\nabla f(x_j^n)\rangle + \frac{2\eta}{L}\|\nabla f(x_j^m)-\nabla f(x_j^n)\|^2\right]$$
$$+144\sum_{j=rK}^{t}\eta^2\|\widehat{g_j^m}-\widehat{g_j^n}-\nabla f(x_j^m)+\nabla f(x_j^n)\|_{H_r^{-1}}^2 + 195\eta^2 K\frac{(1-\beta_2)^2 B_1^2}{(1-\beta_1)^3}\|u_r\|_{H_r^{-1}}^2$$
$$+48\eta^2 K\left(\frac{1-\beta_2}{1-\beta_1}\right)^2 B_1^2\sum_{j=rK}^{t}\|\widehat{g_j^m}\|_{H_r^{-1}}^2 + \frac{24\eta^2 K^2}{\lambda}\cdot\frac{\|2\boldsymbol{\sigma}\|_{2\alpha}^{2\alpha}}{\rho^{2(\alpha-1)}}. \tag{D.96}$$

Note that by definition, $u_r = (1-\beta_1)\sum_{j=1}^{K}\beta_1^{j-1}\mathbb{E}_m\widehat{g_{rK-j}^m} + \beta_1^K u_{r-1}$. By Cauchy-Schwarz inequality,

$$\|u_r\| \le \beta_1^K\|u_{r-1}\| + \sqrt{\sum_{j=1}^{K}\|\mathbb{E}_m\widehat{g_{rK-j}^m}\|^2\sum_{j=1}^{K}(1-\beta_1)^2\beta_1^{2(j-1)}}. \tag{D.97}$$

Therefore, event $E_{t,2}$ implies

$$\|u_r\|^2 \leq \frac{(1-\beta_1)^2\sigma^2 A}{2^{12}(1-\beta_2)^2 B_1^2} \cdot \frac{1-\beta_1}{1-\beta_1^K} \leq \frac{(1-\beta_1)^3\sigma^2 A}{2^{11}(1-\beta_2)^2 B_1^2}. \tag{D.98}$$

By Lemma D.5, and $\|\nabla f(x_j^m)\| \leq G$,

$$\|z_{t+1}^m - z_{t+1}^n\|_{H_r}^2 \leq ① \overset{\text{Lemma D.5}}{+} 6\eta\tau K \cdot \frac{\eta^2\sigma^2}{\lambda^2} KA$$

$$+ \frac{288\eta^2}{\lambda} \sum_{j=rK}^{t} \left[\|\widehat{g_j^m} - \nabla f(x_j^m)\|^2 + \|\widehat{g_j^n} - \nabla f(x_j^n)\|^2\right]$$

$$+ 96\eta^2 K \left(\frac{1-\beta_2}{1-\beta_1}\right)^2 \frac{B_1^2}{\lambda} \sum_{j=rK}^{t} \left(\|\widehat{g_j^m} - \nabla f(x_j^m)\|^2 + G^2\right)$$

$$\overset{(D.98)}{+} \frac{\eta^2\sigma^2 KA}{10\lambda} + \frac{24\eta^2 K^2}{\lambda} \cdot \frac{\|2\boldsymbol{\sigma}\|_{2\alpha}^{2\alpha}}{\rho^{2(\alpha-1)}} \tag{D.99}$$

$$\leq ① + 6\eta\tau K \cdot \frac{\eta^2\sigma^2}{\lambda^2} KA \overset{\text{Lemma B.2}}{+} \frac{2^{10}\eta^2}{\lambda} K\sigma^2$$

$$+ \frac{2^{10}\eta^2}{\lambda} \max_{s\in[M]} \underbrace{\sum_{j=rK}^{t} \left[\|\widehat{g_j^s} - \nabla f(x_j^s)\|^2 - \mathbb{E}_j[\|\widehat{g_j^s} - \nabla f(x_j^s)\|^2]\right]}_{②:\ \text{martingale}}$$

$$+ 96\eta^2 K^2 \left(\frac{1-\beta_2}{1-\beta_1}\right)^2 \frac{B_1^2}{\lambda} G^2 + \frac{\eta^2\sigma^2 KA}{10\lambda} + \frac{24\eta^2 K^2}{\lambda} \cdot \frac{\|2\boldsymbol{\sigma}\|_{2\alpha}^{2\alpha}}{\rho^{2(\alpha-1)}}.$$

Define

$$\zeta_j^{m,n} = \begin{cases} -2\eta(1+\frac{1}{K})^{t-j} \left\langle z_j^m - z_j^n, \widehat{g_j^m} - \widehat{g_j^n} - \mathbb{E}_j[\widehat{g_j^m} - \widehat{g_j^n}]\right\rangle, & \text{if event } E_j \text{ holds,} \\ 0, & \text{otherwise.} \end{cases} \tag{D.100}$$

$$\theta_j^m = \begin{cases} \|\widehat{g_j^m} - \nabla f(x_j^m)\|^2 - \mathbb{E}_j[\|\widehat{g_j^m} - \nabla f(x_j^m)\|^2], & \text{if event } E_j \text{ holds,} \\ 0, & \text{otherwise.} \end{cases} \tag{D.101}$$

Then (D.99) implies $\|z_{t+1}^m - z_{t+1}^n\|_{H_r}^2 \leq \frac{\eta^2\sigma^2}{2\lambda} KA + \sum_{j=rK}^{t} \zeta_j^{m,n} + \frac{2^{10}\eta^2}{\lambda} \max_{s\in[M]} \sum_{j=rK}^{t} \theta_j^s$. Note that by Lemma B.2,

$$|\theta_j^m| \leq 4\rho^2 d \overset{def}{=} c. \tag{D.102}$$

$$\text{Var}_j(\theta_j^m) \leq \mathbb{E}_j[\|\widehat{g_j^m} - \nabla f(x_j^m)\|^4] \leq \sigma^4. \tag{D.103}$$

Let $b = \frac{\sigma^2 KA}{2^{12}}, V = \sigma^4 K$. Then by Lemma B.1, $|\sum_{j=rK}^{t} \theta_j^m| \leq b$ with probability no less than

$$1 - 2\exp\left(\frac{b^2}{2V + 2cb/3}\right) \geq 1 - \frac{\delta}{8MT}. \tag{D.104}$$

This implies with probability no less than $1 - \frac{\delta}{8T}$,

$$|\sum_{j=rK}^{t} \theta_j^m| \leq \frac{\sigma^2 KA}{2^{12}}, \forall m \in [M]. \tag{D.105}$$

Also note that

$$|\zeta_j^{m,n}| \leq 6\eta \cdot \frac{\eta\sigma}{\lambda}\sqrt{KA} \cdot 4\rho\sqrt{d} = \frac{24\eta^2\sigma\rho\sqrt{d}}{\lambda}\sqrt{KA} \overset{def}{=} c. \tag{D.106}$$

$$\mathrm{Var}_j(\zeta_j^{m,n}) \le \left(6\eta \cdot \frac{\eta\sigma}{\lambda}\sqrt{KA}\right)^2 \cdot 2\sigma^2 = \frac{72\eta^4\sigma^4}{\lambda^2}KA. \tag{D.107}$$

Let $b = \frac{\eta^2\sigma^2}{4\lambda}KA, V = \frac{72\eta^4\sigma^4}{\lambda^2}K^2A$. Then by Lemma B.1, $|\sum_{j=rK}^{t}\zeta_j^{m,n}| \le b$ with probability no less than

$$1 - 2\exp\left(\frac{b^2}{2V + 2cb/3}\right) \ge 1 - \frac{\delta}{8M^2T}. \tag{D.108}$$

This implies with probability no less than $1 - \frac{\delta}{8T}$,

$$|\sum_{j=rK}^{t}\zeta_j^{m,n}| \le \frac{\eta^2\sigma^2}{4\lambda}KA, \forall m,n \in [M]. \tag{D.109}$$

We now turn to deal with $\sum_{j=rK}^{t}\|\widehat{g_j^m}\|^2$.

$$\sum_{j=rK}^{t}\|\widehat{g_j^m}\|^2 \le 2\sum_{j=rK}^{t}[\|\widehat{g_j^m} - \nabla f(x_j^m)\|^2 + \|\nabla f(x_j^m)\|^2]$$

$$\le 2\sum_{j=rK}^{t}\left[\|\widehat{g_j^m} - \nabla f(x_j^m)\|^2 - \mathbb{E}_j[\|\widehat{g_j^m} - \nabla f(x_j^m)\|^2]\right] + 2\sum_{j=rK}^{t}\mathbb{E}_j[\|\widehat{g_j^m} - \nabla f(x_j^m)\|^2] + 2KG^2$$

$$\overset{\text{Lemma B.2}}{\le} 2\sum_{j=rK}^{t}\left[\|\widehat{g_j^m} - \nabla f(x_j^m)\|^2 - \mathbb{E}_j[\|\widehat{g_j^m} - \nabla f(x_j^m)\|^2]\right] + 2K(\sigma^2 + G^2). \tag{D.110}$$

Then $\sum_{j=rK}^{t}\|\widehat{g_j^m}\|^2 \le 2\sum_{j=rK}^{t}\theta_j^m + 2K(\sigma^2 + G^2)$ under event $E_t$. Therefore, by (D.105),

$$\sum_{j=rK}^{t}\|\widehat{g_j^m}\|^2 \le \frac{\sigma^2KA}{2^{11}} + 2K(\sigma^2 + G^2) \le \frac{(1-\beta_1)^2\sigma^2A}{2^{12}(1-\beta_2)^2B_1^2}. \tag{D.111}$$

In conclusion, combining (D.105), (D.109), (D.111), we have

$$\mathbb{P}\left\{E_{t,2} \text{ and } \|z_{t+1}^m - z_{t+1}^n\|_{H_r}^2 \le \frac{\eta^2\sigma^2KA}{\lambda}, \sum_{j=rK}^{t}\|\widehat{g_j^m}\|^2 \le \frac{(1-\beta_1)^2\sigma^2A}{2^{12}(1-\beta_2)^2B_1^2} \text{ for all } m,n\right\} \ge \mathbb{P}(E_{t,2}) - \frac{\delta}{4T}. \tag{D.112}$$

$\square$

## D.4 PROOF OF DESCENT LEMMA

After laying all the groundwork above, we are now in the position of showing the main descent lemma.

**Lemma D.11.** *Assume that $\rho \ge \max\{3\sigma_\infty, 2G_\infty\}$ and*

$$\frac{\eta\sigma^2}{\lambda M}\log\frac{T}{\delta} \lesssim \Delta, \quad \frac{\eta\rho\sqrt{d}}{(1-\beta_1)\sqrt{\gamma\lambda}}\log^{\frac{1}{2}}\frac{T}{\delta} \lesssim \sqrt{\Delta}, \quad \frac{\left(\frac{\eta L}{\lambda}\right)^3\log\frac{T}{\delta}}{(1-\beta_1)(\sqrt{\beta_2}-\beta_1)} \lesssim \frac{L\Delta}{\rho^2 d}, \tag{D.113}$$

$$\left(\frac{\eta L}{\lambda}\right)^3\sigma^2KA \lesssim \frac{L\Delta}{T}, \quad \frac{\eta^2\sigma^2}{\lambda\gamma M} \lesssim \frac{\Delta}{T}, \quad \frac{\eta}{\lambda}\frac{\|2\boldsymbol{\sigma}\|_{2\alpha}^{2\alpha}}{\rho^{2(\alpha-1)}} \lesssim \frac{\Delta}{T},$$

*and*

$$(1-\beta_2)B \le \frac{\eta}{4\gamma} \le \frac{\eta L}{4\lambda}, \quad \frac{\eta L}{\lambda} \le \frac{(1-\beta_1)^2}{2^6}. \tag{D.114}$$

*Then the following holds:*

$$\mathbb{P}(E_{t+1}) \geq \mathbb{P}(E_{t,3}) - \frac{\delta}{4T}. \tag{D.115}$$

*Proof.* For any $x \in \mathbb{R}^d$, since $\nabla^2 f(\cdot) \succeq -\tau I_d$ and $H_r \succeq \lambda I_d$, $y \mapsto f(y) + \frac{1}{2\gamma}\|x-y\|_{H_r}^2$ is $(\frac{1}{\gamma} - \frac{\tau}{\lambda})$-convex with respect to $\|\cdot\|_{H_r}$. Note that under event $E_t$, $\overline{z}_t \in \Omega_0$. Let $y_t := \arg\min_y f(y) + \frac{1}{2\gamma}\|\overline{z}_t - y\|_{H_r}^2$ and by Lemma D.4, $y_t \in \Omega_0$. Then

$$f(y_t) + \frac{1}{2\gamma}\|y_t - \overline{z}_t\|_{H_r}^2 \leq f(\overline{z}_{t+1}) + \frac{1}{2\gamma}\|\overline{z}_{t+1} - \overline{z}_t\|_{H_r}^2 - \frac{1}{2}(\frac{1}{\gamma} - \frac{\tau}{\lambda})\|\overline{z}_{t+1} - y_t\|_{H_r}^2. \tag{D.116}$$

Recall that the definition of $\{z_t^m\}$ implies

$$\begin{aligned}
z_{t+1}^m - z_t^m &= -\frac{\eta(H_t^m)^{-1}u_t^m}{1 - \beta_1} + \frac{\eta\beta_1(H_{t-1}^m)^{-1}u_{t-1}^m}{1 - \beta_1} \\
&= -\frac{\eta\beta_1}{1 - \beta_1}[(H_t^m)^{-1} - (H_{t-1}^m)^{-1}]u_{t-1}^m - \eta(H_t^m)^{-1}\widehat{g_t^m} \\
&= -\eta(H_t^m)^{-1}(\widehat{g_t^m} + e_t^m).
\end{aligned} \tag{D.117}$$

Here $e_t^m = \frac{\beta_1}{1 - \beta_1}(I_d - H_t^m(H_{t-1}^m)^{-1})u_{t-1}^m$.

Also, since $\|\overline{z}_{t+1} - \overline{z}_t\| \leq \frac{(1 + \beta_1)\eta\rho\sqrt{d}}{(1 - \beta_1)\lambda} \leq \sqrt{\frac{\Delta\gamma}{160\lambda}} = R_0$, we have $\overline{z}_{t+1} \in \Omega$ and

$$\begin{aligned}
f(\overline{z}_{t+1}) - f(y_t) &\leq f(\overline{z}_t) + \langle\nabla f(\overline{z}_t), \overline{z}_{t+1} - \overline{z}_t\rangle + \frac{L}{2}\|\overline{z}_{t+1} - \overline{z}_t\|^2 - f(y_t) \\
&\leq \langle\nabla f(\overline{z}_t), \overline{z}_{t+1} - y_t\rangle + \frac{\tau}{2}\|\overline{z}_t - y_t\|^2 + \frac{L}{2}\|\overline{z}_{t+1} - \overline{z}_t\|^2 \\
&\leq \langle\nabla f(\overline{z}_t), \overline{z}_{t+1} - y_t\rangle + \frac{\tau}{2\lambda}\|\overline{z}_t - y_t\|_{H_r}^2 + \frac{L}{2\lambda}\|\overline{z}_{t+1} - \overline{z}_t\|_{H_r}^2.
\end{aligned} \tag{D.118}$$

Combine this with (D.116),

$$\begin{aligned}
&\frac{\frac{1}{\eta} + \frac{1}{\gamma} - \frac{\tau}{\lambda}}{2}\|\overline{z}_{t+1} - y_t\|_{H_r}^2 - \frac{\frac{1}{\eta} - \frac{1}{\gamma} + \frac{\tau}{\lambda}}{2}\|\overline{z}_t - y_t\|_{H_r}^2 + \frac{\frac{1}{\eta} + \frac{1}{\gamma} - \frac{L}{\lambda}}{2}\|\overline{z}_{t+1} - \overline{z}_t\|_{H_r}^2 \\
&\leq \left\langle \overline{z}_{t+1} - y_t, \nabla f(\overline{z}_t) + \frac{H_r(\overline{z}_{t+1} - \overline{z}_t)}{\eta} \right\rangle \\
&= \left\langle \overline{z}_t - \eta\mathbb{E}_m[(H_t^m)^{-1}(\widehat{g_t^m} + e_t^m)] - y_t, \nabla f(\overline{z}_t) - H_r\mathbb{E}_m[(H_t^m)^{-1}(\widehat{g_t^m} + e_t^m)] \right\rangle \\
&= \left\langle \overline{z}_t - \eta H_r^{-1}\nabla f(\overline{z}_t) - y_t, \nabla f(\overline{z}_t) - H_r\mathbb{E}_m[(H_t^m)^{-1}(\widehat{g_t^m} + e_t^m)] \right\rangle \\
&\quad + \eta\|\nabla f(\overline{z}_t) - H_r\mathbb{E}_m[(H_t^m)^{-1}(\widehat{g_t^m} + e_t^m)]\|_{H_r^{-1}}^2 \\
&\leq \left\langle \overline{z}_t - \eta H_r^{-1}\nabla f(\overline{z}_t) - y_t, \nabla f(\overline{z}_t) - H_r\mathbb{E}_m[(H_t^m)^{-1}(\widehat{g_t^m} + e_t^m)] \right\rangle \\
&\quad + 4\eta\|\nabla f(\overline{z}_t) - \mathbb{E}_m[\nabla f(x_t^m)]\|_{H_r^{-1}}^2 + 4\eta\|\mathbb{E}_m[\nabla f(x_t^m) - \widehat{g_t^m}]\|_{H_r^{-1}}^2 \\
&\quad + 4\eta\left\|\mathbb{E}_m[(H_r(H_t^m)^{-1} - I_d)\widehat{g_t^m}]\right\|_{H_r^{-1}}^2 + 4\eta\left\|\mathbb{E}_m[(H_t^m)^{-1}e_t^m]\right\|_{H_r}^2.
\end{aligned} \tag{D.119}$$

By Lemma D.4, we have

$$
\begin{aligned}
& \left\langle \overline{z}_t - \eta H_r^{-1} \nabla f(\overline{z}_t) - y_t, \nabla f(\overline{z}_t) - H_r \mathbb{E}_m[(H_t^m)^{-1} \widehat{g_t^m}] \right\rangle \\
= & \left\langle \overline{z}_t - \eta H_r^{-1} \nabla f(\overline{z}_t) - y_t, \nabla f(\overline{z}_t) - \mathbb{E}_m[\nabla f(x_t^m)] \right\rangle \\
& + \left\langle \overline{z}_t - \eta H_r^{-1} \nabla f(\overline{z}_t) - y_t, \mathbb{E}_m[\nabla f(x_t^m) - \widehat{g_t^m}] \right\rangle \\
& + \left\langle \overline{z}_t - \eta H_r^{-1} \nabla f(\overline{z}_t) - y_t, \mathbb{E}_m[(I_d - H_r(H_t^m)^{-1}) \widehat{g_t^m}] \right\rangle \\
\overset{(D.44)}{\leq} & \frac{\gamma}{16} \|\nabla f(y_t)\|_{H_r^{-1}}^2 + 8\gamma \|\nabla f(\overline{z}_t) - \mathbb{E}_m[\nabla f(x_t^m)]\|_{H_r^{-1}}^2 + 8\gamma \left\| \mathbb{E}_m[(H_r(H_t^m)^{-1} - I_d)\widehat{g_t^m}] \right\|_{H_r^{-1}}^2 \\
& + \left\langle \overline{z}_t - \eta H_r^{-1} \nabla f(\overline{z}_t) - y_t, \mathbb{E}_m[\nabla f(x_t^m) - \widehat{g_t^m}] \right\rangle .
\end{aligned}
\tag{D.120}
$$

Also,

$$
\left\langle \overline{z}_t - \eta H_r^{-1} \nabla f(\overline{z}_t) - y_t, -H_r \mathbb{E}_m[(H_t^m)^{-1} e_t^m] \right\rangle \leq \frac{\gamma}{16} \|\nabla f(y_t)\|_{H_r^{-1}}^2 + 4\gamma \left\| \mathbb{E}_m[(H_t^m)^{-1} e_t^m] \right\|_{H_r}^2
\tag{D.121}
$$

Further noticing that $\eta \leq \dfrac{\gamma}{4}$ and by AM-GM inequality, we conclude that

LHS of (D.119)

$$
\begin{aligned}
\leq & \frac{\gamma}{8} \|\nabla f(y_t)\|_{H_r^{-1}}^2 + 9\gamma \|\nabla f(\overline{z}_t) - \mathbb{E}_m[\nabla f(x_t^m)]\|_{H_r^{-1}}^2 + 9\gamma \left\| \mathbb{E}_m[(H_r(H_t^m)^{-1} - I_d)\widehat{g_t^m}] \right\|_{H_r^{-1}}^2 \\
& + 4\eta \left\| \mathbb{E}_m[\nabla f(x_t^m) - \widehat{g_t^m}] \right\|_{H_r^{-1}}^2 + 5\gamma \left\| \mathbb{E}_m[(H_t^m)^{-1} e_t^m] \right\|_{H_r}^2 \\
& + \left\langle \overline{z}_t - \eta H_r^{-1} \nabla f(\overline{z}_t) - y_t, \mathbb{E}_m[\nabla f(x_t^m) - \widehat{g_t^m}] \right\rangle .
\end{aligned}
\tag{D.122}
$$

If $t \bmod K \equiv -1$, then $r(t+1) = r(t) + 1 = r + 1$ and event $E_{t,1}$ implies

$$
H_r^{-1} H_{r+1} \preceq 1 + (1 - \beta_2) B \preceq 1 + \frac{\eta}{4\gamma},
\tag{D.123}
$$

$$
\begin{aligned}
f_\gamma^{H_{r+1}}(\overline{z}_{t+1}) & \leq f(y_t) + \frac{1}{2\gamma} \|\overline{z}_{t+1} - y_t\|_{H_{r+1}}^2 \\
& \leq f(y_t) + \frac{1 + \eta/4\gamma}{2\gamma} \|\overline{z}_{t+1} - y_t\|_{H_r}^2 .
\end{aligned}
\tag{D.124}
$$

On the other hand, if $t \bmod K \not\equiv -1$, then $r(t+1) = r(t) = r$,

$$
f_\gamma^{H_{r(t+1)}}(\overline{z}_{t+1}) \leq f(y_t) + \frac{1}{2\gamma} \|\overline{z}_{t+1} - y_t\|_{H_r}^2 .
\tag{D.125}
$$

Hence the following always holds:

$$f_\gamma^{H_{r(t+1)}}(\bar{z}_{t+1}) \le f_\gamma^{H_r}(\bar{z}_t) - \frac{1}{2\gamma}\|\bar{z}_t - y_t\|_{H_r}^2 + \frac{1+\eta/4\gamma}{2\gamma}\|\bar{z}_{t+1} - y_t\|_{H_r}^2$$

$$\overset{(D.122)}{\le} f_\gamma^{H_r}(\bar{z}_t) - \frac{7\gamma^{-1}}{8\gamma(\eta^{-1}+\gamma^{-1})}\|\bar{z}_t - y_t\|_{H_r}^2$$

$$+ \frac{(1+\eta/4\gamma)\left[\frac{1}{8}\|\nabla f(y_t)\|_{H_r^{-1}}^2 + 9\|\nabla f(\bar{z}_t) - \mathbb{E}_m[\nabla f(x_t^m)]\|_{H_r^{-1}}^2 + 9\left\|\mathbb{E}_m[(H_r(H_t^m)^{-1} - I_d)\widehat{g_t^m}]\right\|_{H_r^{-1}}^2\right]}{\eta^{-1}+\gamma^{-1}-\tau/\lambda}$$

$$+ \frac{(1+\eta/4\gamma)\left[4\eta\left\|\mathbb{E}_m[\nabla f(x_t^m) - \widehat{g_t^m}]\right\|_{H_r^{-1}}^2 + 5\gamma\left\|\mathbb{E}_m[(H_t^m)^{-1}e_t^m]\right\|_{H_r}^2\right]}{\gamma(\eta^{-1}+\gamma^{-1}-\tau/\lambda)}$$

$$+ \frac{(1+\eta/4\gamma)\left\langle \bar{z}_t - \eta H_r^{-1}\nabla f(\bar{z}_t) - y_t, \mathbb{E}_m[\nabla f(x_t^m) - \widehat{g_t^m}]\right\rangle}{\gamma(\eta^{-1}+\gamma^{-1}-\tau/\lambda)}$$

$$\overset{(D.42)}{\le} f_\gamma^{H_r}(\bar{z}_t) - \frac{\eta}{8}\|\nabla f(y_t)\|_{H_r^{-1}}^2 + \frac{5\eta^2}{\lambda\gamma}\|\mathbb{E}_m[\nabla f(x_t^m) - \widehat{g_t^m}]\|^2 + 6\eta\left\|\mathbb{E}_m[(H_t^m)^{-1}e_t^m]\right\|_{H_r}^2$$

$$+ \frac{10\eta}{\lambda}\|\nabla f(\bar{z}_t) - \mathbb{E}_m[\nabla f(x_t^m)]\|^2 + 10\eta\left\|\mathbb{E}_m[(H_r(H_t^m)^{-1} - I_d)\widehat{g_t^m}]\right\|_{H_r^{-1}}^2$$

$$+ \frac{1+\eta/4\gamma}{\gamma(\eta^{-1}+\gamma^{-1}-\tau/\lambda)}\left\langle \bar{z}_t - \eta H_r^{-1}\nabla f(\bar{z}_t) - y_t, \mathbb{E}_m[\nabla f(x_t^m) - \widehat{g_t^m}]\right\rangle.$$

$$(D.126)$$

Sum over $t$ and we get

$$f_\gamma^{H_{r(t+1)}}(\bar{z}_{t+1}) \le f_\gamma^\lambda(x_0) - \frac{\eta}{8}\sum_{j=0}^t \|\nabla f(y_j)\|_{H_{r(j)}^{-1}}^2 + \frac{5\eta^2}{\lambda\gamma}\sum_{j=0}^t \|\mathbb{E}_m[\nabla f(x_j^m) - \widehat{g_j^m}]\|^2 + 6\eta\sum_{j=0}^t\left\|\mathbb{E}_m[(H_j^m)^{-1}e_j^m]\right\|_{H_{r(j)}}^2$$

$$+ \frac{10\eta}{\lambda}\sum_{j=0}^t\|\nabla f(\bar{z}_j) - \mathbb{E}_m[\nabla f(x_j^m)]\|^2 + 10\eta\sum_{j=0}^t\left\|\mathbb{E}_m[(H_{r(j)}(H_j^m)^{-1} - I_d)\widehat{g_j^m}]\right\|_{H_{r(j)}^{-1}}^2$$

$$+ \underbrace{\frac{1+\eta/4\gamma}{\gamma(\eta^{-1}+\gamma^{-1}-\tau/\lambda)}\sum_{j=0}^t\left\langle \bar{z}_j - \eta H_{r(j)}^{-1}\nabla f(\bar{z}_j) - y_j, \mathbb{E}_m[\nabla f(x_j^m) - \widehat{g_j^m}]\right\rangle}_{(*)}.$$

$$(D.127)$$

By AM-GM inequality and notice that $\bar{x}_t, \bar{z}_t \in \Omega$,

$$\|\nabla f(\bar{z}_t) - \mathbb{E}_m[\nabla f(x_t^m)]\|^2$$
$$\le 2\|\nabla f(\bar{z}_t) - \nabla f(\bar{x}_t)\|^2 + 2\|\nabla f(\bar{x}_t) - \mathbb{E}_m[\nabla f(x_t^m)]\|^2 \qquad (D.128)$$
$$\le 2L^2\|\bar{z}_t - \bar{x}_t\|^2 + 2\|\nabla f(\bar{x}_t) - \mathbb{E}_m[\nabla f(x_t^m)]\|^2.$$

Under event $E_{t,3}$,

$$\left\|\mathbb{E}_m[(H_r(H_t^m)^{-1} - I_d)\widehat{g_t^m}]\right\|_{H_r^{-1}}^2 \le (1-\beta_2)^2 B^2 \mathbb{E}_m\left[\|\widehat{g_t^m}\|_{H_r^{-1}}^2\right]. \qquad (D.129)$$

$$\left\|\mathbb{E}_m[(H_t^m)^{-1}e_t^m]\right\|_{H_r}^2 \le 4\left(\frac{\beta_1(1-\beta_2)}{1-\beta_1}\right)^2 B^2 \mathbb{E}_m\left[\|u_{t-1}^m\|_{H_r^{-1}}^2\right]. \qquad (D.130)$$

By the definition of $u_{t-1}^m$, we have

$$\mathbb{E}_m\left[\|u_{t-1}^m\|_{H_r^{-1}}^2\right] \le (1-\beta_1)\sum_{j=0}^{t-1}\beta_1^{t-j-1}\mathbb{E}_m\left[\|\widehat{g_j^m}\|_{H_r^{-1}}^2\right]$$

$$\le \frac{(1-\beta_1)}{\beta_2^{K/2}}\sum_{j=0}^{t-1}(\beta_1/\sqrt{\beta_2})^{t-j-1}\mathbb{E}_m\left[\|\widehat{g_j^m}\|_{H_{r(j)}^{-1}}^2\right]. \qquad (D.131)$$

Plug these inequalities above in (D.127),

$$f_\gamma^{H_{r(t+1)}}(\overline{z}_{t+1}) \leq f_\gamma^\lambda(x_0) - \frac{\eta}{8} \sum_{j=0}^t \|\nabla f(y_j)\|^2_{H_{r(j)}^{-1}} + \frac{5\eta^2}{\lambda\gamma} \sum_{j=0}^t \|\mathbb{E}_m[\nabla f(x_j^m) - \widehat{g_j^m}]\|^2$$

$$\overset{(D.128)}{+} \frac{20\eta}{\lambda} \sum_{j=0}^t \left[ L^2\|\overline{z}_j - \overline{x}_j\|^2 + \|\nabla f(\overline{x}_j) - \mathbb{E}_m[\nabla f(x_j^m)]\|^2 \right]$$

$$\overset{(D.129)\text{-}(D.131)}{+} \eta\left( \frac{48\beta_1^2}{(1-\beta_1)(\sqrt{\beta_2}-\beta_1)} + 10 \right)(1-\beta_2)^2 B^2 \sum_{j=0}^t \mathbb{E}_m\left[ \|\widehat{g_j^m}\|^2_{H_{r(j)}^{-1}} \right] + (*).$$
(D.132)

By AM-GM inequality and Lemma D.4,

$$\mathbb{E}_m\left[ \|\widehat{g_t^m}\|^2_{H_r^{-1}} \right] \leq 4\mathbb{E}_m\left[ \|\widehat{g_t^m} - \nabla f(x_t^m)\|^2_{H_r^{-1}} + \|\nabla f(x_t^m) - \nabla f(\overline{x}_t)\|^2_{H_r^{-1}} \right.$$

$$\left. + \|\nabla f(\overline{x}_t) - \nabla f(\overline{z}_t)\|^2_{H_r^{-1}} + \|\nabla f(\overline{z}_t)\|^2_{H_r^{-1}} \right]$$

$$\leq \frac{4}{\lambda}\left[ \mathbb{E}_m\|\widehat{g_t^m} - \nabla f(x_t^m)\|^2 + L^2\mathbb{E}_m[\|x_t^m - \overline{x}_t\|^2] + L^2\|\overline{z}_t - \overline{x}_t\|^2 \right] + \frac{16(\gamma L)^2}{\lambda^2}\|\nabla f_\gamma^{H_r}(\overline{z}_t)\|^2_{H_r^{-1}}.$$
(D.133)

Therefore, we achieve that

$$f_\gamma^{H_{r(t+1)}}(\overline{z}_{t+1}) \leq f_\gamma^{H_0}(x_0) - \frac{\eta}{9} \sum_{j=0}^t \|\nabla f(y_j)\|^2_{H_{r(j)}^{-1}} + \frac{5\eta^2}{\lambda\gamma} \sum_{j=0}^t \|\mathbb{E}_m[\nabla f(x_j^m) - \widehat{g_j^m}]\|^2$$

$$+ \frac{40\eta}{\lambda} \sum_{j=0}^t \left[ L^2\|\overline{z}_j - \overline{x}_j\|^2 + \|\nabla f(\overline{x}_j) - \mathbb{E}_m[\nabla f(x_j^m)]\|^2 \right]$$

$$+ \frac{160\eta(1-\beta_2)^2 B^2}{\lambda(1-\beta_1)(\sqrt{\beta_2}-\beta_1)} \sum_{j=0}^t \left[ \mathbb{E}_m\|\widehat{g_j^m} - \nabla f(x_j^m)\|^2 + L^2\mathbb{E}_m[\|x_j^m - \overline{x}_j\|^2] \right] + (*).$$
(D.134)

By (D.160), (D.164) in Lemma D.12, under event $E_{t,3}$,

$$\|\overline{z}_j - \overline{x}_j\|^2 \leq \left( \frac{\beta_1}{1-\beta_1} \right)^2 \left[ 64\eta^2\left( \|\nabla f(\overline{z}_j)\|^2_{H_{r(j)}^{-2}} + \frac{L^2}{\lambda^2}\Lambda_{j-1} \right) \right.$$

$$\left. + \frac{36\eta^2}{\lambda^2}(1-\beta_1) \sum_{i=r(j)K}^{j-1} \beta_1^{j-i-1}\left[ \frac{\eta^2 L^2\sigma^2}{\lambda^2}KA + \mathbb{E}_m\|\widehat{g_i^m} - \nabla f(x_i^m)\|^2 \right] \right].$$
(D.135)

Hence

$$\sum_{j=0}^t \|\overline{z}_j - \overline{x}_j\|^2 \leq \left( \frac{\beta_1}{1-\beta_1} \right)^2 \left[ 64\eta^2 \sum_{j=0}^t \left( \|\nabla f(\overline{z}_j)\|^2_{H_{r(j)}^{-2}} + \frac{L^2}{\lambda^2}\Lambda_{j-1} \right) \right.$$

$$\left. + \frac{36\eta^2}{\lambda^2} \sum_{j=0}^{t-1}\left[ \frac{\eta^2 L^2\sigma^2}{\lambda^2}KA + \mathbb{E}_m\|\widehat{g_j^m} - \nabla f(x_j^m)\|^2 \right] \right].$$
(D.136)

Additionally by Lemma D.12,

$$\Lambda_t + \frac{(1-\beta_1)^2}{2} \sum_{j=0}^{t-1}\Lambda_j \leq \frac{64\eta^2}{1-\beta_1} \sum_{j=0}^t \|\nabla f(\overline{z}_j)\|^2_{H_{r(j)}^{-2}}$$

$$+ \frac{36\eta^2}{\lambda^2}(1-\beta_1) \sum_{j=0}^{t-1}\left[ \frac{\eta^2 L^2\sigma^2}{\lambda^2}KA + \mathbb{E}_m\|\widehat{g_j^m} - \nabla f(x_j^m)\|^2 \right].$$
(D.137)

Therefore, by noticing that $\Lambda_t \geq 0$ and $\dfrac{\eta L}{\lambda} \leq \dfrac{(1-\beta_1)^2}{16}$,

$$\sum_{j=0}^{t} \|\overline{z}_j - \overline{x}_j\|^2 \leq 2\left(\frac{\eta\beta_1}{1-\beta_1}\right)^2 \left[64\sum_{j=0}^{t}\|\nabla f(\overline{z}_j)\|_{H_{r(j)}^{-2}}^2 + \frac{36}{\lambda^2}\sum_{j=0}^{t-1}\left[\frac{\eta^2 L^2 \sigma^2}{\lambda^2}KA + \mathbb{E}_m\|\widehat{g_j^m} - \nabla f(x_j^m)\|^2\right]\right] \tag{D.138}$$

For the third term of RHS of (D.130),

$$\frac{5\eta^2}{\lambda\gamma}\sum_{j=0}^{t}\|\mathbb{E}_m[\nabla f(x_j^m) - \widehat{g_j^m}]\|^2 \leq \frac{10\eta^2}{\lambda\gamma}\sum_{j=0}^{t}\left[\|\mathbb{E}_m[\widehat{g_j^m} - \mathbb{E}_j[\widehat{g_j^m}]]\|^2 + \|\mathbb{E}_m[\nabla f(x_j^m) - \mathbb{E}_j[\widehat{g_j^m}]]\|^2\right]$$

$$\overset{\text{Lemma B.2}}{\leq} \frac{10\eta^2}{\lambda\gamma}\sum_{j=0}^{t}\left[\|\mathbb{E}_m[\widehat{g_j^m} - \mathbb{E}_j[\widehat{g_j^m}]]\|^2 + \frac{\|2\boldsymbol{\sigma}\|_{2\alpha}^{2\alpha}}{\rho^{2(\alpha-1)}}\right]$$

$$\leq \underbrace{\frac{10\eta^2}{\lambda\gamma}\sum_{j=0}^{t}\left[\|\mathbb{E}_m[\widehat{g_j^m} - \mathbb{E}_j[\widehat{g_j^m}]]\|^2 - \mathbb{E}_j\left[\|\mathbb{E}_m[\widehat{g_j^m} - \mathbb{E}_j[\widehat{g_j^m}]]\|^2\right]\right]}_{\text{①: martingale}}$$

$$+ \frac{10\eta^2 T}{\lambda\gamma}\left[\frac{\|2\boldsymbol{\sigma}\|_{2\alpha}^{2\alpha}}{\rho^{2(\alpha-1)}} + \frac{\sigma^2}{M}\right] \tag{D.139}$$

For the $(*)$ term of RHS of (D.130),

$$\frac{1+\eta/4\gamma}{\gamma(\eta^{-1}+\gamma^{-1}-\tau/\lambda)}\sum_{j=0}^{t}\left\langle \overline{z}_j - \eta H_{r(j)}^{-1}\nabla f(\overline{z}_j) - y_j, \mathbb{E}_m[\nabla f(x_j^m) - \widehat{g_j^m}]\right\rangle$$

$$= \frac{1+\eta/4\gamma}{\gamma(\eta^{-1}+\gamma^{-1}-\tau/\lambda)}\sum_{j=0}^{t}\left\langle \overline{z}_j - \eta H_{r(j)}^{-1}\nabla f(\overline{z}_j) - y_j, \mathbb{E}_m[\nabla f(x_j^m) - \mathbb{E}_j[\widehat{g_j^m}]]\right\rangle$$

$$+ \underbrace{\frac{1+\eta/4\gamma}{\gamma(\eta^{-1}+\gamma^{-1}-\tau/\lambda)}\sum_{j=0}^{t}\left\langle \overline{z}_j - \eta H_{r(j)}^{-1}\nabla f(\overline{z}_j) - y_j, \mathbb{E}_m[\mathbb{E}_j[\widehat{g_j^m}] - \widehat{g_j^m}]\right\rangle}_{\text{②: martingale}} \tag{D.140}$$

$$\overset{\text{AM-GM}}{\leq} \frac{2\eta}{\gamma}\sum_{j=0}^{t}\left[\frac{1}{120\gamma}\|H_{r(j)}(\overline{z}_j - y_j) - \eta\nabla f(\overline{z}_j)\|_{H_{r(j)}^{-1}}^2 + 30\gamma\frac{\|2\boldsymbol{\sigma}\|_{2\alpha}^{2\alpha}}{\lambda\rho^{2(\alpha-1)}}\right] + ②$$

$$\overset{(D.44)}{\leq} \frac{\eta}{60}\sum_{j=0}^{t}\|\nabla f(y_j)\|_{H_{r(j)}^{-1}}^2 + \frac{60\eta T}{\lambda}\frac{\|2\boldsymbol{\sigma}\|_{2\alpha}^{2\alpha}}{\rho^{2(\alpha-1)}} + ②$$

Here we remark that ② is a martingale because $H_{r(j)}$ only depends on stochastic gradients drawn strictly before round $r(j)$ and thus independent of $\widehat{g_j^m}$, which is drawn during round $r(j)$.

Plug (D.138),(D.139), (D.140) in (D.130),

$$
\begin{aligned}
f_\gamma^{H_{r(t+1)}}(\overline{z}_{t+1}) &\le f_\gamma^\lambda(x_0) - \frac{\eta}{12}\sum_{j=0}^t \|\nabla f(y_j)\|_{H_{r(j)}^{-1}}^2 + \text{①} + \frac{10\eta^2 T}{\lambda\gamma}\left[\frac{\|2\boldsymbol{\sigma}\|_{2\alpha}^{2\alpha}}{\rho^{2(\alpha-1)}} + \frac{\sigma^2}{M}\right] \\
&\quad + \frac{40\eta}{\lambda}\sum_{j=0}^t\left[\frac{72(\eta L\beta_1)^2}{(\lambda(1-\beta_1))^2}\left[\frac{\eta^2 L^2\sigma^2}{\lambda^2}KA + \mathbb{E}_m\|\widehat{g_j^m} - \nabla f(x_j^m)\|^2\right] + \frac{\eta^2 L^2\sigma^2}{\lambda^2}KA\right] \\
&\quad + \frac{160\eta(1-\beta_2)^2 B^2}{\lambda(1-\beta_1)(\sqrt{\beta_2}-\beta_1)}\sum_{j=0}^t\left[\mathbb{E}_m\|\widehat{g_j^m} - \nabla f(x_j^m)\|^2 + \frac{\eta^2 L^2\sigma^2}{\lambda^2}KA\right] \\
&\quad + \frac{60\eta T}{\lambda}\frac{\|2\boldsymbol{\sigma}\|_{2\alpha}^{2\alpha}}{\rho^{2(\alpha-1)}} + \text{②} \\
&\le f_\gamma^\lambda(x_0) - \frac{\eta}{12}\sum_{j=0}^t \|\nabla f(y_j)\|_{H_{r(j)}^{-1}}^2 + \text{①} + \frac{10\eta^2 T}{\lambda\gamma}\left[\frac{\|2\boldsymbol{\sigma}\|_{2\alpha}^{2\alpha}}{\rho^{2(\alpha-1)}} + \frac{\sigma^2}{M}\right] \\
&\quad + \frac{160\eta}{\lambda}\frac{[18(\frac{\eta L\beta_1}{\lambda})^2 + (1-\beta_2)^2 B^2]}{(1-\beta_1)(\sqrt{\beta_2}-\beta_1)}\sum_{j=0}^t\left[\mathbb{E}_m\|\widehat{g_j^m} - \nabla f(x_j^m)\|^2\right] \\
&\quad + \frac{160\eta T}{\lambda}\cdot\left[\frac{1}{4} + \frac{18(\frac{\eta L\beta_1}{\lambda})^2 + (1-\beta_2)^2 B^2}{(1-\beta_1)(\sqrt{\beta_2}-\beta_1)}\right]\cdot\frac{\eta^2 L^2\sigma^2}{\lambda^2}KA \\
&\quad + \frac{60\eta T}{\lambda}\frac{\|2\boldsymbol{\sigma}\|_{2\alpha}^{2\alpha}}{\rho^{2(\alpha-1)}} + \text{②} \\
&\le f_\gamma^\lambda(x_0) - \frac{\eta}{12}\sum_{j=0}^t \|\nabla f(y_j)\|_{H_{r(j)}^{-1}}^2 + \text{①} + \frac{10\eta^2 T}{\lambda\gamma}\left[\frac{\|2\boldsymbol{\sigma}\|_{2\alpha}^{2\alpha}}{\rho^{2(\alpha-1)}} + \frac{\sigma^2}{M}\right] \\
&\quad + \underbrace{\frac{160\eta}{\lambda}\frac{20(\frac{\eta L}{\lambda})^2}{(1-\beta_1)(\sqrt{\beta_2}-\beta_1)}\sum_{j=0}^t\mathbb{E}_m\left[\|\widehat{g_j^m}-\nabla f(x_j^m)\|^2 - \mathbb{E}_j\left[\|\widehat{g_j^m}-\nabla f(x_j^m)\|^2\right]\right]}_{\text{③: martingale}} \\
&\quad + \frac{50\eta T}{\lambda}\cdot\frac{\eta^2 L^2\sigma^2}{\lambda^2}\left(KA + \frac{64}{(1-\beta_1)(\sqrt{\beta_2}-\beta_1)}\right) + \frac{60\eta T}{\lambda}\frac{\|2\boldsymbol{\sigma}\|_{2\alpha}^{2\alpha}}{\rho^{2(\alpha-1)}} + \text{②} \\
&\le f_\gamma^\lambda(x_0) - \frac{\eta}{12}\sum_{j=0}^t \|\nabla f(y_j)\|_{H_{r(j)}^{-1}}^2 + \frac{10\eta^2\sigma^2}{\lambda\gamma M}T + \frac{60\eta T}{\lambda}\cdot\frac{\eta^2 L^2\sigma^2}{\lambda^2}KA + \frac{60\eta T}{\lambda}\frac{\|2\boldsymbol{\sigma}\|_{2\alpha}^{2\alpha}}{\rho^{2(\alpha-1)}} \\
&\quad + \text{①} + \text{②} + \text{③}.
\end{aligned}
$$
(D.141)

where in the third inequality, we apply $(1-\beta_2)B \le \dfrac{\eta L}{\lambda}$.

For ①, define

$$
\theta_j = \begin{cases} \dfrac{10\eta^2}{\lambda\gamma}\left[\left\|\mathbb{E}_m[\widehat{g_j^m} - \mathbb{E}_j[\widehat{g_j^m}]]\right\|^2 - \mathbb{E}_j\left[\left\|\mathbb{E}_m[\widehat{g_j^m} - \mathbb{E}_j[\widehat{g_j^m}]]\right\|^2\right]\right], & \text{if event } E_j \text{ holds,} \\ 0, & \text{otherwise.} \end{cases}
$$
(D.142)

Then event $E_t$ implies $\text{①} = \sum_{j=0}^t \theta_j$ and notice that

$$
|\theta_j| \le \frac{10\eta^2}{\lambda\gamma}\cdot 4\rho^2 d = \frac{40\eta^2\rho^2 d}{\lambda\gamma} \overset{def}{=} c,
$$
(D.143)

$$
\text{Var}_j(\theta_j) \le \left(\frac{10\eta^2}{\lambda\gamma}\right)^2\mathbb{E}_j\left[\|\mathbb{E}_m[\widehat{g_j^m} - \mathbb{E}_j[\widehat{g_j^m}]]\|^2\right]^2 \overset{\text{Lemma B.3}}{\le} 1600\left(\frac{\eta^2\sigma^2}{\lambda\gamma M}\right)^2.
$$
(D.144)

Let $b = \Delta/4$, $V = 1600T\left(\frac{\eta^2\sigma^2}{\lambda\gamma M}\right)^2$. Then by Lemma B.1, $|\sum_{j=0}^{t}\theta_j| \le b$ with probability no less than

$$1 - 2\exp\left(-\frac{b^2}{2V + 2cb/3}\right) \ge 1 - \frac{\delta}{12T}. \tag{D.145}$$

For ③, define

$$\xi_j = \begin{cases} \frac{160\eta}{\lambda}\frac{20(\frac{\eta L}{\lambda})^2}{(1-\beta_1)(\sqrt{\beta_2}-\beta_1)}\left(\mathbb{E}_m\left[\|\widehat{g_j^m} - \nabla f(x_j^m)\|^2 - \mathbb{E}_j[\|\widehat{g_j^m} - \nabla f(x_j^m)\|^2]\right]\right), & \text{if event } E_j \text{ holds,} \\ 0, & \text{otherwise.} \end{cases} \tag{D.146}$$

Note that

$$|\xi_j| \le \frac{160\eta}{\lambda}\frac{20(\frac{\eta L}{\lambda})^2}{(1-\beta_1)(\sqrt{\beta_2}-\beta_1)}\cdot 4\rho^2 d \stackrel{def}{=} c \tag{D.147}$$

$$\mathrm{Var}_j(\xi_j) \le \left(\frac{160\eta}{\lambda}\frac{20(\frac{\eta L}{\lambda})^2}{(1-\beta_1)(\sqrt{\beta_2}-\beta_1)}\right)^2\frac{\mathbb{E}_j\mathbb{E}_m\|\widehat{g_j^m} - \nabla f(x_j^m)\|^4}{M}$$

$$\le \left(\frac{160\eta}{\lambda}\frac{20(\frac{\eta L}{\lambda})^2}{(1-\beta_1)(\sqrt{\beta_2}-\beta_1)}\right)^2\frac{\sigma^4}{M}. \tag{D.148}$$

Let $b = \Delta/4$, $V = \left(\frac{160\eta}{\lambda}\frac{20(\frac{\eta L}{\lambda})^2}{(1-\beta_1)(\sqrt{\beta_2}-\beta_1)}\right)^2\frac{T\sigma^4}{M}$. Then by Lemma B.1, $|\sum_{j=0}^{t}\xi_j| \le b$ with probability no less than

$$1 - 2\exp\left(-\frac{b^2}{2V + 2cb/3}\right) \ge 1 - \frac{\delta}{12T}. \tag{D.149}$$

For ②, define

$$\zeta_j = \begin{cases} \frac{1 + \eta/4\gamma}{\gamma(\eta^{-1} + \gamma^{-1} - \tau/\lambda)}\left\langle \overline{z}_j - \eta H_{r(j)}^{-1}\nabla f(\overline{z}_j) - y_j, \mathbb{E}_m[\mathbb{E}_j[\widehat{g_j^m}] - \widehat{g_j^m}]\right\rangle, & \text{if event } E_j \text{ holds,} \\ 0, & \text{otherwise.} \end{cases} \tag{D.150}$$

Then event $E_t$ implies ② $= \sum_{j=0}^{t}\zeta_j$ and notice that by Lemma D.4,

$$\|\overline{z}_j - \eta H_{r(j)}^{-1}\nabla f(\overline{z}_j) - y_j\|^2 \le \frac{\left\|H_{r(j)}(\overline{z}_j - y_j) - \eta\nabla f(\overline{z}_j)\right\|^2_{H_{r(j)}^{-1}}}{\lambda}$$

$$\le \frac{\gamma^2\|\nabla f_\gamma^{H_{r(j)}}(\overline{z}_j)\|^2_{H_{r(j)}^{-1}}}{\lambda}$$

$$\le \frac{2\gamma\Delta}{\lambda}. \tag{D.151}$$

Therefore,

$$|\zeta_j| \le \frac{2\eta}{\gamma}\cdot\sqrt{\frac{2\gamma\Delta}{\lambda}}\cdot 2\rho\sqrt{d} = 4\eta\rho\sqrt{\frac{2\Delta d}{\gamma\lambda}} \stackrel{def}{=} c, \tag{D.152}$$

$$\mathrm{Var}_j(\zeta_j) \le \left(\frac{2\eta}{\gamma}\right)^2\cdot\frac{\gamma^2}{\lambda}\|\nabla f(y_j)\|^2_{H_{r(j)}^{-1}}\cdot\frac{\sigma^2}{M} \le \frac{4\eta^2\sigma^2}{\lambda M}\|\nabla f(y_j)\|^2_{H_{r(j)}^{-1}}. \tag{D.153}$$

Let $b = \Delta/4$, $V = \frac{100\eta\sigma^2\Delta}{\lambda M}$. Then by Lemma B.1,

$$\mathbb{P}\left\{|\sum_{j=0}^{t}\zeta_j| > b \text{ and } \sum_{j=0}^{t}\mathrm{Var}_j(\zeta_j) \le V\right\} \le 2\exp\left(-\frac{b^2}{2V + 2cb/3}\right) \le \frac{\delta}{12T}. \tag{D.154}$$

Note that by Lemma D.4 and event $E_t$,

$$\|\nabla f(y_t)\|^2_{H^{-1}_{r(t)}} \leq \frac{2}{\gamma}(f^{H_{r(t)}}_\gamma(\overline{z}_t) - \min f^\lambda_\gamma) \leq \frac{4\Delta}{\gamma}. \tag{D.155}$$

$$\sum_{j=0}^{t} \text{Var}_j(\zeta_j) \leq \frac{4\eta^2\sigma^2}{\lambda M}\sum_{j=0}^{t}\|\nabla f(y_j)\|^2_{H^{-1}_{r(j)}} \leq \frac{4\eta^2\sigma^2}{\lambda M}\cdot(\frac{24\Delta}{\eta}+\frac{4\Delta}{\gamma}) \leq V. \tag{D.156}$$

Therefore, combining ①, ②, ③, with probability no less than $\mathbb{P}(E_{t,3}) - 3\cdot\frac{\delta}{12T}$, event $E_{t,3}$ holds and $|\sum_{j=0}^{t}\zeta_j| \leq \frac{\Delta}{4}, |\sum_{j=0}^{t}\theta_j| \leq \frac{\Delta}{4}, |\sum_{j=0}^{t}\xi_j| \leq \frac{\Delta}{4}$. These implies

$$f^{H_{r(t+1)}}_\gamma(\overline{z}_{t+1}) - \min f^\lambda_\gamma \leq \frac{7}{4}\Delta - \frac{\eta}{12}\sum_{j=0}^{t}\|\nabla f(y_j)\|^2_{H^{-1}_{r(j)}} + \frac{10\eta^2\sigma^2}{\lambda\gamma M}T + \frac{60\eta T}{\lambda}\cdot\frac{\eta^2 L^2\sigma^2}{\lambda^2}KA + \frac{60\eta T}{\lambda}\frac{\|2\boldsymbol{\sigma}\|^{2\alpha}_{2\alpha}}{\rho^{2(\alpha-1)}}$$

$$\leq 2\Delta - \frac{\eta}{12}\sum_{j=0}^{t}\|\nabla f^{H_{r(j)}}_\gamma(\overline{z}_j)\|^2_{H^{-1}_{r(j)}}. \tag{D.157}$$

In the last inequality, we apply

$$\frac{10\eta^2\sigma^2}{\lambda\gamma M}T \leq \frac{\Delta}{12}, \quad \frac{60\eta}{\lambda}T\cdot\frac{\eta^2 L^2\sigma^2}{\lambda^2}KA \leq \frac{\Delta}{12}, \quad \frac{60\eta T}{\lambda}\frac{\|2\boldsymbol{\sigma}\|^{2\alpha}_{2\alpha}}{\rho^{2(\alpha-1)}} \leq \frac{\Delta}{12} \tag{D.158}$$

Therefore, we can conclude that $\mathbb{P}(E_{t+1}) \geq \mathbb{P}(E_{t,3}) - \frac{\delta}{4T}$. $\qquad\square$

**Lemma D.12.** *Define* $\Lambda_t := \sum_{j=0}^{t-1} a_{t,j}\|\overline{x}_j - \overline{x}_{j+1}\|^2$ *where* $a_{t,j} := \beta_1^{t-j-1}(t-j+\frac{\beta_1}{1-\beta_1})$. *Under the same conditions in Lemma D.11, event* $E_{t,3}$ *implies*

$$\Lambda_t \leq \left(1-\frac{(1-\beta_1)^2}{2}\right)\Lambda_{t-1} + \frac{64\eta^2}{1-\beta_1}\|\nabla f(\overline{z}_t)\|^2_{H^{-2}_r}$$

$$+ \frac{36\eta^2}{\lambda^2}(1-\beta_1)\sum_{j=rK}^{t-1}\beta_1^{t-j-1}\left[\frac{\eta^2 L^2\sigma^2}{\lambda^2}KA + \mathbb{E}_m\|\widehat{g^m_j} - \nabla f(x^m_j)\|^2\right]. \tag{D.159}$$

*Proof.* By the update rule, it always holds that

$$\|\overline{z}_t - \overline{x}_t\|^2 = (\frac{\beta_1}{1-\beta_1})^2\|\overline{x}_t - \overline{x}_{t-1}\|^2. \tag{D.160}$$

By AM-GM inequality and event $E_{t,1}$,

$$\|\overline{x}_t - \overline{x}_{t-1}\|^2 = \eta^2\|\mathbb{E}_m(H^m_{t-1})^{-1}u^m_{t-1}\|^2$$

$$\leq 2\eta^2\|\mathbb{E}_m(H^m_{t-1})^{-1}\overline{u}_{t-1}\|^2 + \frac{2\eta^2}{\lambda^2}\mathbb{E}_m\|u^m_{t-1} - \overline{u}_{t-1}\|^2 \tag{D.161}$$

$$\leq 4\eta^2\|\mathbb{E}_m H^{-1}_r\overline{u}_{t-1}\|^2 + \frac{2\eta^2}{\lambda^2}\mathbb{E}_m\|u^m_{t-1} - \overline{u}_{t-1}\|^2.$$

Event $E_{t,1}$ implies $z_j^m, x_j^m \in \mathbf{conv}(\mathbf{B}_{R_0}(\Omega))$ for all $j \leq t$ and thus

$$
\begin{aligned}
\mathbb{E}_m \|u_{t-1}^m - \overline{u}_{t-1}\|^2 &\leq (1-\beta_1) \sum_{j=rK}^{t-1} \beta_1^{t-j-1} \mathbb{E}_m[\|\widehat{g_j^m} - \overline{g}_j\|^2] \\
&\leq 2(1-\beta_1) \sum_{j=rK}^{t-1} \beta_1^{t-j-1} \mathbb{E}_m \left[ \|\widehat{g_j^m} - \nabla f(x_j^m)\|^2 + \|\nabla f(x_j^m) - \mathbb{E}_m \nabla f(x_j^m)\|^2 \right] \\
&\leq 2(1-\beta_1) \sum_{j=rK}^{t-1} \beta_1^{t-j-1} \mathbb{E}_m \left[ L^2 \|x_j^m - \overline{x}_j\|^2 + \|\widehat{g_j^m} - \nabla f(x_j^m)\|^2 \right] \\
&\leq 2(1-\beta_1) \sum_{j=rK}^{t-1} \beta_1^{t-j-1} \left[ \frac{\eta^2 L^2 \sigma^2}{\lambda^2} KA + \mathbb{E}_m \|\widehat{g_j^m} - \nabla f(x_j^m)\|^2 \right].
\end{aligned}
\tag{D.162}
$$

$$
\begin{aligned}
\frac{1}{4} \|\overline{u}_{t-1}\|_{H_r^{-2}}^2 &\leq \left\| (1-\beta_1) \sum_{j=0}^{t-1} \beta_1^{t-j-1} \nabla f(\overline{x}_t) \right\|_{H_r^{-2}}^2 + \left\| (1-\beta_1) \sum_{j=0}^{t-1} \beta_1^{t-j-1} [\nabla f(\overline{x}_j) - \nabla f(\overline{x}_t)] \right\|_{H_r^{-2}}^2 \\
&\quad + \left\| (1-\beta_1) \sum_{j=0}^{t-1} \beta_1^{t-j-1} \mathbb{E}_m[\nabla f(x_j^m) - \nabla f(\overline{x}_j)] \right\|_{H_r^{-2}}^2 + \left\| (1-\beta_1) \sum_{j=0}^{t-1} \beta_1^{t-j-1} \mathbb{E}_m[\widehat{g_j^m} - \nabla f(x_j^m)] \right\|_{H_r^{-2}}^2 \\
&\leq \|\nabla f(\overline{x}_t)\|_{H_r^{-2}}^2 + \frac{(1-\beta_1)}{\lambda^2} \sum_{j=0}^{t-1} \beta_1^{t-j-1} L^2 \|\overline{x}_j - \overline{x}_t\|^2 \\
&\quad + \frac{(1-\beta_1)}{\lambda^2} \sum_{j=0}^{t-1} \beta_1^{t-j-1} \left[ \|\mathbb{E}_m[\widehat{g_j^m} - \nabla f(x_j^m)]\|^2 + \|\mathbb{E}_m[\nabla f(x_j^m) - \nabla f(\overline{x}_j)]\|^2 \right] \\
&\leq 2 \|\nabla f(\overline{z}_t)\|_{H_r^{-2}}^2 + \frac{2L^2}{\lambda^2} \|\overline{z}_t - \overline{x}_t\|^2 + \frac{(1-\beta_1)}{\lambda^2} \sum_{j=0}^{t-1} \beta_1^{t-j-1} L^2 (t-j) \sum_{i=j}^{t-1} \|\overline{x}_i - \overline{x}_{i+1}\|^2 \\
&\quad + \frac{(1-\beta_1)}{\lambda^2} \sum_{j=0}^{t-1} \beta_1^{t-j-1} \left[ \|\mathbb{E}_m[\widehat{g_j^m} - \nabla f(x_j^m)]\|^2 + \|\mathbb{E}_m[\nabla f(x_j^m) - \nabla f(\overline{x}_j)]\|^2 \right] \\
&\leq 2 \|\nabla f(\overline{z}_t)\|_{H_r^{-2}}^2 + \frac{2L^2}{\lambda^2} \|\overline{z}_t - \overline{x}_t\|^2 + \frac{L^2}{\lambda^2} \sum_{j=0}^{t-1} a_{t,j} \|\overline{x}_j - \overline{x}_{j+1}\|^2 \\
&\quad + \frac{(1-\beta_1)}{\lambda^2} \sum_{j=0}^{t-1} \beta_1^{t-j-1} \left[ \|\mathbb{E}_m[\widehat{g_j^m} - \nabla f(x_j^m)]\|^2 + \|\mathbb{E}_m[\nabla f(x_j^m) - \nabla f(\overline{x}_j)]\|^2 \right].
\end{aligned}
\tag{D.163}
$$

Here $a_{t,j} := \beta_1^{t-j-1}(t - j + \frac{\beta_1}{1-\beta_1})$. For $j \leq t - 2$, we have $a_{t,j} \leq \beta_1(2 - \beta_1)a_{t-1,j}$. Since $\Lambda_t = \sum_{j=0}^{t-1} a_{t,j}\|\overline{x}_j - \overline{x}_{j+1}\|^2$, we conclude that

$$\|\overline{x}_t - \overline{x}_{t-1}\|^2 \leq 64\eta^2 \left[ \|\nabla f(\overline{z}_t)\|^2_{H_r^{-2}} + \frac{L^2}{\lambda^2}\Lambda_{t-1} \right] + \frac{4\eta^2}{\lambda^2}(1 - \beta_1) \sum_{j=rK}^{t-1} \beta_1^{t-j-1} \left[ \frac{\eta^2 L^2 \sigma^2}{\lambda^2}KA + \mathbb{E}_m\|\widehat{g_j^m} - \nabla f(x_j^m)\|^2 \right]$$

$$+ \frac{32\eta^2(1-\beta_1)}{\lambda^2} \sum_{j=0}^{t-1} \beta_1^{t-j-1} \left[ \|\mathbb{E}_m[\widehat{g_j^m} - \nabla f(x_j^m)]\|^2 + \|\mathbb{E}_m[\nabla f(x_j^m) - \nabla f(\overline{x}_j)]\|^2 \right]$$

$$\leq 64\eta^2 \left[ \|\nabla f(\overline{z}_t)\|^2_{H_r^{-2}} + \frac{L^2}{\lambda^2}\Lambda_{t-1} \right]$$

$$+ \frac{36\eta^2}{\lambda^2}(1 - \beta_1) \sum_{j=rK}^{t-1} \beta_1^{t-j-1} \left[ \frac{\eta^2 L^2 \sigma^2}{\lambda^2}KA + \mathbb{E}_m\|\widehat{g_j^m} - \nabla f(x_j^m)\|^2 \right],$$

$$\tag{D.164}$$

and

$$\Lambda_t \leq \beta_1(2 - \beta_1)\Lambda_{t-1} + \frac{1}{1-\beta_1}\|\overline{x}_t - \overline{x}_{t-1}\|^2. \tag{D.165}$$

This completes the proof. $\qquad\square$

## D.5 FURTHER DISCUSSION

**Compared to other results under centralized weakly convex setting.** Theorem D.2 can reduce to Minibatch Adam (by substituting $M, K$ with 1 and $\sigma$ with $\frac{\sigma}{\sqrt{MK}}$ in (D.27) (Petrov, 1992)), and the convergence guarantee is

$$\frac{\lambda}{R} \sum_{r=0}^{R-1} \|\nabla f_\gamma^{H_r}(\overline{z}_r)\|^2_{H_r^{-1}} = \tilde{\mathcal{O}} \left( \frac{L\Delta}{R} + \sqrt{\frac{\lambda\Delta\sigma^2}{\gamma MKR}} + \left( \frac{L\Delta\sigma^{\frac{\alpha}{\alpha-1}}}{(MK)^{\frac{\alpha}{2(\alpha-1)}}R} \right)^{\frac{2(\alpha-1)}{3\alpha-2}} \right). \tag{D.166}$$

Therefore, in centralized setting with iteration number $R$ and batch size 1, our guarantee for squared norm of gradient of Moreau envelope is

$$\tilde{\mathcal{O}} \left( \frac{L\Delta}{R} + \sqrt{\frac{\lambda\Delta\sigma^2}{\gamma R}} + \left( \frac{L\Delta\sigma^{\frac{\alpha}{\alpha-1}}}{R} \right)^{\frac{2(\alpha-1)}{3\alpha-2}} \right). \tag{D.167}$$

The last term is induced by the bias of clipped gradient. For simplicity, let $R \gtrsim \frac{L\Delta}{\sigma^2}$ so that the last term can be dominated by the first term. Then we obtain

$$\tilde{\mathcal{O}} \left( \frac{L\Delta}{R} + \sqrt{\frac{\lambda\Delta\sigma^2}{\gamma MKR}} \right). \tag{D.168}$$

In the previous literature of weakly convex function (Davis & Drusvyatskiy, 2019; Alacaoglu et al., 2020; Mai & Johansson, 2021), $f$ is typically non-smooth and stochastic gradient is assumed to have bounded second order moment. This is weaker than the smoothness assumption but stronger than that of noise with bounded moment. There are a few existing results for smooth objective (Davis & Drusvyatskiy, 2019; Mai & Johansson, 2020; Deng & Gao, 2021), but they set $\tau = L$. Overall, our result is the first convergence guarantee for smooth weakly convex function with $\tau \ll L$ and bounded-moment noise.

**Dependence on $\beta_2$.** The default setting of $\beta_2$ in the Adam optimizer of PyTorch is 0.999, which is a constant close to 1. Adam with small $\beta_2$ has been shown to diverge in some examples (Reddi et al., 2019). However, if it is too close to 1, *e.g.*, $\beta_2 \geq 1 - \mathcal{O}(T^{-1})$, then the denominator would

be too stagnant to provide adaptivity. Therefore, to derive a proper range for $\beta_2$ is crucial in the theoretical analysis of Adam.

On the other hand, $\beta_2$ is notoriously difficult to handle even under centralized setting. In finite sum case, Zou et al. (2019) assumes $\beta_2 \geq 1 - \mathcal{O}(T^{-1})$. Shi et al. (2020) suggests that $\beta_2 \geq 1 - \mathcal{O}(n^{-3.5})$ suffices, where $n$ is sample size. Zhang et al. (2022b) claims Adam can converge to the neighborhood of stationary points with constant radius if $\beta_2 \geq 1 - \mathcal{O}(n^{-3})$. Further, Wang et al. (2022) shows Adam can converge to stationary points if $\beta_2$ is sufficiently close to 1, but the explicit bound is missing. In streaming data case, Défossez et al. (2020) shows $\beta_2$ can be a constant but relies on the bounded gradient assumption. (Li et al., 2024c) suggests $\beta_2 \geq 1 - \tilde{\mathcal{O}}(T^{-\frac{1}{2}})$.

As for distributed setting, works discussing the range of $\beta_2$ are much fewer. Our theory requires $\beta_2 \geq 1 - \tilde{\mathcal{O}}(K^{-\frac{3}{2}}R^{-\frac{1}{2}})$. For distributed Adam, Karimireddy et al. (2020a); Zhao et al. (2022) fixed the denominator during local iterations and thus did not discuss the range of $\beta_2$. To the best of our knowledge, our result is the first one to show the $\tilde{\mathcal{O}}(R^{-\frac{1}{2}})$ dependence with respect to $R$. Nevertheless, it is an interesting question to improve the dependence on $K$. Since $K$ is usually a constant in practice, our results suggest $\beta_2 \geq 1 - \tilde{\mathcal{O}}(\mathcal{R}^{-\frac{1}{2}})$ in essence. Still, we believe that the dependence on $K$ has room for improvement. We leave this for future work.

**Dependence on $\lambda$.** $\lambda$ in the denominator of Adam is aimed to avoid numerical instability, and usually a small constant in practice. Note $H_r = \mathbf{diag}(\sqrt{V_r + \lambda^2})$ and $v_r$ is the EMA of squared past gradients. Informally, $v_r$ vanishes as $r$ grows and thus $H_r$ would gradually reduce to $\lambda I_d$. In the worst case, $H_r$ can be bounded by a constant. In conclusion, the LHS in (4.9) is roughly the averaged squared gradient norm if $\lambda$ is not too small. It is worth noting that $\lambda$ can be arbitrarily small or even 0 in (Défossez et al., 2020; Wang et al., 2022; 2024). However, their results all depend on $\mathbf{poly}(d)$. It is still an interesting question to get dimension-free result with small $\lambda$.

**Dependence on $\beta_1$.** The default setting of $\beta_1$ in PyTorch is 0.9, a constant away from 0 and 1. In the centralized setting, Li et al. (2024c) requires $\beta_1 = 1 - \mathcal{O}(T^{-\frac{1}{2}})$ to converge, which is too large. Défossez et al. (2020) shows $\mathcal{O}\left((1-\beta_1)^{-1}\right)$, which is the state of the art result to the best of our knowledge. However, it relies on the bounded gradient assumption. Regarding the dependence on $\beta_1$, our convergence rate in Theorem D.1 suggests $\mathcal{O}\left((1-\beta_1)^{-2}\right)$. Although it also supports any constant choice of $\beta_1$, we leave the exploration of better dependence for future work.

# E    FAILURE OF STANDARD SGD WITH HEAVY-TAILED NOISE

The convergence of standard SGD in high probability is widely studied. If we assume the noises are light-tailed, *e.g.*, sub-exponential, sub-gaussian, then SGD can get high probability bound depending on $\log \frac{1}{\delta}$. However, if only finite variance is assumed, Sadiev et al. (2023) has shown that standard SGD fails to get a high probability bound having logarithmic dependence on $\frac{1}{\delta}$. In fact, this claim is still valid when the stochastic noises only have finite $\alpha$th-moment, as shown in Theorem E.1 below. Therefore, gradient clipping is necessary to get the $\log \frac{1}{\delta}$ bound.

**Theorem E.1.** *For any $\varepsilon > 0$, $\delta \in (0,1)$, and SGD with the iteration number $T$ and learning rate $\eta$, there exists an 1D-problem satisfying Assumption 1, 2, 3, 4, with $\Omega = \mathbb{R}$ and $L = \mu$, such that, if $0 < \eta \leq 1/L$, then*

$$\mathbb{P}\{f(x_T) - f_* \geq \varepsilon\} \leq \delta \implies T = \tilde{\Omega}\left(\frac{\sigma}{\delta^{1/\alpha}}\sqrt{\frac{L}{\varepsilon}}\right). \tag{E.1}$$

*Proof.* We follow the construction of the counter example in Sadiev et al. (2023). To prove the above theorem, we consider a simple 1D-problem $f(x) = Lx^2/2$. It is easy to see that the considered problem is $L$-strongly convex, $L$-smooth, and has optimum at $x_* = 0$. We construct the noise in an adversarial way with respect to the parameters of the SGD. Concretely, the noise depends on the

number of iterates $t$, learning rate $\eta$, target precision $\varepsilon$, the starting point $x_0$, and the moment bound $\sigma$ such that

$$\nabla F(x_t; \xi_t) = Lx_t - \sigma\xi_t, \tag{E.2}$$

where

$$\xi_t = \begin{cases} 0, & \text{if } t < T - 1 \text{ or } (1 - \eta L)^T |x_0| > \sqrt{\dfrac{2\varepsilon}{L}}, \\ \begin{cases} -A, & \text{with probability } \dfrac{1}{2A^\alpha}, \\ 0, & \text{with probability } 1 - \dfrac{1}{A^\alpha}, \\ A, & \text{with probability } \dfrac{1}{2A^\alpha}, \end{cases} & \text{otherwise} \end{cases} \tag{E.3}$$

where $A = \max\left\{\dfrac{2\sqrt{\frac{2\varepsilon}{L}}}{\eta\sigma}, 1\right\}$. We note that $\mathbb{E}[\xi_t] = 0$ and $\mathbb{E}[\nabla F(x_t; \xi_t)] = \nabla f(x_t)$. Furthermore,

$$\mathbb{E}[|\xi_t|^\alpha] \le \frac{1}{2A^\alpha} A^\alpha + \frac{1}{2A^\alpha} A^\alpha = 1, \tag{E.4}$$

which implies that Assumption 3 holds.

We are interested in the situation when

$$\mathbb{P}\{f(x_T) - f_* \ge \varepsilon\} \le \delta, \tag{E.5}$$

for $\delta \in (0, 1)$. We first prove that this implies $(1 - \eta L)^T |x_0| \le \sqrt{\dfrac{2\varepsilon}{L}}$. To do that we proceed by contradiction and assume that

$$(1 - \eta L)^T |x_0| > \sqrt{\frac{2\varepsilon}{L}}. \tag{E.6}$$

By construction, this implies that $\xi_t = 0, \forall t \in \{0, \cdots, T - 1\}$. This, in turn, implies that $x_T = (1 - \eta L)^T x_0$, and further, by (E.6) that

$$\mathbb{P}\{f(x_T) - f_* \ge \varepsilon\} = \mathbb{P}\left\{|x_T| \ge \sqrt{\frac{2\varepsilon}{L}}\right\} = 1.$$

Thus, the contradiction shows that $(1 - \eta L)^T |x_0| \le \sqrt{\dfrac{2\varepsilon}{L}}$. Using (E.3), we obtain

$$f(x_T) - f_* = \frac{L}{2}\left[(1 - \eta L)^T x_0 + \eta\sigma\xi_{T-1}\right]^2. \tag{E.7}$$

Furthermore,

$$\begin{aligned} \mathbb{P}\{f(x_T) - f_* \ge \varepsilon\} &= \mathbb{P}\left\{\left|(1 - \eta L)^T x_0 + \eta\sigma\xi_{T-1}\right| \ge \sqrt{\frac{2\varepsilon}{L}}\right\} \\ &= \mathbb{P}\left\{|\eta\sigma\xi_{T-1}| \ge \sqrt{\frac{2\varepsilon}{L}} + (1 - \eta L)^T |x_0|\right\} \\ &\ge \mathbb{P}\left\{|\eta\sigma\xi_{T-1}| \ge 2\sqrt{\frac{2\varepsilon}{L}}\right\} \\ &= \mathbb{P}\left\{|\xi_{T-1}| \ge \frac{2\sqrt{\frac{2\varepsilon}{L}}}{\eta\sigma}\right\}. \end{aligned} \tag{E.8}$$

Now if $\dfrac{2\sqrt{\frac{2\varepsilon}{L}}}{\eta\sigma} < 1$ then $A = 1$. Therefore,

$$1 = \mathbb{P}\left\{|\xi_{T-1}| \ge \frac{2\sqrt{\frac{2\varepsilon}{L}}}{\eta\sigma}\right\} \le \mathbb{P}\{f(x_T) - f_* > \varepsilon\} \le \delta, \tag{E.9}$$

yielding contradiction, which implies that $\dfrac{2\sqrt{\frac{2\varepsilon}{L}}}{\eta\sigma} \geq 1$, *i.e.*, $\eta \leq 2\sqrt{\dfrac{2\varepsilon}{L\sigma^2}}$. In this case, $A = \dfrac{2\sqrt{\frac{2\varepsilon}{L}}}{\eta\sigma}$ and we have

$$\delta \geq \mathbb{P}\left\{f(x_T) - f_* \geq \varepsilon\right\} \geq \mathbb{P}\left\{|\xi_{T-1}| \geq \frac{2\sqrt{\frac{2\varepsilon}{L}}}{\eta\sigma}\right\} = \frac{1}{A^\alpha}. \tag{E.10}$$

This implies that $\eta \leq \dfrac{2\delta^{1/\alpha}}{\sigma}\sqrt{\dfrac{2\varepsilon}{L}}$. Combining this inequality with $T \geq \dfrac{1}{2\eta L}\log\dfrac{Lx_0^2}{2\varepsilon}$ yields

$$T = \Omega\left(\frac{\sigma}{\delta^{1/\alpha}}\sqrt{\frac{L}{\varepsilon}}\log\frac{Lx_0^2}{2\varepsilon}\right). \tag{E.11}$$

This concludes the proof. $\qquad\square$

