# OpenReview forum: "Convergence of Distributed Adaptive Optimization with Local Updates"
_ICLR.cc/2025/Conference — ICLR 2025 Poster_

### Official Review · Reviewer_3h9b · 2024-10-23

**Soundness:** 3
**Presentation:** 3
**Contribution:** 3
**Rating:** 8
**Confidence:** 2

**Summary:**

The authors consider problems from distributed optimization and federated learning, where communication can be a bottleneck. In this setup, they analyze the celebrated Local SGD method with momentum, clipping, and Adam steps. In this context, they provide new theoretical convergence rates under heavy-tailed distribution assumptions that hold with high probability.

**Strengths:**

The considered problem is important in modern optimization. Using local steps is one of the possible strategies to mitigate the communication burden. The assumptions and goals of the paper are very challenging. I have not checked the proofs in detail (that is why I choose confidence 2), but the final theorems seem to be reasonable and consistent with those in previous papers. Overall, this paper appears to be a solid and technical mathematical work that extends the known results of Local SGD.

**Weaknesses:**

At the same time, I want to point to some weaknesses:
1. The choice of Assumption 3 is unusual. In most previous works, this assumption holds with $1 < \alpha \leq 2,$ while the authors choose $\alpha \geq 4,$ which is a much stronger assumption. What is the reason for that? The reason should be discussed in the main part.
2. In many places of the paper (abstract, contributions, ...), there is a claim that Local SGDM outperforms the minibatch counterparts. But this is not true in general. [1] point to some regimes when Local SGD is worse. Moreover, (4.5) can be much better than (4.4) if $\sigma$ and $M$ are large. In this regime, the third term in (4.4) can dominate.
3. I agree that when $K$ is large, Local SGDM can outperform Minibatch SGDM. However, looking at (4.4), due to the third term, it is clear that taking $K \to \infty$ is a suboptimal choice (we can't just take $K \to \infty$ because one global iteration would take an infinitely long time), so there should be an optimal choice of $K^*,$ and I'm not convinced that for this choice of $K^*$ Local SGDM will be better than Minibatch SGDM.
4. The rate (4.9) of Local ADAM, in the way it is presented now, does not provide meaningful convergence guarantees due to the norm $||\cdot||_{H_r^{-1}}.$ What happens if the maximum eigenvalues of $H_r$ are huge? I think the authors should bound $H_r$ somehow to get a a more meaningful rate.

Conclusion: Despite all these weaknesses, I think that the authors did a very nontrivial and technical mathematical work, which I appreciate. However, due to time limits and technicalities, I cannot check the proofs in detail, so I will choose a low confidence.

[1]: Woodworth, Blake, et al. "Is local SGD better than minibatch SGD?." International Conference on Machine Learning. PMLR, 2020.

**Questions:**

(see weaknesses)

---

> ### Author Response · Authors · 2024-11-19
>
> Thanks for recognizing that our mathematical work is a solid extension to previous results! We address your concerns as follows.
>
> **W1**: The assumption on $\alpha\geq 4$ is uncommon.
>
> **A1**: As mentioned in Remark 1, in most previous works on high probability convergence, the noise assumptions can be divided into two categories: light tail with bounded exponential moment (sub-gaussian, sub-exponential, bounded), or finite $\alpha$-moment with $1<\alpha\leq 2$. We remark that ***although $\alpha\geq 4$ is rarely used in the literature, it is still substantially weaker than the sub-gaussian type assumption***. And as discussed in Appendix E, if assuming only the polynomial moment is bounded, SGD fails to get a logarithmic dependence on $1/\delta$, where $\delta$ is the confidence level. Therefore, ***$\alpha\geq 4$ is essentially a type of heavy tailed noise assumption.***
>
> In our settings, the assumption $\alpha\geq 4$ is made to make the results much more clean and concise. In the proof, we only use this noise assumption to bound the variance of certain random variables. For instance, in eq (C.44), we need to bound the 4-th moment $\mathbb{E}\|\nabla f(x)-\text{clip}(g,\rho)\|^4$ where $g$ is the stochastic gradient. If $\alpha\geq 4$, then we can directly apply the noise assumption and get $\sigma^4$. On the other hand, if $1<\alpha< 4$, then for large $\rho$ this quantity can be bounded by $\mathcal{O}(\sigma^\alpha\rho^{4-\alpha})$ by Lemma B.2 (also refer to Lemma 5.1 in [1]). The subsequent analysis is similar but will lead to a worse dependence on $\rho$. Therefore, our techniques can be easily generalized to the case when $\alpha<4$, but the results will be much more messy, which is also beyond the scope of this paper to show the benefits of local updates in adaptive optimization.
>
> **W2**: Inaccurate claims on "Local SGDM outperforms the minibatch counterparts".
>
> **A2**: Thanks for the advice. The advantages of Local SGDM appear only in the large $M,K$ regime. The original statement is indeed inaccurate and we have adjusted it accordingly.
>
> **W3**: The optimal choice of K and the advantages of Local SGDM over mini-batch SGDM.
>
> **A3**: One doesn't need to take $K\rightarrow\infty$ to exhibit the adavantages of Local SGDM. It suffices to let $M, K$ large enough so that the noise term $\frac{\sigma}{\sqrt{MKR}}$ (or $\frac{\sigma^2}{\mu MKR}$ in strongly convex case) will be dominated. In this case, the convergence of Minibatch SGDM will be impeded by the first term in eq (4.5), which only depends on $R$. Moreover, since our convergence rate for SGDM is almost the same with SGD in [2], the discussion therein on the choice of $K$ is also valid in our setting. Hence, the theoretical adavantages of Local SGDM can be guaranteed in large $M$ and large $K$ regime.
>
> **W4**: The convergence criteria of Local Adam based on $\|\cdot\|_{H_r^{-1}}$ is not meaningful.
>
> **A4**: The matrix $H_r$ is indeed algorithmic-dependent and we leave the expression in Theorem 3 in the main text to in order to be consistent with previous papers. Nevertheless, in Theorem D.3 in appendix, we rigorously analyze the bound of $H_r$ and provide the convergence rate of euclidean norm of gradient, which will introduce an additional multiplier $(1+\frac{G_\infty+\sigma_\infty}{\lambda})$. From this we see that large $\lambda$ can indeed enhance the rate. However, we point out that the bound on $H_r$ there is very pessmistic and large $\lambda$ is also not aligned with common practice of Adam. We leave the impact of the choice of $\lambda$ as an interesting future work.
>
>
> [1] Sadiev, Abdurakhmon, et al. "High-probability bounds for stochastic optimization and variational inequalities: the case of unbounded variance." International Conference on Machine Learning. PMLR, 2023.
>
> [2] Woodworth, Blake, et al. "Is local SGD better than minibatch SGD?." International Conference on Machine Learning. PMLR, 2020.

---

### Official Review · Reviewer_ouZP · 2024-10-25

**Soundness:** 2
**Presentation:** 2
**Contribution:** 2
**Rating:** 6
**Confidence:** 4

**Summary:**

This paper analyzes the convergence rates of Adam/SGDM with local steps, showing that local steps can improve the convergence rates in the homogeneous setting.

**Strengths:**

* This paper analyzes the convergence rates of Local SGDM and Local Adam, showing that their convergence rates are better than Minibatch SGDM and Adam.
* Overall, the reviewer feels that the result shown in this paper is not very surprising, but it is solid.

**Weaknesses:**

* In Assumption 3, the authors assume $\alpha \geq 4$. The reviewer feels that this assumption is a bit different from the assumption commonly used in the existing literature. For instance, [1] used a similar assumption (see Assumption 1), while they called that the stochastic noise is "heavy-tailed" when $\alpha<2$. Although the authors claimed that it is easy to extend their analysis to the case where $\alpha>4$ in Remark 1, the reviewer feels that the authors should show the analysis with arbitrary $\alpha$ if this extension is really easy.
* In Theorem 1 and 2, the authors omit the dependence of $\beta_1$ by using $1 - \beta_1 = \Omega(1)$. Thus, we can not discuss the effect of $\beta_1$ on the convergence rate. The reviewer feels that it would be better to show the precise convergence rate, at least in the Appendix.
* Theorem 1 and 2 used the assumption that $\| x - x^\star \| \leq O( \| x_0 - x^\star\| )$. Thus, these theorems only hold when the set of feasible solutions is bounded. Similar to these assumptions, Theorem 3 used the assumption that $f(x) - f^\star$ is bounded.


## Reference
[1] Zhang et. al., Why are Adaptive Methods Good for Attention Models?, in NeurIPS 2020

**Questions:**

See the weakness section.

---

> ### Author Response · Authors · 2024-11-19
>
> We are glad the reviewer finds our results solid! We address the concerns as follows.
>
> **W1**: The assumption on $\alpha\geq 4$ is uncommon.
>
> **A1**: As mentioned in Remark 1, in most previous works on high probability convergence, the noise assumptions can be divided into two categories: light tail with bounded exponential moment (sub-gaussian, sub-exponential, bounded), or finite $\alpha$-moment with $1<\alpha\leq 2$. We remark that ***although $\alpha\geq 4$ is rarely used in the literature, it is still substantially weaker than the sub-gaussian type assumption***. And as discussed in Appendix E, if assuming only the polynomial moment is bounded, SGD fails to get a logarithmic dependence on $1/\delta$, where $\delta$ is the confidence level. Therefore, ***$\alpha\geq 4$ is essentially a type of heavy tailed noise assumption.***
>
> In our settings, the assumption $\alpha\geq 4$ is made to make the results much more clean and concise. In the proof, we only use this noise assumption to bound the variance of certain random variables. For instance, in eq (C.44), we need to bound the 4-th moment $\mathbb{E}\|\nabla f(x)-\text{clip}(g,\rho)\|^4$ where $g$ is the stochastic gradient. If $\alpha\geq 4$, then we can directly apply the noise assumption and get $\sigma^4$. On the other hand, if $1<\alpha< 4$, then for large $\rho$ this quantity can be bounded by $\mathcal{O}(\sigma^\alpha\rho^{4-\alpha})$ by Lemma B.2 (also refer to Lemma 5.1 in [1]). The subsequent analysis is similar but will lead to a worse dependence on $\rho$. Therefore, our techniques can be easily generalized to the case when $\alpha<4$, but the results will be much more messy, which is also beyond the scope of this paper to show the benefits of local updates in adaptive optimization.
>
> **W2**: The lack of clear dependence on $\beta_1$.
>
> **A2**: Thanks for the advice. We omit $\beta_1$ to make the results more concise. In fact we keep the dependence on $\beta_1$ in the main theorems in appendix (Theorem C.3, Theorem D.1), based on which we obtain the ultimate convergence rate by plugging in the parameters. We will also make this more clear in the main text.
>
> **W3**: The assumptions implies that the set of feasbile solutions are bounded.
>
> **A3**: Our Assumption 2,4,5 is mainly following the previous work ([1]). We admit that this type of assumptions is still restricted in some sense, but it is much more general than the commonly used global smoothness (convexity) assumption, which is the case when $\Omega=\mathbb{R}^d$. We hope future works can explore more relaxed assumptions.
>
> [1] Sadiev, Abdurakhmon, et al. "High-probability bounds for stochastic optimization and variational inequalities: the case of unbounded variance." International Conference on Machine Learning. PMLR, 2023.

---

> ### Comment · Reviewer_ouZP · 2024-11-22
>
> The reviewer thank the authors for their response.
>
> The reviewer agrees with Reviewer FvoJ and still believes that it would be better to add the results with $\alpha < 4$ if the authors would like to say "our analysis methods can be easily extended to the case where $\alpha < 4$" in Remark 1. The reviewer personally believes that the authors must not say it is easy without showing the results with $\alpha < 4$ since it might cause trouble for future studies that consider the case with $\alpha < 4$.

---

> > ### Author Response · Authors · 2024-11-22
> >
> > Thanks for your helpful feedback. We have adjusted the statement about "easily" in Remark 1 to "but our analysis methods can be extended to the case where $\alpha < 4$ with some additional technical computations". Please let us know if you have any other concerns.

---

### Official Review · Reviewer_jJw9 · 2024-11-01

**Soundness:** 3
**Presentation:** 3
**Contribution:** 3
**Rating:** 8
**Confidence:** 3

**Summary:**

The paper studies distributed adaptive algorithms with local updates in the homogeneous data regime, aiming to establish theoretical guarantees proving the benefits of local iterations in reducing communication complexity. Specifically, the authors analyze a distributed variant of Adam - Local Adam with gradient clipping, which also reduces to Local SGD with momentum (Local SGDM). They establish that both Local SGDM and Local Adam can outperform their minibatch SGDM / minibatch Adam in convex / weakly convex settings, respectively.

**Strengths:**

- The work is the first to offer high-probability bounds for distributed optimization algorithms with local steps.

- Some assumptions are relatively weak; for example, smoothness and (strong) convexity are required on a subset rather than the entire space.

- The first theoretical convergence guarantees showing that Local SGDM and Local Adam can outperform their minibatch versions in some regimes (large $M$ and $K$ regime, where $M$ is the number of clients and $K$ is the number of local steps).

- The paper is well-written and easy to follow. The authors clearly introduce the problem and the paper's contributions.

- Despite a few limitations, this work represents an important theoretical contribution to understanding the empirical success of Adam with local steps as observed in prior research.

**Weaknesses:**

- The paper addresses only the homogeneous data case, where all clients have access to the same data. Client drift from data heterogeneity - one of the main challenges for local training methods - is not explored.

- The noise assumptions are somewhat restrictive (see, e.g., [1]). Specifically, the authors assume a bounded $\alpha$-moment of the noise with $\alpha \geq 4$. In contrast, most works only assume this condition for $\alpha \in (1,2]$, and recent high-probability and in-expectation convergence rates have been obtained under the strictly weaker condition $\alpha \in (1,2)$ [2].

[1] Khaled, Ahmed, and Peter Richtárik. "Better theory for SGD in the nonconvex world." arXiv preprint arXiv:2002.03329 (2020).

[2] Puchkin, Nikita, Eduard Gorbunov, Nickolay Kutuzov, and Alexander Gasnikov. "Breaking the heavy-tailed noise barrier in stochastic optimization problems." In International Conference on Artificial Intelligence and Statistics, pp. 856-864. PMLR, 2024.

**Questions:**

- What are the advantages of the proposed algorithms over non-adaptive methods in terms of theoretical guarantees?

- The paper claims that "there is no end-to-end convergence guarantee for distributed adaptive algorithms with local iterations." However, this seems inaccurate. For instance, [3] introduces the Extrapolated SPPM algorithm, which replaces local SGD-type training with the computation of proximal terms, and [4] provides tighter convergence theory for the same algorithm. Both approaches use adaptive strategies based on gradient diversity and Polyak step sizes, and their results hold in heterogeneous data settings.

[3] Li, Hanmin, Kirill Acharya, and Peter Richtarik. "The Power of Extrapolation in Federated Learning." arXiv preprint arXiv:2405.13766 (2024).

[4] Anyszka, Wojciech, Kaja Gruntkowska, Alexander Tyurin, and Peter Richtárik. "Tighter Performance Theory of FedExProx." arXiv preprint arXiv:2410.15368 (2024).

---

> ### Author Response · Authors · 2024-11-19
>
> Thanks for regarding our work as an important theoretical contribution to understand the empirical success of Adam with local step! We address your concerns as follows.
>
> **W1**: The paper addresses only the homogeneous data case.
>
> **A1**: We'd like to mention that homogeneous case is practical for distributed training of large language models, where a huge dataset is often assigned to multiple computation nodes in a homogeneous way. The heterogeneous case is definitely another interesting topic but the analysis will also get more challenging as well. We hope this work can inspire more researches on the local updates in heterogenous case.
>
> **W2**: The noise assumptions on $\alpha\geq 4$ are restrictive compared to previous works.
>
> **A2**: As mentioned in Remark 1, in most previous works on high probability convergence, the noise assumptions can be divided into two categories: light tail with bounded exponential moment (sub-gaussian, sub-exponential, bounded), or finite $\alpha$-moment with $1<\alpha\leq 2$. We remark that ***although $\alpha\geq 4$ is rarely used in the literature, it is still substantially weaker than the sub-gaussian type assumption***. And as discussed in Appendix E, if assuming only the polynomial moment is bounded, SGD fails to get a logarithmic dependence on $1/\delta$, where $\delta$ is the confidence level. Therefore, ***$\alpha\geq 4$ is essentially a type of heavy tailed noise assumption.***
>
> In our settings, the assumption $\alpha\geq 4$ is made to make the results much more clean and concise. In the proof, we only use this noise assumption to bound the variance of certain random variables. For instance, in eq (C.44), we need to bound the 4-th moment $\mathbb{E}\|\nabla f(x)-\text{clip}(g,\rho)\|^4$ where $g$ is the stochastic gradient. If $\alpha\geq 4$, then we can directly apply the noise assumption and get $\sigma^4$. On the other hand, if $1<\alpha< 4$, then for large $\rho$ this quantity can be bounded by $\mathcal{O}(\sigma^\alpha\rho^{4-\alpha})$ by Lemma B.2 (also refer to Lemma 5.1 in [1]). The subsequent analysis is similar but will lead to a worse dependence on $\rho$. Therefore, our techniques can be easily generalized to the case when $\alpha<4$, but the results will be much more messy, which is also beyond the scope of this paper to show the benefits of local updates in adaptive optimization.
>
> **Q1**: What are the theoretical advantages of adaptive methods over non-adaptive one?
>
> **A3**: In fact, even in centralized case (single machine), we are still not aware of any theoretical convergence result that can show the benefits of adaptive methods over non-adaptive one. Nevertheless, given the strong widespread practical improvements of Adam over SGD, we believe it is a useful first step to show that adaptive methods perform at least as well as SGD. Likely more assumptions would be necessary to show the theoretical improvements. This is still an interesting open question and we hope our work can motivate related researches.
>
> **Q2**: The statement "the first end-to-end convergence guarantee for distributed adaptive algorithms with local iterations" is not accurate.
>
> **A4**: Thanks for sharing these references and we have included them in the revision. Here "end-to-end" refers to the common gradient based local optimizer, instead of the exact proximal operator in [2,3], which is inaccessible in most cases. Also the "adaptive" means Adam-like algorithms. We will make this statement more precise.
>
> [1] Sadiev, Abdurakhmon, et al. "High-probability bounds for stochastic optimization and variational inequalities: the case of unbounded variance." International Conference on Machine Learning. PMLR, 2023.
>
> [2] Li, Hanmin, Kirill Acharya, and Peter Richtarik. "The Power of Extrapolation in Federated Learning." arXiv preprint arXiv:2405.13766 (2024).
>
> [3] Anyszka, Wojciech, et al. "Tighter Performance Theory of FedExProx." arXiv preprint arXiv:2410.15368 (2024).

---

### Official Review · Reviewer_FvoJ · 2024-11-02

**Soundness:** 3
**Presentation:** 3
**Contribution:** 4
**Rating:** 6
**Confidence:** 2

**Summary:**

The paper analysed the Local Adam method, which performs local Adam-like updates and intermittent communication. The paper provides a high probability convergence analysis for local Adam with heavy-tail noise, and demonstrated its advantage against mini-batch sgd in some regimes. The analysis also covers the local SGDM method.

**Strengths:**

The paper is a technical tour de force and the first one to analyse local methods with adaptive updates without some artificial assumption. The algorithm also incorporates the clipping mechanism to deal with the heavy-tailed noise. I think overall this is a pretty impressive display of technical achievement.

**Weaknesses:**

see questions.

**Questions:**

I did not go into the technical details in the appendices so I cannot attest to the correctness of the statements, but I have some questions/concerns with the followings in the main text:

1. In page 2 line 77 the author claimed that "The conventional in-expectation rate seems fail to capture some important properties like heavy/light tailed noise distribution." Can the author please elaborate on this? It seems to me that the in-expectation convergence is stronger than the high probability convergence presented in this paper since one should be able to convert the in-expectation convergence to high probability convergence using Markov inequality.

2. The paper claim to cover the $(L_0,L_1)$ smoothness case, but I fail to see how is this the case? It seems to me that all the theorems presented in the main paper are based on Assumption 2, which is just the usual smoothness assumption.

3. Regarding Assumption 3:
    - I was under the impression that most existing works make their heavy-tail noise assumption in the form: $\mathbb{E}\lVert \nabla F(x,\xi)-\nabla f(x)\rVert^\alpha\leq \sigma^\alpha$, i.e. a upper bound on the norm of the deviation. Here it seems to be an entry-wise upper bound. Can the author please elaborate more on how does the result based on the entry-wise upper bound translate to the usual result based on the usual norm upper bound?
    - Remark 1: it seems a bit curious that the authors does not deal with the case $1<\alpha \leq 2$ which is what most existing works deal with, but instead consider the case with $\alpha \geq 4$? The authors claim that it's easy to extend the analysis to the case where $\alpha < 4$, then why not just include it here which will certainly make the results more easily comparable to the existing results?
    - Regarding the $\lVert \mathbf{\sigma}\rVert_{2\alpha}d^{\frac{1}{2}-\frac{1}{2\alpha}}= O(\sigma)$ assumption in the theorems. This seems to relate to the entry-wise noise bound. Can the authors elaborate on this and its consequences in the convergence rate (e.g. whether or not it's introducing additional dimension dependence compared to other works)?

4. For Theorem 3, can the author please elaborate more on the convergence criteria $\lVert \cdot\rVert^2_{H_r^{-1}}$. I understand that some existing works used convergence criteria similar to this one, but it nonetheless seems a bit unnatural to me. In particular, since $H_r$ is defined with respect to $v_r$ and $\lambda$, algorithm iterates and hyperparameter, it seems that the convergence is algorithm/parameter-dependent. Wouldn't we be able to claim any convergence rate if we set $\lambda$ large enough? Could the authors please explain the choice of $\lambda$ as well?

I would be happy to raise my score if the authors can provide more clarities regarding these questions/concerns.

---

> ### Author Response · Authors · 2024-11-19
>
> Thanks for appreciating our technical contributions! We address your concerns as follows.
>
> **Q1**: The in-expectation convergence is stronger than the high probability convergence due to the Markov inequality.
>
> **A1**: We kindly disagree with the point that in-expectation result is stronger than high probability one. By Markov inequality, the in-expectation bound can indeed be transferred to high probability bound, but with $\mathcal{O}(\frac{1}{\delta})$ depedence, where $\delta$ is the confidence level. This is much worse than our result which has $\mathcal{O}(poly(\log\frac{1}{\delta}))$ dependence. On the other hand, if we have high probability convergence bound depending only on $\log\frac{1}{\delta}$, then we can easily bound the expectation by $\mathbb{E}[X]=\int_0^\infty \mathbb{P}(X\geq x)dx$ for any non-negative random variable $X$. Therefore, the high probability result is generally stronger than the in-expectation result.
>
> **Q2**: All the theorems are based on Assumption 2, which is just the usual smoothness assumption instead of $(L_0,L_1)$ smoothness.
>
> **A2**: We'd like to point out that Assumption 2 is much weaker than the usual smoothness assumption since we only have a bounded smoothness $L$ in a subset $\Omega$ of $\mathbb{R}^d$, where $\Omega$ is usually definded as a sub-level set of the loss function in our theorems. We remark that it is substantially differnt from the usual global smoothness. As have mentioned in line 238-245 and the footmark therein, our assumption can be implied by $(L_0, L_1)$ for a given sub-level set $\Omega$, and in general, it is a non-trival part of our analysis to show that the iterates stay in $\Omega$ with high probably. Hence $(L_0,L_1)$ smoothness can be covered by Assumption 2.
>
> **Q3**: How can entrywise noise bound in Assumption 3 be translated to the usual norm bound? How to interprete the assumption $\||\boldsymbol{\sigma}\||_{2\alpha} d^{\frac{1}{2}-\frac{1}{2\alpha}}=\mathcal{O}(\sigma)$ in terms of dimension dependence?
>
> **A3**: As we aim to deal with entry-wise clipping operator, it is natural to assume entry-wise noise bound instead of common norm bound ([1]). In Assumption 3, when $\alpha\geq 2$, we will have $\mathbb{E}\||\nabla F(x;\xi) - \nabla f(x)\||^\alpha\leq \||\boldsymbol{\sigma}\||^\alpha=\sigma^\alpha$ by Holder inequality, where $\||\cdot\||$ is the $\ell_2$ norm. Additionally, if the noise is isotropic so that $\sigma_i\asymp \sigma_1$ for any $1\leq i\leq d$, then we have $\||\boldsymbol{\sigma}\||\_{2\alpha}=(\sum_i\sigma_i^{2\alpha})^{\frac{1}{2\alpha}}\asymp d^{\frac{1}{2\alpha}}\sigma_1\asymp d^{\frac{1}{2\alpha}-\frac{1}{2}}\sigma$. Therefore, the assumption $\||\boldsymbol{\sigma}\||_{2\alpha} d^{\frac{1}{2}-\frac{1}{2\alpha}}=\mathcal{O}(\sigma)$ does not introduce additional dimension dependence term.
>
> **Q4**: Why to assume $\alpha\geq 4$?
>
> **A4**: As mentioned in Remark 1, in most previous works on high probability convergence, the noise assumptions can be divided into two categories: light tail with bounded exponential moment (sub-gaussian, sub-exponential, bounded), or finite $\alpha$-moment with $1<\alpha\leq 2$. We remark that ***although $\alpha\geq 4$ is rarely used in the literature, it is still substantially weaker than the sub-gaussian type assumption***. And as discussed in Appendix E, if assuming only the polynomial moment is bounded, SGD fails to get a logarithmic dependence on $1/\delta$, where $\delta$ is the confidence level. Therefore, ***$\alpha\geq 4$ is essentially a type of heavy tailed noise assumption.***
>
> In our settings, the assumption $\alpha\geq 4$ is made to make the results much more clean and concise. In the proof, we only use this noise assumption to bound the variance of certain random variables. For instance, in eq (C.44), we need to bound the 4-th moment $\mathbb{E}\|\nabla f(x)-\text{clip}(g,\rho)\|^4$ where $g$ is the stochastic gradient. If $\alpha\geq 4$, then we can directly apply the noise assumption and get $\sigma^4$. On the other hand, if $1<\alpha< 4$, then for large $\rho$ this quantity can be bounded by $\mathcal{O}(\sigma^\alpha\rho^{4-\alpha})$ by Lemma B.2 (also refer to Lemma 5.1 in [1]). The subsequent analysis is similar but will lead to a worse dependence on $\rho$. Therefore, our techniques can be easily generalized to the case when $\alpha<4$, but the results will be much more messy, which is also beyond the scope of this paper to show the benefits of local updates in adaptive optimization.

---

> > ### Comment · Reviewer_FvoJ · 2024-11-21
> >
> > I thank the authors for the clarifications and I will raise the score.
> >
> > I would still strongly encourage the authors to include the case with $\alpha < 4$ and make a detailed comparison to existing results, at least in the appendix. It's better to not exclude results just because they seem less concise.

---

> > > ### Author Response · Authors · 2024-11-22
> > >
> > > Thanks for your helpful feedback. We have adjusted the statement about "easily" in Remark 1 to "but our analysis methods can be extended to the case where $\alpha < 4$ with some additional technical computations". But unfortunately we are not able to provide the full analysis for $\alpha<4$ due the limited time for rebuttal, which we hope can be done as future work. Please let us know if you have any other concerns.

---

> ### Author Response · Authors · 2024-11-19
>
> **Q5**: How to interprete the convergence criteria of Local Adam based on $\|\cdot\|_{H_r^{-1}}$? In what way will it be affected by the choice of $\lambda$?
>
> **A5**: The matrix $H_r$ is indeed algorithmic-dependent and we leave the expression in Theorem 3 in the main text to in order to be consistent with previous papers. Nevertheless, in Theorem D.3 in appendix, we rigorously analyze the bound of $H_r$ and provide the convergence rate of euclidean norm of gradient, which will introduce an additional multiplier $(1+\frac{G_\infty+\sigma_\infty}{\lambda})$. From this we see that large $\lambda$ can indeed enhance the rate. However, we point out that the bound on $H_r$ there is very pessmistic and large $\lambda$ is also not aligned with common practice of Adam. We leave the impact of the choice of $\lambda$ as an interesting future work.
>
> [1] Zhang, Jingzhao, et al. "Why are adaptive methods good for attention models?." Advances in Neural Information Processing Systems 33 (2020): 15383-15393.
>
> [2] Sadiev, Abdurakhmon, et al. "High-probability bounds for stochastic optimization and variational inequalities: the case of unbounded variance." International Conference on Machine Learning. PMLR, 2023.

---

### Author Response · Authors · 2024-11-19
**Global comments**

Thanks for the reviewers' helpful feedback and we are glad that all the reviewers recognize our work as a solid first step of understanging theoretical benefits of Adam-type algorithms with local updates. Since every reviewer mentions the Assumption 3 on $\alpha\geq 4$, we'd like to make some brief global comments here.

As mentioned in Remark 1, in most previous works on high probability convergence, the noise assumptions can be divided into two categories: light tail with bounded exponential moment (sub-gaussian, sub-exponential, bounded), or finite $\alpha$-moment with $1<\alpha\leq 2$. We remark that ***although $\alpha\geq 4$ is rarely used in the literature, it is still substantially weaker than the sub-gaussian type assumption***. And as discussed in Appendix E, if assuming only the polynomial moment is bounded, SGD fails to get a logarithmic dependence on $1/\delta$, where $\delta$ is the confidence level. Therefore, ***$\alpha\geq 4$ is essentially a type of heavy tailed noise assumption.*** Here we use $\alpha\geq 4$ to make the results much more clean and concise.

Please see more details in the individual response. Thanks again for all the reviewers' helpful feedback.

---

### Meta-Review · Area_Chair_tkHN · 2024-12-18

**Metareview:**

The paper analyzes distributed adaptive algorithms with local updates and intermittent communication in the homogeneous setting. The results are derived for a special variant of Adam, Local Adam with gradient clipping, which also reduces to Local SGD with momentum (Local SGDM).

The main result of the paper demonstrates that Local SGDM and Local Adam can outperform their minibatch counterparts in convex and weakly convex settings under certain parameter regimes.

The reviewers noted that the heavy-tail noise assumption with parameter $\alpha \geq 4$ seems somewhat unusual. Additionally, the comments on the $(L_0, L_1)$ assumption were considered slightly misleading, as the analysis applies to any function with bounded smoothness over a bounded set and does not provide specific insights for the class of $(L_0, L_1)$ functions.

On the other hand, the reviewer commended the rigorous analysis and highlighted that the results, which demonstrate the benefits of local steps in SGDM and Local Adam, represent a meaningful contribution to the literature on local update methods and could inspire future research in this area.

**Additional Comments On Reviewer Discussion:**

The reviewers inquired about extending the analysis to the regime $1 < \alpha \leq 2$, which was described as a trivial extension in the original submission. In the rebuttal, the authors clarified that this extension was more challenging than initially implied and did not provide it. The reviewers agreed with the proposed changes to the wording but expressed interest in seeing the derivations. This did not impact the reviewer's overall evaluation of this work and the final decision.

---

### Decision · Program_Chairs · 2025-01-22

Accept (Poster)